# Proteome census upon nutrient stress reveals Golgiphagy membrane receptors

Kelsey L. Hickey[1,2,5], Sharan Swarup[1,2,3,5], Ian R. Smith[1,4], Julia C. Paoli[1,2], Enya Miguel Whelan[1], Joao A. Paulo[1] & J. Wade Harper[1,2✉]

During nutrient stress, macroautophagy degrades cellular macromolecules, thereby providing biosynthetic building blocks while simultaneously remodelling the proteome[1,2]. Although the machinery responsible for initiation of macroautophagy has been well characterized[3,4], our understanding of the extent to which individual proteins, protein complexes and organelles are selected for autophagic degradation, and the underlying targeting mechanisms, is limited. Here we use orthogonal proteomic strategies to provide a spatial proteome census of autophagic cargo during nutrient stress in mammalian cells. We find that macroautophagy has selectivity for recycling membrane-bound organelles (principally Golgi and endoplasmic reticulum). Through autophagic cargo prioritization, we identify a complex of membrane-embedded proteins, YIPF3 and YIPF4, as receptors for Golgiphagy. During nutrient stress, YIPF3 and YIPF4 interact with ATG8 proteins through LIR motifs and are mobilized into autophagosomes that traffic to lysosomes in a process that requires the canonical autophagic machinery. Cells lacking YIPF3 or YIPF4 are selectively defective in elimination of a specific cohort of Golgi membrane proteins during nutrient stress. Moreover, YIPF3 and YIPF4 play an analogous role in Golgi remodelling during programmed conversion of stem cells to the neuronal lineage in vitro. Collectively, the findings of this study reveal prioritization of membrane protein cargo during nutrient-stress-dependent proteome remodelling and identify a Golgi remodelling pathway that requires membrane-embedded receptors.

Mammalian cells remodel their proteomes in response to changes in nutrient stress through transcriptional, translational and degradative mechanisms[1,2]. Central to these responses are proteasomal and autophagy-dependent degradative mechanisms that remove superfluous or damaged organelles and proteins to allow recycling of building blocks for cellular remodelling[1]. Macroautophagy is considered to result in nonspecific capture of bulk cytoplasmic contents within autophagosomes, the biogenesis of which is dependent on the ULK1–FIP200 kinase complex, the VPS34 class III phosphoinositide 3-kinase and ATG8 lipidation machinery, including ATG7 (ref. 4). However, recent work indicates that selective forms of endoplasmic reticulum (ER) degradation by autophagy may be an integral part of the autophagic response to nutrient stress[5–10]. With ER-phagy, multiple partially redundant transmembrane ER proteins function as receptors to recruit core autophagy machinery, including the ULK1–FIP200 kinase complex[6], to initiate phagophore biogenesis proximal to the ER membrane[10]. LC3-interaction regions (LIRs) within these receptors associate with the LIR-docking site (LDS) in lipidated ATG8 proteins (six orthologues in humans—LC3A, LC3B and LC3C (also called MAP1LC3A, MAP1LC3B and MAP1LC3C) and GABARAP, GABARAPL1 and GABARAPL2) to facilitate ER engulfment within the phagophore[10].

Beyond ER-phagy, we have a limited understanding of cargo selectivity during macroautophagy. Ubiquitin-binding cargo receptors that function to recognize ubiquitylated autophagic cargo seem to play limited roles in cargo selection during nutrient stress, although a subset of these have been linked with microautophagy through the endosomal sorting complexes required for transport system[11,12]. As such, several questions have emerged. First, it is unclear which proteins, protein complexes and organelles are susceptible to autophagic degradation during nutrient stress. Second, it is unknown whether there are additional pathways for selective cargo degradation within the macroautophagy program and, if so, how they are regulated. Third, it is unclear how the fraction of protein molecules degraded by autophagy scales with the total abundance of that protein within the cell and across individual subcellular compartments. In short, the degree of selectivity of macroautophagy is unknown. Here we use complementary proteomic approaches to develop a proteome census for nutrient-stress-dependent macroautophagy in mammalian cells, revealing prioritization of Golgi and ER proteins for autophagic recycling and facilitating the identification of membrane-embedded Golgiphagy receptors—YIPF3 and YIPF4—that are also necessary for remodelling of Golgi during in vitro neurogenesis.

[1]Department of Cell Biology, Harvard Medical School, Boston, MA, USA. [2]Aligning Science Across Parkinson's (ASAP) Collaborative Research Network, Chevy Chase, MD, USA. [3]Present address: Casma Therapeutics, Cambridge, MA, USA. [4]Present address: Velia Therapeutics, San Diego, CA, USA. [5]These authors contributed equally: Kelsey L. Hickey, Sharan Swarup. ✉e-mail: wade_harper@hms.harvard.edu

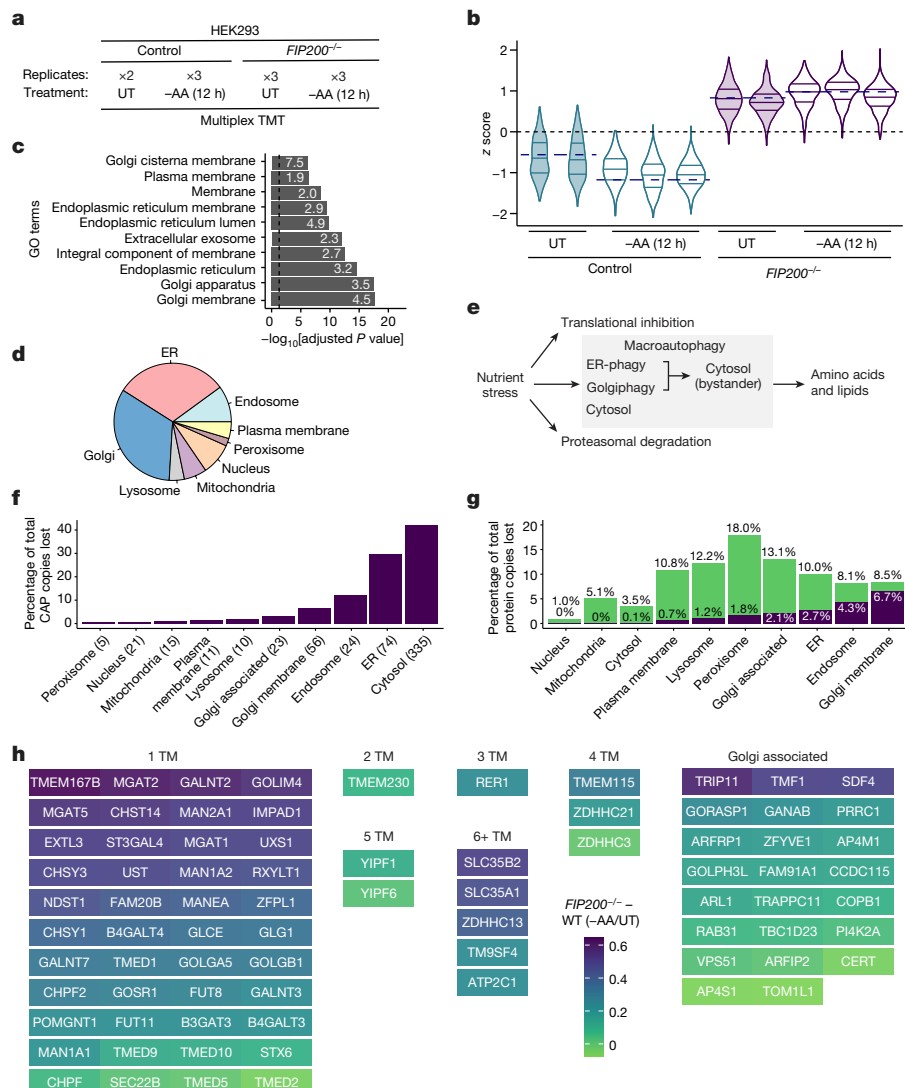

**Fig. 1 | Proteome census reveals Golgi and ER proteins as prioritized clients during macroautophagy. a**, Method for quantifying proteome alterations through autophagy in response to withdrawal of amino acids (−AA). UT, untreated. **b**, Violin plots for proteins identified as CAPs (*n* = 684) in WT and *FIP200⁻/⁻* HEK293 cells without or with amino acid withdrawal (12 h). Navy dashed lines: median value for known autophagy proteins. **c**, Top ten Gene Ontology terms for CAPs from cells subjected to amino acid withdrawal. *P* values were calculated using two-sided Fisher's exact test and adjusted for multiple comparisons using the Benjamini–Hochberg method. **d**, Frequency of proteins with the indicated subcellular localization for the CAPs (amino acid withdrawal). **e**, Schematic depicting selective autophagy within the macroautophagy pathway. See text for details. **f**, Among CAPs, percentage of total protein copy numbers lost upon amino acid withdrawal. **g**, Percentage of all protein copies lost from CAPs (purple) or other mechanisms (green) by amino acid withdrawal for subcellular compartments (1.2829 × 10⁶ total). **h**, Golgi proteins from CAPs coloured by FIP200-dependent turnover during amino acid withdrawal.

## Nutrient stress autophagic profiling

To uncover the selectivity of macroautophagy during nutrient stress, we measured total protein levels in HEK293 cells with or without key autophagy factors (ATG7 or FIP200). Cells were left untreated or subjected to nutrient starvation (Earle's balanced salt solution (EBSS) treatment or amino acid withdrawal) for 12 h before total proteome analysis through tandem mass tagging (TMT) proteomics[13] (Fig. 1a and Supplementary Table 1). The expected downregulation of the mTOR substrate ULK1 and 4EBP1 phosphorylation was observed during nutrient stress (Extended Data Fig. 1a–c). From about 8,000 proteins quantified, we observed starvation and ATG7- or FIP200-dependent reduction in the abundance of several known autophagy receptors (for example, TEX264, CCPG1, CALCOCO1 and SQSTM1) and ATG8 proteins (LC3B and GABARAPL2; Extended Data Fig. 1d–f), allowing us to generate a consensus profile of median abundance changes (Extended

Data Fig. 1d–f). To identify proteins exhibiting a similar abundance profile, we calculated the root-mean-square error (RMSE) from our known autophagy cargo profile for every protein quantified, across all treatments and replicates (Methods and Extended Data Fig. 1g). Proteins with lower RMSE more closely resemble the normalized abundance profile of known autophagy cargo proteins and ideally should be enriched in receptors or clients of autophagy (Extended Data Fig. 1g).

This approach identified 732 and 684 proteins—referred to as candidate autophagy proteins (CAPs)—whose abundance profile is concordant with starvation- and autophagy-dependent turnover, and genetically decoupled from other starvation-dependent responses: decreased abundance with EBSS treatment or amino acid withdrawal that is blocked by deletion of *ATG7* or *FIP200*, respectively (Fig. 1b and Extended Data Fig. 2a–c). Gene Ontology analysis revealed that the top ten Gene Ontology terms for CAPs were enriched in terms related

to ER and Golgi (Fig. 1c and Extended Data Fig. 2d). When CAPs were compared with all other quantified proteins, Golgi and ER were found to be the most over-represented compartments across those examined (Fig. 1d and Extended Data Fig. 2e–j). Most proteins annotated within cytosolic, nuclear, plasma membrane or mitochondrial compartments were above the RMSE cutoff in both the EBSS and amino acid withdrawal treatments, whereas the ER and Golgi compartments exhibited a predominate proportion of their constituent proteins below the RMSE cutoff (Extended Data Fig. 2b,c,h–j). The enrichment of ER and Golgi proteins within CAPs with both EBSS treatment and amino acid withdrawal was particularly striking (Extended Data Fig. 2e,f). Across the two independent experiments with distinct types of nutrient stress, 187 proteins were common to both sets of CAPs. The common proteins, compared with non-overlapping proteins, are even further over-represented in Golgi and ER localization (Extended Data Fig. 2g). Golgi proteins fall into two major classes—Golgi membrane proteins containing one or more transmembrane segments and peripheral Golgi-associated proteins that spend part of their life history in association with Golgi. CAPs were strongly enriched in Golgi membrane proteins with both EBSS and amino acid withdrawal, as compared with Golgi-associated proteins (Extended Data Figs. 2h–j and 3a). Although cytosolic proteins constitute the largest single group of CAPs (>300 proteins), the overlap found with the two types of nutrient stress was substantially less than that seen with Golgi and ER compartments (Extended Data Fig. 3b,c). Thus, selective degradation of Golgi and ER underlies this form of macroautophagy (Fig. 1e).

## Proteome census for macroautophagy

Although the ER and Golgi compartments represent 4.4 and 0.8% of the proteome, respectively, their proteins were markedly more enriched as CAPs compared to the much more abundant cytosolic proteins (59%)[14]. This finding led us to consider how the fraction of protein molecules degraded by autophagy scales with the total abundance of that protein within the cell and across individual subcellular compartments. A priori, abundant cellular complexes might be considered as likely autophagy substrates to provide recycled amino acids without markedly affecting cellular homoeostasis. However, consistent with previous studies[13], our results do not identify abundant cytosolic complexes such as the ribosome and proteasome as CAPs (Extended Data Figs. 2b,c and 3d). This probably reflects the major role of translational suppression and non-autophagic degradation of these proteins during starvation coupled with their very high abundance, such that an insufficient number of protein molecules are degraded by autophagy to score as CAPs[13].

To test how the fraction of molecules degraded by autophagy scales with total protein abundance, we merged estimates for absolute protein abundance and quantitative proteome measurements during starvation with the goal of providing a 'proteome census' for nutrient stress. First, we estimated protein copy number per cell using the proteome ruler method[15] by extrapolating mass spectrometry (MS[1]) signal from relative TMT intensities (Methods) in untreated wild-type (WT) cells. We then inferred each protein's loss in estimated absolute abundance on the basis of the protein's relative fold change upon amino acid withdrawal. Autophagy-dependent protein copy number loss for each cellular compartment spans about 5 orders of magnitude in abundance across about 6,800 proteins quantified, indicating that macroautophagy does not degrade only the most abundant cytosolic, ER and Golgi proteins (Extended Data Fig. 3e,f and Supplementary Table 2). In fact, the abundance rank for CAPs is not substantially different from that for all other proteins, although at the level of subcellular compartments, organelles exhibit differing degrees of selectivity (Extended Data Fig. 3f,g).

On the basis of absolute abundance estimates, we calculated the total number of protein copies per cell that were degraded for CAPs according to their subcellular compartment. Most protein copies degraded, as a percentage of the total CAP molecules lost, are contributed by ER, endosome, Golgi and cytosol, but unexpectedly, the number of protein molecules contributed by ER and Golgi rivals that of the cytosol (Fig. 1f and Extended Data Fig. 4a–j). Given that non-autophagy-based degradation and translational suppression also play a role in determining protein abundance during starvation[12,13], we calculated the fractional contribution of protein abundance loss from each CAP relative to the total abundance loss during starvation for each individual compartments. About 80% of the reduction in protein abundance of Golgi membrane proteins could be attributed to the CAPs that are prioritized for autophagic recycling, with endosomes and the ER also having a substantial amount of protein loss from CAPs (Fig. 1g). By contrast, only about 3% of the changes in the copy number of cytosolic proteins could be attributed to the abundance loss from CAPs (Fig. 1g). Analogous results were obtained when our data were mapped onto absolute abundance estimates previously reported in HEK293T cells[15] or derived from MS data measured by data-independent acquisition, with absolute abundance estimates that correlated well with data herein (Extended Data Fig. 4a–j). Thus, Golgi and ER represent major targets for autophagy in response to nutrient stress with a larger fraction of their individual proteomes being subjected to turnover than that of the cytosol, despite a much larger (>10-fold) copy number of cytosolic proteins[14].

Most Golgi CAPs with amino acid withdrawal contained transmembrane segments, with only a limited number of Golgi-associated proteins (Fig. 1h). In total, 46% (79/172) of proteins classified as being either in Golgi membranes or closely associated with Golgi exhibited features of autophagy clients in response to nutrient stress. Although the endoplasmic reticulum–Golgi intermediate compartment (ERGIC) compartment has been suggested to be a source of membranes for ATG8 lipidation[16], highly validated ERGIC proteins (LMAN1 (also called ERGIC53), LMAN2, ERGIC1 and ERGIC3) as well as COPI/II proteins did not exhibit a proteomic profile consistent with autophagic turnover, consistent with these compartments not being precursors for Golgiphagy (Extended Data Figs. 2h–j and 4k,l).

## Golgiphagy receptor identification

Although several membrane-embedded ER-phagy receptors have been reported, membrane-embedded Golgiphagy receptors are unknown[5–10,17,18]. To search for candidate receptors, we first identified HEK293 and HeLa cell Golgi proteins whose abundance in total proteomes was reduced by nutrient stress (EBSS) in an ATG7-dependent manner (Fig. 2a,b, Extended Data Fig. 5a–h and Supplementary Table 3). In parallel, we used proximity biotinylation in triple-knockout ΔLC3 or ΔRAP HeLa cells[19] reconstituted with WT or LDS-mutant[20] APEX2–LC3B or APEX2–GABARAPL2, respectively, to identify Golgi proteins in proximity to ATG8 in a LIR-dependent manner (Fig. 2c). Cells were left untreated or subjected to nutrient stress (EBSS, 3 h) in the presence of bafilomycin A1 (BafA1) to block lysosomal degradation before proximity biotinylation and proteomics (Fig. 2c,d, Extended Data Fig. 6a–f and Supplementary Table 4). To prioritize candidate receptors, we generated a composite ranking that combines the extent of starvation- and autophagy-dependent degradation with ATG8 interaction for each protein detected across each dataset (Methods and Supplementary Table 5). The utility of this approach is indicated by the presence of TEX264, CCPG1, SQSTM1 and two ATG8 proteins within the top ten ranked proteins (Fig. 2e). The highest-ranked Golgi protein (ranked seventh) was YIPF4 (Fig. 2e), which exhibited strong LDS-dependent enrichment with GABARAPL2 proximity biotinylation and to a lesser extent with LC3B (Fig. 2d and Extended Data Fig. 6e,f). A previous study also reported an LDS-dependent interaction between overexpressed LC3B and two YIPF proteins, YIPF3 and YIPF4, under basal conditions[7]. Although YIPF3 was not detected by

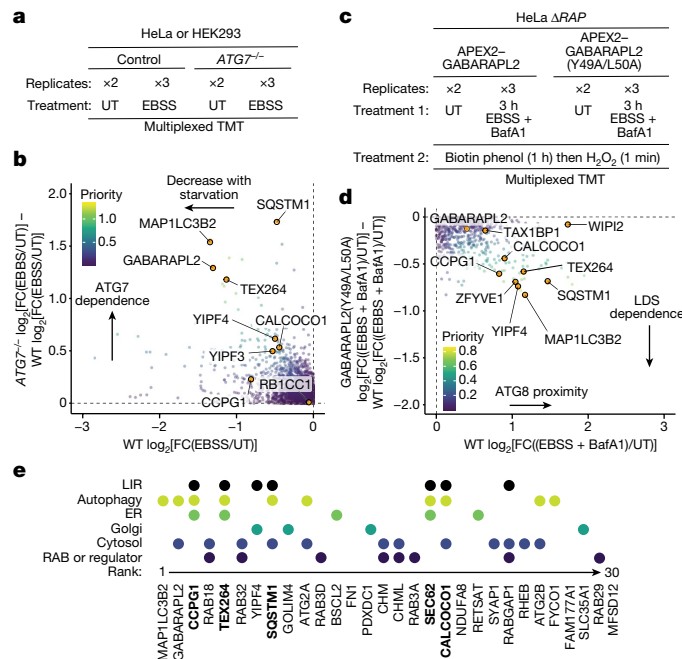

**Fig. 2 | Orthogonal proteomics for Golgiphagy receptor identification.**
**a**, TMT pipeline to measure relative protein abundance during nutrient stress (EBSS) with or without ATG7. **b**, Plots of $ATG7^{-/-}$ $\log_2[FC(EBSS/UT)]$ − WT $\log_2[FC(EBSS/UT)]$ versus WT $\log_2[FC(EBSS/UT)]$, in which FC represents fold change, for HeLa cells treated with EBSS for 18 h with priority for individual proteins scaled on the basis of the inset colour code. Full plots are shown in Extended Data Fig. 5g,h. **c**, Ten-plex TMT APEX2–ATG8 pipeline to capture autophagy receptors during nutrient stress (EBSS + BafA1, 4 h) with or without LDS. At 3 h post-nutrient stress, cells were supplemented with biotin phenol (1 h) and then treated with $H_2O_2$ for 1 min followed by quenching (Methods). **d**, APEX2 proximity labelling plots of GABARAPL2(Y49A/L50A) $\log_2[FC((EBSS + BafA1)/UT)]$ − WT $\log_2[FC((EBSS + BafA1)/UT)]$ versus WT $\log_2[FC((EBSS + BafA1)/UT)]$ in which priority for individual proteins is scaled on the basis of the inset colour code. Full plots are shown in Extended Data Fig. 6e,f. **e**, Top ranked proteins ($n = 30$) on the basis of summed individual rankings for global proteomics and ATG8 proximity biotinylation (Methods) displayed on the basis of their subcellular localization, autophagy involvement and known or candidate LIR motif. Known autophagic cargo receptors are in bold font.

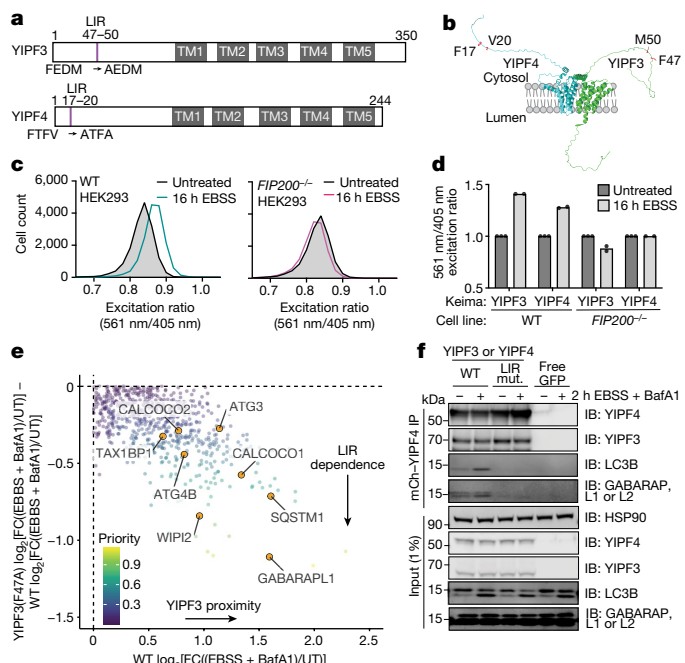

**Fig. 3 | LIR-containing YIPF3 and YIPF4 undergo autophagic flux and associate with autophagy machinery during macroautophagy. a**, Domain structures of YIPF3 and YIPF4 showing the locations of transmembrane segments and N-terminal candidate LIR motifs. Transmembrane domains (TM1–TM5) are shown in grey; single-letter code amino acid sequences for WT and mutant LIR motif labelled below are indicated below. **b**, Colabfold model for the YIPF3–YIPF4 complex. Candidate LIR motifs are shown in red. **c**, Keima–YIPF3 HEK293 cells (± FIP200) were untreated or subjected to nutrient stress for 16 h before flow cytometry. Frequency distributions of 561 nm/405 nm excitation ratios are shown ($n = 10,000$ cells per condition). **d**, Bar graph of median values of the biological duplicate experiments for 561 nm/405 nm excitation ratios for Keima–YIPF3 or Keima–YIPF4 with or without FIP200. **e**, APEX2 proximity labelling plots of YIPF3(F47A) $\log_2[FC((EBSS + BafA1)/UT)]$ − WT $\log_2[FC((EBSS + BafA1)/UT)]$ versus WT $\log_2[FC((EBSS + BafA1)/UT)]$ in which priority for individual proteins is scaled on the basis of the inset colour code. Full plots are shown in Extended Data Fig. 7h,i. **f**, RFP–Trap immunoprecipitates (IP) of WT and LIR-mutant GFP–YIPF3 and mCherry (mCh)–YIPF4 in HEK293 $YIPF3^{-/-}YIPF4^{-/-}$ cells that were untreated or starved of amino acids (2 h + BafA1) were immunoblotted (IB) with the indicated antibodies. L1, GABARAPL1; L2, GABARAPL2. This experiment was repeated in biological triplicate with similar results.

proximity biotinylation and therefore was not prominent in the composite ranking, its abundance profile was similar to those of other bona fide receptors in global proteomics experiments (Fig. 2b and Extended Data Fig. 5g,h). We therefore focused on YIPF3 and YIPF4 as candidate Golgiphagy receptors.

## YIPF3 and YIPF4 interact with ATG8 proteins

YIPF3 and YIPF4 are members of a family of Golgi proteins that contain five transmembrane segments and cytosolic amino-terminal regions harbouring candidate LIRs[21] (Figs. 2e and 3a,b). Although poorly studied, YIPF3 and YIPF4 co-immunoprecipitate when overexpressed and are thought to form heterodimers[22]. ColabFold implementation of AlphaFold[23] predicts a YIPF3–YIPF4 heterodimer, with both N-terminal regions being largely unstructured (Fig. 3b). YIPF3 stability probably requires association with YIPF4, as deletion of *YIPF4* in HeLa cells resulted in loss of YIPF3 (Extended Data Fig. 7a).

To directly examine YIPF3 and YIPF4 as autophagic substrates, we fused the fluorescent Keima protein to YIPF3 and YIPF4 (Extended Data Fig. 7b). Keima undergoes a change in chromophore resting state upon trafficking to the acidic lysosome compartment (pH ≈ 4.5), allowing flux measurements in single cells by flow cytometry[24].

Keima–YIPF3 and Keima–YIPF4 flux increased upon nutrient stress in a $FIP200^{-/-}$-dependent manner (Fig. 3c,d), analogous to observations for membrane-bound ER-phagy receptors[5,7]. To explore the YIPF3–YIPF4 complex interactions during nutrient stress, we stably expressed APEX2–YIPF3 or APEX2–YIPF4 in HeLa cells lacking YIPF3 or YIPF4, respectively, with functional or mutated LIR motifs and carried out proximity biotinylation (EBSS + BafA1, 3 h; Extended Data Fig. 7c–i and Supplementary Table 6). Among the most enriched proteins with YIPF3 and YIPF4 were GABARAPL1, WIPI1/2, ATG3 and ATG4B (Fig. 3e and Extended Data Fig. 7h,i). Interaction with GABARAPL1 was dependent on a functional LIR motif, indicating that YIPF3 and YIPF4 are in proximity to ATG8 proteins during nutrient stress and providing reciprocal validation of ATG8 proximity biotinylation (Fig. 3e and Extended Data Fig. 7h,i).

Autophagic flux in HeLa cells lacking all six ATG8 proteins (ΔLC3 and ΔRAP) can be rescued by a single GABARAP orthologue; however, although LC3 proteins are not generally required for flux[19,25], they are nevertheless incorporated into autophagosomes together with GABARAPs and many cargo receptors associate broadly with both classes of ATG8

orthologues[10]. To confirm the interaction between the YIPF3–YIPF4 complex and ATG8 proteins, we reconstituted *YIPF3⁻/⁻ YIPF4⁻/⁻* HEK293 cells with exogenous copies of WT or LIR-mutant GFP–YIPF3 and mCherry–YIPF4 and confirmed Golgi localization using immunofluorescence (Extended Data Fig. 8a). mCherry–YIPF4 co-precipitated both LC3B and GABARAP proteins during nutrient stress (amino acid withdrawal, 2 h) and the interaction relied on the presence of a functional LIR motif (Fig. 3f). Likewise, ectopically expressed Flag–LC3B associated with YIPF3 and YIPF4 basally and in the context of nutrient stress in a manner that required LIR and LDS functions (Extended Data Fig. 8b). YIPF3 and YIPF4 were degraded in response to nutrient stress in HeLa cells lacking all three LC3 orthologues (ΔLC3) to an extent similar to that seen with WT cells, but were stable in HeLa cells lacking all three GABARAP orthologues (ΔRAP) or HeLa cells lacking all six ATG8 proteins (ΔLC3 and ΔRAP), respectively (Extended Data Fig. 8c), consistent with GABARAP providing an essential role[19].

## YIPF4 mobilization into autolysosomes

Previous studies suggest that ER-phagy receptors promote ER capture through templating of phagophore formation on the ER membrane, with phagophore closure coupled to scission of the ER membrane generating ER within autophagosomes that then fuse with lysosomes[26–28]. To examine YIPF4 behaviour during nutrient stress, we created WT or *FIP200⁻/⁻* HEK293 cells in which the endogenous N terminus of YIPF4 was edited to append a monomeric neon green fluorescent protein (mNEON; Methods and Extended Data Fig. 9a). mNEON–YIPF4 co-localized with the Golgi marker GOLGB1 (Fig. 4a) and showed no obvious *cis* or *trans* Golgi preference on the basis of *cis* (GOLGA2) and *trans* (TGN46) markers (Extended Data Fig. 9b). Strikingly, within 3 h of starvation (EBSS + BafA1), numerous mNEON–YIPF4⁺ and YIPF3⁺ puncta were observed (Fig. 4b and Extended Data Fig. 9c). Notably, a subset of mNEON–YIPF4 puncta were found to co-localize with LAMP1, indicating trafficking to the lysosome (Fig. 4c). Moreover, the appearance of mNEON–YIPF4 puncta required FIP200 and VPS34 (Fig. 4d–f and Extended Data Fig. 9d), suggesting an essential role for autophagy in YIPF3 and YIPF4 capture from Golgi during nutrient stress, as is also seen with ER-phagy receptors[26–28]. Consistent with such a role, our results show that a subset of mNEON–YIPF4 puncta also co-localized with LC3B puncta (Fig. 4g and Extended Data Fig. 9e). We next visualized mNEON–YIPF4 and mCherry–LC3B simultaneously using live-cell imaging. Upon starvation with EBSS + BafA1 (2 h), multiple mNEON–YIPF4 puncta were found to be surrounded by mCherry–LC3 in single confocal slices through the cell (Fig. 4h). Notably, mNEON–YIPF4 puncta track with LC3B signal over several successive frames, consistent with YIPF4 presence within autophagosomes and autolysosomes (Fig. 4i and Supplementary Video 1). Additionally, some autolysosomes have several mNEON–YIPF4 puncta, consistent with YIPF4⁺ autophagosomes merging with a single lysosome (Fig. 4h), as has been seen previously with ER-phagy[13]. There is no evidence of a role for ubiquitylation in this process, as the E1 ubiquitin-activating enzyme inhibitor TAK243 (ref. 29) had no effect on the liberation of mNEON–YIPF4 puncta in response to nutrient stress (Fig. 4j and Extended Data Fig. 9d), and HeLa cells[30] lacking the major ubiquitin-binding autophagy receptors p62, OPTN, NDP52, NBR1 and TAXBP1 exhibited the same extent of YIPF3 and YIPF4 turnover as observed in WT cells (Extended Data Fig. 9f).

## Role of YIPF3 and YIPF4 in Golgiphagy by proteomics

To examine the role of YIPF3 and YIPF4 in proteome remodelling, we included *YIPF4⁻/⁻* HEK293 cells (Extended Data Fig. 7a) in the same TMT proteomics experiment examining FIP200-dependent cargo upon amino acid withdrawal (Fig. 1a, Extended Data Fig. 1b,c and Supplementary Table 1). Although *YIPF4* deletion had little effect on

degradation of non-Golgi proteins, the abundance of 79 Golgi proteins within CAPs was increased, albeit not to the extent seen with *FIP200* deletion (Extended Data Fig. 10a). The contribution of YIPF4 was largely specific to Golgi membrane proteins, with little effect on Golgi-associated proteins (Fig. 5b and Extended Data Fig. 10b). The specificity of YIPF4 for Golgiphagy is further indicated by correlation plots of *YIPF4⁻/⁻* and *FIP200⁻/⁻* cells with or without amino acid withdrawal (Fig. 5b and Extended Data Fig. 10a), in which ER protein abundance was stabilized in *FIP200⁻/⁻* cells but unaffected in *YIPF4⁻/⁻* cells (Fig. 5b and Extended Data Fig. 10a). The landscape of YIPF4-dependent Golgiphagy is compared with FIP200-dependent Golgiphagy clients in Fig. 5c. In total, 30 of 54 Golgi membrane proteins that are stabilized in *FIP200⁻/⁻* cells were also stabilized upon *YIPF4* deletion (*YIPF4⁻/⁻* log₂[FC(−AA/UT)] − WT log₂[FC(−AA/UT)] > 0.2), whereas only 5 out of 23 Golgi-associated proteins were stabilized (Fig. 5c). The results of immunoblotting for a subset of Golgi proteins in *FIP200⁻/⁻* and *YIPF3⁻/⁻ YIPF4⁻/⁻* HEK293 cells were consistent with proteomics data (Extended Data Fig. 10c,d). To further verify these findings, we created HeLa cells lacking YIPF3 or YIPF4 and compared their proteomes with *ATG7⁻/⁻* HeLa cells in response to EBSS (Extended Data Fig. 10e and Supplementary Table 7). Consistently, organelle correlation plots of *YIPF3⁻/⁻* or *YIPF4⁻/⁻* versus *ATG7⁻/⁻* from HeLa cells also show selectivity for Golgi membrane protein turnover during nutrient stress (Extended Data Fig. 10e). Thus, YIPF3 and YIPF4 act as selective Golgiphagy receptors in two different cell lines.

GALNT2 was among the most strongly stabilized Golgi cargo in cells lacking FIP200 or YIPF4 (Fig. 5c). Consistent with a role in Golgiphagy, a reduced flux of GALNT2 tagged with a Keima reporter was observed in *YIPF3⁻/⁻ YIPF4⁻/⁻* and *FIP200⁻/⁻* cells upon starvation (Extended Data Fig. 10f). We note that the number of ATG9⁺ vesicles and Golgi morphology were largely unaffected by deletion of *YIPF3* and *YIPF4* (Extended Data Fig. 11a), and YIPF3 and YIPF4 are not detected in ATG9-containing vesicles[31]. Thus, YIPF3 and YIPF4 proteins act as an autophagy receptor that facilitates the turnover of a cohort of Golgi proteins during nutrient starvation.

The ubiquitin-binding autophagy adaptor CALCOCO1 has been reported to contribute to Golgi and ER turnover during nutrient stress[17,18]. However, YIPF3, YIPF4 and the ER-phagy receptor TEX264 were degraded in *CALCOCO1⁻/⁻* HeLa cells to an extent similar to that seen in control cells, but degradation was blocked in *ATG7⁻/⁻* cells (Extended Data Fig. 11b). Proteomic analysis of *CALCOCO1⁻/⁻* cells in response to EBSS revealed an extent of Golgi membrane protein turnover comparable to that of control cells, whereas *YIPF4⁻/⁻* cells in the same experiment exhibited the expected stabilization of Golgi membrane proteins (Extended Data Fig. 11c and Supplementary Table 8). Finally, CALCOCO1 turnover in response to nutrient stress did not depend on YIPF3 and YIPF4 (Extended Data Fig. 11b). These data indicate that if CALCOCO1 is involved in Golgi membrane turnover by autophagy, the mechanism is distinct from that regulated by YIPF3 and YIPF4.

## Golgiphagy during neuronal differentiation

Conversion of human embryonic stem (ES) cells to induced neurons (iNeurons) in vitro is associated with remodelling of both ER and Golgi through autophagy, as assessed using proteomics in *ATG12⁻/⁻* cells[32]. In this context, YIPF3 and YIPF4 were among the most stabilized Golgi proteins[32]. Therefore, to examine the potential involvement of YIPF3 and YIPF4 in Golgi remodelling beyond nutrient stress, we created *YIPF4⁻/⁻* human ES cells, differentiated control, *ATG12⁻/⁻*, and *YIPF4⁻/⁻* human ES cells into iNeurons, and quantified proteomes at days 0 and 12 (Fig. 5d, Extended Data Fig. 12a–c and Supplementary Table 9). The expected alterations in the abundance of pluripotency and neurogenesis factors when comparing human ES cells with iNeurons were observed in all genotypes, indicating that *ATG12* or *YIPF4*

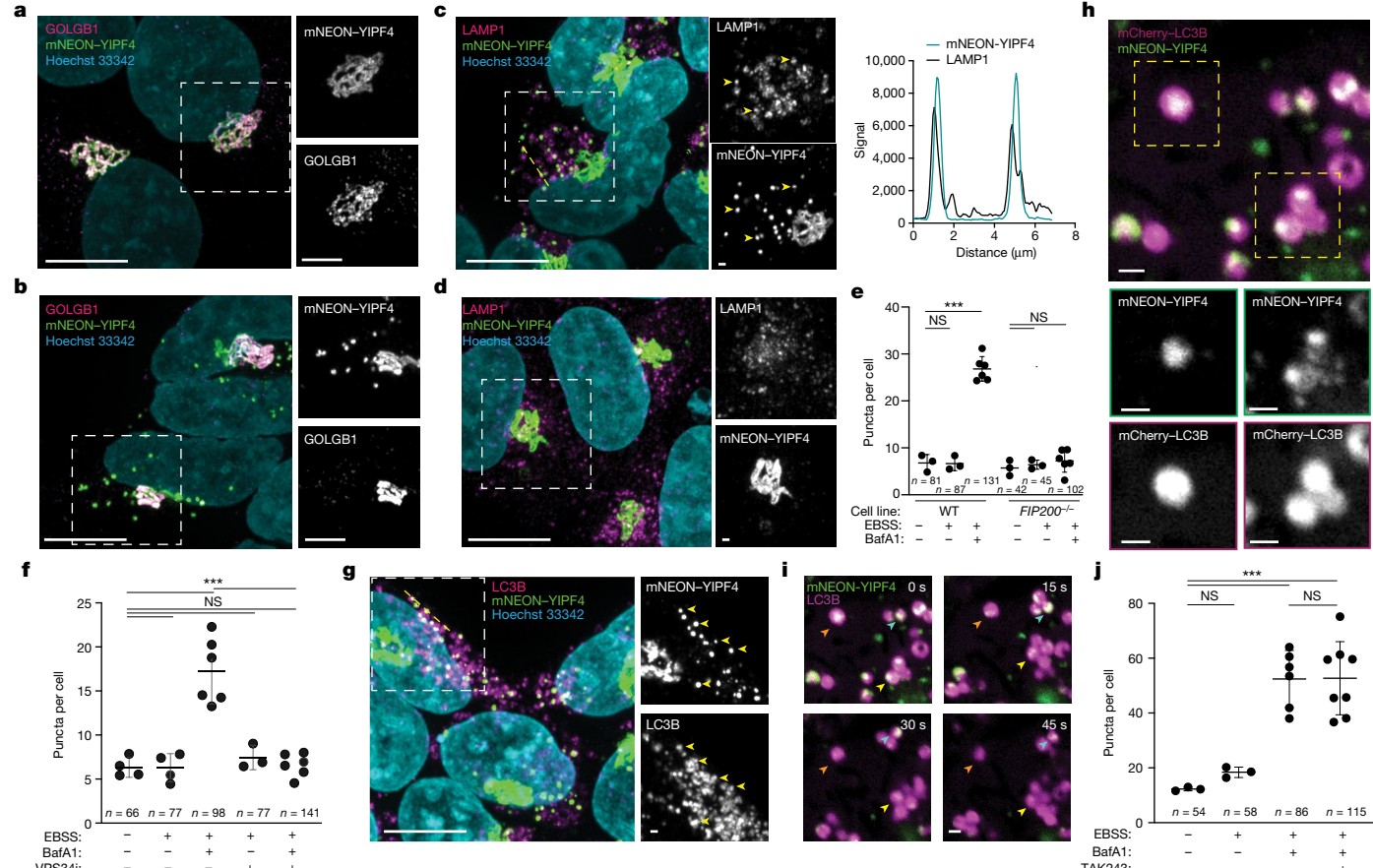

**Fig. 4 | YIPF4 mobilization into autophagosomes during nutrient stress.**
**a,b,** Confocal micrographs of HEK293 cells expressing endogenous mNEON–YIPF4 co-stained with GOLGB1 (magenta) with (**b**) or without (**a**) nutrient stress (3 h with BafA1). Hoechest 33342 labels nuclei (cyan). Scale bars, 5 μm (right) and 10 μm (left). **c,** Cells as in **b** immunostained with anti-LAMP1 (magenta). Line scan region indicated with dashed yellow line, LAMP1-positive mNEON–YIPF4 puncta indicated with yellow arrowheads (left). Line scans for LAMP1 and mNEON signal as a histogram (right). Scale bars, 1 μm (right) and 10 μm (left). **d,** As in **c** but using *FIP200*[−/−] cells. **e,** Number of mNEON–YIPF4 puncta per cell for the indicated treatments in cells ± FIP200. Each dot represents one image in which mNEON and nuclei were counted. ***$P < 0.05$ (two-tailed Mann–Whitney test); Left to right: $P > 0.9999, P = 0.0238, P = 0.7, P = 0.318$. Lines, mean values; error bars, s.d. NS, not significant. $n$ = total number of cells analyzed for each condition. **f,** mNEON–YIPF4 puncta in cells treated as in **b** but with or without addition of VPS34i were quantified as in **e**. Two-tailed Mann–Whitney *P* values

from left to right: $P = 0.8857, P = 0.0095, P = 0.4, P = 0.6095, P = 0.0022$. Lines, mean values; error bars, s.d. NS, not significant. $n$ = total number of cells analyzed for each condition. **g,** Cells treated as in **b** were immunostained with anti-LC3B (magenta). Yellow arrowheads indicate YIPF4[+] puncta overlapping LC3B[+] structures. Scale bars, 1 μm (right) and 10 μm (left). **h,i,** HEK293 cells expressing mNEON–YIPF4 and mCherry–LC3B were subjected to live-cell confocal microscopy 2 h post EBSS treatment and single confocal slices through cells are shown. The time series in **i** shows coincident movement of mNEON and mCherry signal over successive frames (arrowheads). Scale bars, 1 μm. **j,** Number of mNEON–YIPF4 puncta per cell for the indicated treatments in cells 3 h post EBSS, quantified as in **e**. Lines, mean values; error bars, s.d. NS, not significant. ***$P < 0.05$ (two-tailed Mann–Whitney test). From left to right: $P = 0.1, P = 0.0238, P = 0.0121, P > 0.9999$. $n$ = total number of cells analyzed for each condition. For all micrographs (**a–d,g,i**), all experiments were carried out in biological triplicate with similar results.

deletion did not alter differentiation (Methods, Extended Data Fig. 12c and Supplementary Table 9). Consistent with observations in HeLa cells, YIPF3 levels in ES cells lacking YIPF4 were reduced (Extended Data Fig. 12d). As expected[32], we observed accumulation of ER and Golgi proteins in *ATG12*[−/−] cells through differentiation (Extended Data Fig. 12b). Strikingly, *YIPF4*[−/−] iNeurons exhibited selective accumulation of Golgi membrane proteins to an extent approaching that observed in *ATG12*[−/−] iNeurons (Fig. 5e and Extended Data Fig. 12b,e), and with a pattern of accumulation similar to that of nutrient-stress-derived CAPs (Fig. 5f). These results highlight broader functions of YIPF3 and YIPF4 as autophagy-based Golgi remodellers in response to both nutrient stress and cell state changes.

## Discussion

Although macroautophagy is often considered to target bulk cytosol non-selectively[1], our proteome census suggests an alternative model

wherein targeted degradation of ER and Golgi constitute major programs within macroautophagy (Fig. 1e). ER and Golgi collectively account for about 6% of protein copies per cell[14], but the subset of their proteins within CAPs account for about 50% of all CAP protein copies lost (Fig. 1f,g and Extended Data Fig. 4d–h), despite a much larger total copy number for cytosolic proteins (about 59% of cellular proteome[14]; Fig. 1e). Golgi-resident YIPF3 and YIPF4 proteins fulfil the criteria of selective Golgiphagy receptors: interaction with ATG8s, autophagosomal capture, degradation by autophagy and necessary for signal-dependent degradation of a cohort of primarily Golgi membrane proteins. The correlation between Golgi cargo stabilization in starvation and neuronal differentiation systems suggests a common biochemical program for selection of proteins for turnover by Golgiphagy. The data reported here can be explored using our Cellular Autophagy Regulation and GOlgiphagy (CARGO) web resource (Extended Data Fig. 13). After publication of our preprint[33] describing the identification of YIPF3 and YIPF4 as Golgiphagy receptors and during review of the

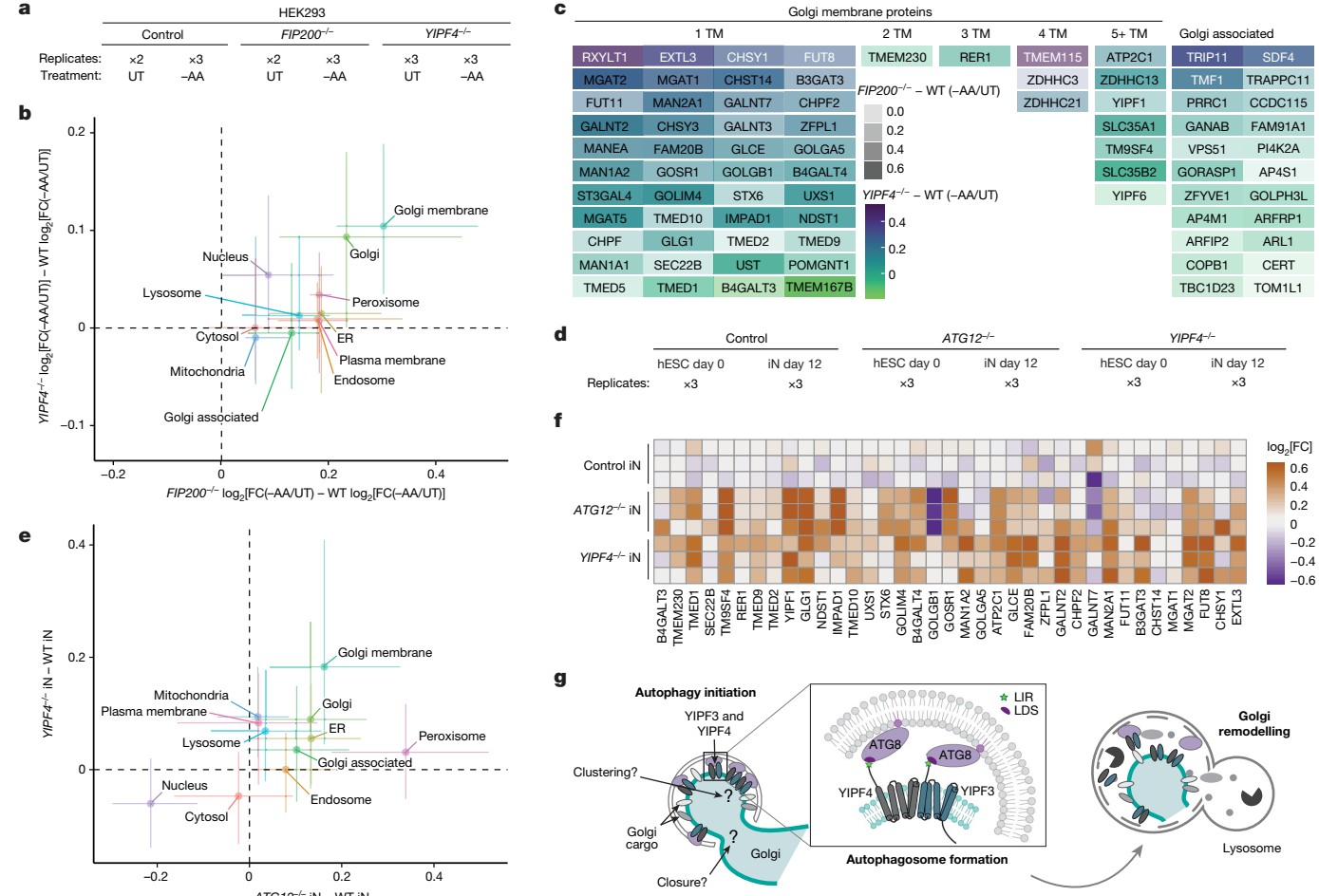

**Fig. 5 | YIPF3 and YIPF4 mediate the autophagy-based recycling of Golgi proteins during nutrient stress and neuronal differentiation in vitro.**
**a**, Method for global proteome alterations through YIPF4 or FIP200 in response to nutrient stress. **b**, Correlation plot of CAPs for alterations in protein abundance for the indicated subcellular compartments during amino acid withdrawal for *YIPF4*$^{-/-}$ − WT cells (*y* axis) versus *FIP200*$^{-/-}$ − WT cells (*x* axis). Points are the median of each distribution, and lines represent the 25–75% quantile. **c**, Classification of Golgi proteins that exhibit YIPF4- or FIP200-dependent degradation in response to amino acid withdrawal (12 h), with the number of transmembrane segments for each membrane protein, as well as Golgi-associated proteins, shown. Grey density scale, FIP200 dependence; colour scale, YIPF4 dependence. **d**, Workflow for analysis of *ATG12*$^{-/-}$ and *YIPF4*$^{-/-}$ iNeurons (iN). **e**, Correlation plot of CAPs for alterations in protein abundance for the indicated subcellular compartments during in vitro differentiation for *YIPF4*$^{-/-}$ − WT iNeurons (*y* axis) versus *ATG12*$^{-/-}$ − WT iNeurons (*x* axis). Points are the median of each distribution, and lines represent the 25–75% quantile. **f**, Heatmap of log$_2$[FC] values from *ATG12*$^{-/-}$ and *YIPF4*$^{-/-}$ iNeurons for the indicated proteins identified as Golgi CAPs in response to nutrient stress. **g**, Model of YIPF3- and YIPF4-mediated Golgiphagy upon nutrient starvation. Aspects of how YIPF3, YIPF4 and other Golgi cargo are selected for capture as well as how autophagic Golgi vesicles are formed remain to be delineated, as indicated by a question mark.

revised version of this work, a related preprint[34] was posted. The data in the latter preprint support the role of YIPF3 and YIPF4 as Golgiphagy receptors in response to nutrient stress.

LIR motifs in transmembrane ER-phagy receptors are thought to concentrate in 'bud-like' nanodomains that can recruit FIP200–ULK1 and/or ATG8 proteins to nucleate phagophore assembly in situ[10,28], and we propose an analogous mechanism for Golgiphagy (Fig. 5g). Further work is required to elucidate biochemical mechanisms underlying YIPF3- and YIPF4-dependent Golgi capture, upstream signals that may initiate the process and any links with Golgi quality control associated with misfolded secretory proteins, as observed with ER-phagy[10]. As with ER-phagy and in light of the differential turnover of Golgi proteins in *FIP200*$^{-/-}$ and *YIPF3*$^{-/-}$ *YIPF4*$^{-/-}$ cells in response to nutrient stress, it seems likely that additional Golgiphagy receptors exist. In addition, our data suggest that Golgiphagy is distinct from the proposed involvement of ERGIC or Golgi as a lipid source for autophagosomes and is not linked in an obvious way with secretory pathways. This raises the question of why macroautophagy prioritizes membrane-bound organelles. We speculate that the preference for ER and Golgi reflects an evolutionarily

programmed pathway that prioritizes the recycling of lipids as well as proteins during nutrient stress.

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

# Methods

## Reagents

**Antibodies.** Antibodies were as follows: ATG7 (Cell Signaling Technology, 8558S; RRID: AB_10831194; dilution 1:1,000), FIP200 (Cell Signaling Technology, 12436; RRID: AB_2797913; dilution 1:1,000), LC3B (MBL International, M186-3; RRID: AB_10897859; dilution 1:1,000), ULK1 (Cell Signaling Technology 8054; RRID: AB_11178668; dilution 1:1,000), phospho-ULK1 (ser757) (Cell Signaling Technology 14202; RRID: AB_2665508; dilution 1:1,000), 4EBP1 (Cell Signaling Technology 9644; RRID: AB_2097841; dilution 1:1,000), phospho-4EBP1 (Thr37/46) (Cell Signaling Technology 2855; RRID: AB_560835; dilution 1:1,000), TEX264 (Sigma, HPA017739; RRID: AB_1857910; dilution 1:1,000), tubulin (Abcam, ab7291; RRID: AB_2241126; dilution 1:1,000), YIPF3 (Invitrogen PA566621; RRID: AB_2664704; dilution 1:1,000), YIPF4 (Sino Biological 202844-T46; dilution 1:1,000), HSP90 (Proteintech 60318; RRID: AB_2881429; dilution 1:1,000), CALCOCO1 (Abclonal A7987; RRID: AB_2768684; dilution 1:1,000), LAMP1 (Cell Signaling Technology 9091; RRID: AB_2687579; dilution 1:1,000), GOLGB1 (also known as giantin; abcamab37266; RRID: AB_880195; dilution 1:1,000), GOLGA2 (Proteintech 11308; RRID: AB_2919024; dilution 1:1,000), PCNA (Santa Cruz PC10; sc-56 RRID: AB_628110; dilution 1:1,000), IRDye 800CW goat anti-rabbit IgG H+L (LI-COR, 925-32211; RRID: AB_2651127; dilution 1:10,000), IRDye 680RD goat anti-mouse IgG H+L (LI-COR, 926-680; RRID: AB_10956588; dilution 1:10,000), goat anti-rabbit IgG, HRP-linked IgG (Cell Signaling Technology 7074P2, RRID: AB_2099233 dilution 1:10,000), goat anti-rabbit IgG HRP conjugate (Bio-Rad 1706515; RRID: AB_11125142; dilution 1:10,000), goat anti-mouse IgG HRP conjugate (Bio-Rad 1706516; RRID: AB_11125547; dilution 1:10,000), goat anti-mouse IgG (H+L) cross-adsorbed secondary antibody, Alexa Fluor 568 (A-11004; RRID: AB_2534072), goat anti-rabbit IgG (H+L) cross-adsorbed secondary antibody, Alexa Fluor 647 (A-21244; RRID: AB_2535812).

**Chemicals, peptides and recombinant proteins.** The following were used: FluoroBrite Dulbecco's modified Eagle's medium (DMEM; Thermo Fisher, A1896701), benzonase nuclease HC (Millipore, 71205-3), urea (Sigma, catalogue number U5378), sodium dodecyl sulfate (Bio-Rad, catalogue number 1610302), high-glucose and high-pyruvate DMEM (Gibco/Invitrogen, 11995), low-glucose DMEM without amino acids (US Biological, D9800-13), TCEP (Gold Biotechnology), puromycin (Gold Biotechnology, P-600-100), protease inhibitor cocktail (Sigma-Aldrich, P8340), PhosSTOP (Sigma-Aldrich, 4906845001), trypsin (Promega, V511C), LysC (Wako Chemicals, 129-02541), EPPS (Sigma-Aldrich, catalogue number E9502), 2-chloroacetamide (Sigma-Aldrich, C0267), TMT 11plex Label Reagent (Thermo Fisher, catalogue numbers 90406 and A34807), TMTpro 16plex Label Reagent (Thermo Fisher, catalogue number A44520), hydroxylamine solution (Sigma catalogue number 438227), Empore SPE Disks C18 (3M - Sigma-Aldrich catalogue number 66883-U), Sep-Pak C18 Cartridge (Waters catalogue numbers WAT054960 and WAT054925), SOLA HRP SPE Cartridge, 10 mg (Thermo Fisher, catalogue number 60109-001), High-pH Reversed-Phase Peptide Fractionation Kit (Thermo Fisher, catalogue number 84868), Bio-Rad Protein Assay Dye Reagent Concentrate (Bio-Rad, catalogue number 5000006) and EBSS (Sigma-Aldrich catalogue number E3024).

## Cell lines

HEK293 (human embryonic kidney, fetus, ATCC CRL-1573, RRID: CVCL_0045) and HeLa (cervical carcinoma cell line CCL-2; RRID: CVCL_0030) cells were grown in high-glucose and high-pyruvate DMEM supplemented with 10% fetal calf serum and maintained in a 5% $CO_2$ incubator at 37 °C. Cell line authentication was provided by the vendor, and karyotyping (GTG-banded karyotype) of HEK293 cells (from ATCC) was also carried out by the cytogenomics core laboratory at Brigham and Women's Hospital. Cells were maintained at <80% confluency throughout the course of experiments. HeLa cells lacking MAP1LC3 ($\Delta LC3$) or GABARAP ($\Delta RAP$) proteins were from a previous study[19]. Culture of human ES cells or iNeurons was carried out as described at https://doi.org/10.17504/protocols.io.br9em93e. In brief, human ES cells (H9, WiCell Institute) with *TRE3G-NGN2* integrated into the AAVS site have been previously described[32] and were cultured in E8 medium on Matrigel-coated plates. To generate iNeurons (i3-neurons) from ES cells, cells were plated at $2 \times 10^5$ cells per millilitre on day 0 on plates coated with Matrigel in ND1 medium (DMEM/F12, 1× N2 (Thermo Fisher), human brain-derived neurotrophic factor (10 ng ml$^{-1}$, PeproTech)), human neurotrophin-3 NT3 (10 ng ml$^{-1}$, PeproTech), 1× nonessential amino acids, human laminin (0.2 µg ml$^{-1}$) and doxycycline (2 µg ml$^{-1}$). The medium was replaced with ND1 the next day. The next day, the medium was replaced with ND2 neurobasal medium, 1× B27, 1× Glutamax, brain-derived neurotrophic factor (10 ng ml$^{-1}$), NT3 (10 ng ml$^{-1}$) and doxycycline (2 µg ml$^{-1}$). On days 4 and 6, 50% of the medium was changed with fresh ND2. On day 7, cells were replated at $4 \times 10^5$ cells per well in ND2 medium supplemented with Y27632 (rock inhibitor; 10 µM). The medium was replaced the next day with fresh ND2 and on day 10 onwards 50% medium change was carried out until the experimental day (day 14 of differentiation unless otherwise noted). Pluripotency and neurogenesis markers exhibited the expected changes for all genotypes (Extended Data Fig. 12c) and visual inspection demonstrated the expected pattern of axons and dendrites for all genotypes.

**Nutrient starvation experiments.** Cells were plated in 10-cm or 15-cm, 6-well dishes the night before nutrient stress. DMEM was removed and cells were washed three times with DPBS and then resuspended in EBSS or DMEM lacking amino acids prepared as described previously[5] (and in https://doi.org/10.17504/protocols.io.yxmvm32nbl3p/v1). For whole-cell proteomics experiments, cells were resuspended in EBSS or medium lacking amino acids as described previously[7] for 12–18 h. For APEX2 proximity labelling and imaging experiments, cells were resuspended in EBSS + BafA1 (100 nM) for 3–4 h in the presence or absence of the indicated inhibitors.

## CRISPR–Cas9 gene editing

*YIPF4-*, *FIP200-* and *ATG7-*knockout in HEK293 cells and *ATG7-*, *YIPF4-* and *CALCOCO1-*knockout in HeLa cell lines were carried out by plasmid-based transfection of Cas9–gRNA using the pX459 plasmid as described previously[35] and at https://doi.org/10.17504/protocols.io.6qpvr3462vmk/v1. The following gRNAs, designed using the CHOPCHOP website (http://chopchop.cbu.uib.no/), were used: YIPF4: 5′-ATCTCGCGGCGACTCCCAAC-3′ and 5′-CGGCCTATGCCCCCACTAAC-3′; FIP200: 5′-ACTACGATTGACACTAAAGA-3′; ATG7 HEK293: 5′-ATCCAAGGCACTACTAAAAG-3′; CALCOCO1: 5′-AAGTTGACTCCACCACGGGA-3′ and 5′-CTAAGCCGGGCACCATCCCG-3′; YIPF3: 5′-CCATTTCGGGCGCCGCCCGC-3′ and 5′-GGCGGCGCCCGAAATGGAGC-3′. Puromycin selection was carried out 24–48 h after the transfection. Cells were given a day to recover from puromycin selection, and then single cells were sorted into a 96-well plate using fluorescence-activated cell sorting (FACS) on a SONY SH800S sorter. Individual clones were screened for deletion of the relevant gene by immunoblotting cell extracts with antibodies specific to the designed gene product. For N-terminal tagging of the *YIPF4* locus, the gRNA 5′-TCGCCGCGAGATGCAGCCTC-3′ was cloned into pX459 and co-transfected with a repair template containing an mNEON Green cassette flanked by homology arms (pSMART-mNEON-YIPF4) into HEK293 and HEK293 *FIP200*$^{-/-}$ cells using Lipofectamine 3000 (as described at https://dx.doi.org/10.17504/protocols.io.5jyl8pj9dg2w/v1). After 7 days, a population of cells for both genotypes was sorted for the same level of mNEON Green signal. For deletion of *YIPF4* in human ES cells (H9), gRNA (5′-AAGAGGTTATGGCTGGCTTC-3′) was ordered from Synthego. A 0.6 µg quantity of sgRNA was incubated with 3 µg SpCas9 protein for 10 min at room temperature and electroporated into $2 \times 10^5$

H9 cells using the Neon transfection system (Thermo Fisher) as described at https://doi.org/10.17504/protocols.io.rm7vzxy44gx1/v1. Out-of-frame deletions were verified by DNA sequencing with Illumina MiSeq and by immunoblotting. All cell lines were demonstrated to be mycoplasma negative.

## Cell lysis and immunoblotting assay

A protocol for cell lysis and immunoblotting can be found at: https://doi.org/10.17504/protocols.io.4r3l226e4l1y/v1. Cells were cultured in the presence of the corresponding stress to 60–80% confluency in 10-cm or 15-cm, 6-well dishes. After the medium was removed, the cells were washed with DPBS three times. To lyse cells, urea buffer (8 M urea, 50 mM Tris pH 7.5, 150 mM NaCl, containing mammalian protease inhibitor cocktail (Sigma), PhosSTOP, and 20 units per millilitre of Benzonase (Millipore)) was added directly onto the cells. Cell lysates were collected by cell scrapers and sonicated on ice for 10 s at level 5, and lysates were cleared by centrifugation (15,000 r.p.m., 10 min at 4 °C). The concentration of the supernatant was measured by the BCA assay. For immunoblotting, the whole-cell lysate was denatured by the addition of LDS sample buffer supplemented with 100 mM dithiothreitol (DTT), followed by boiling at 95 °C for 5 min. A 10–20 µg quantity of each lysate was loaded onto a 4–20% Tris-Glycine gel (Thermo Fisher) or a 4–12% NuPAGE Bis-Tris gel (Thermo Fisher), followed by SDS–PAGE with Tris-glycine SDS running buffer (Thermo Fisher) or MOPS SDS running buffer (Thermo Fisher), respectively. For chemiluminescence western blots, the proteins were electro-transferred to PVDF membranes (0.45 µm, Millipore), and then the total protein was stained using Ponceau (Thermo Fisher). The membrane was then blocked with 5% non-fat milk (room temperature, 60 min) incubated with the indicated primary antibodies (4 °C, overnight), washed three times with TBST (total 30 min), and further incubated with either HRP-conjugated anti-rabbit or anti-mouse secondaries (1:5,000) for 1 h. After a thorough wash with TBST for 30 min, membranes were treated with Lightning Plus Chemiluminescence Reagent (PerkinElmer, NEL104001EA) after mixing the Enhanced Luminol Reagent and the Oxidizing Reagent 1:1. Mixed Chemiluminescence Reagent was added to the blot and incubated with gentle rocking for 1 min before imaging of the blot using the Bio-Rad ChemiDoc Imaging System. For the LI-COR western blots, the proteins were electro-transferred to nitrocellulose membranes and then the total protein was stained using Ponceau (Thermo Fisher). The membrane was then blocked with LI-COR blocking buffer at room temperature for 1 h. Then membranes were incubated with the indicated primary antibodies (4 °C, overnight), washed three times with TBST (total 30 min), and further incubated with either fluorescent IRDye 680RD goat anti-Mouse IgG H+L, or IRDye 800CW goat anti-rabbit IgG H+L secondary antibody (1:10,000) at room temperature for 1 h. After a thorough wash with TBST for 30 min, the near-infrared signal was detected using an OdysseyCLx imager and quantified using ImageStudioLite (LI-COR).

## mCherry–YIPF4 and Flag–LC3B immunoprecipitation

Detailed protocols can be found at https://doi.org/10.17504/protocols.io.8epv5xj9ng1b/v1. Double-knockout (*YIPF3$^{-/-}$ YIPF4$^{-/-}$*) HEK293 cells were reconstituted with mCherry–YIPF4 (WT or LIR mutant) and GFP–YIPF3 (WT or LIR mutant) constructs and sorted for equal expression levels. Immunofluorescence was used to confirm proper localization of both YIPF3 and YIPF4. Then cells were plated on 10-cm plates and grown to 70% confluency. Cells were left untreated or starved using amino acid withdrawal for 2 h in the presence of BafA1 (100 nM). Cells were washed twice with cold PBS and then lysed in 0.8 ml NP-40 lysis buffer (100 mM Tris pH 7.4, 150 mM KCl, 0.1% NP-40, 0.5 mM EDTA, 1× HALT (Roche) protease inhibitors, PhosSTOP tabs). A 1.5 mg quantity of protein from each sample was added to 15 µl of washed RFP–TRAP beads (ChromoTek, number rta) and incubated for 2 h while rotating at 4 °C. Beads were washed three times with lysis buffer and eluted in

1× LDS loading dye at 94 °C for 5 min. For Flag–LC3B immunoprecipitation, 1.5 mg of protein from each sample was added to 20 µl of washed Pierce anti-Flag beads (number A36797) and incubated for 2 h while rotating at 4 °C. Beads were washed three times with lysis buffer and eluted in 1× LDS loading dye at 94 °C for 5 min.

## Flow cytometry for Keima analysis

A detailed protocol can be found at https://doi.org/10.17504/protocols.io.yxmvm3y8nl3p/v1. Corresponding cells were plated onto 96-well plates 1 day before the nutrient stress. The cells were washed twice with PBS and resuspended in DMEM or EBSS to start the 16-h starvation. After starvation, cells were treated with trypsin and quenched with phenol red-free DMEM. Cells were filtered and analysed by flow cytometry (Attune NxT, Thermo Fisher) using the high-throughput autosampler (CyKick). The data were processed by FlowJo software and plotted using GraphPad Prism.

## Confocal microscopy

Protocols for microscopy can be found at https://doi.org/10.17504/protocols.io.5jyl8pj9dg2w/v1. For fixed cells, cells were plated onto 18- or 22-mm glass coverslips (No. 1.5, 22 × 22-mm glass diameter, VWR 48366-227) the day before nutrient stress. DMEM was removed and cells were washed three times with DPBS, followed by resuspension in EBSS with the appropriate inhibitor(s) (SAR405, BafA1, TAK243). After starvation treatment, cells were fixed using 4% PFA followed by permeabilization with 0.5% Triton-X100. Cells were blocked in 3% BSA for 30 min, followed by incubation in primary antibodies (1:200 dilution) for 1 h at room temperature. Cells were washed three times with DPBS + 0.02% Tween-20, followed by incubation in secondary (Alexafluor conjugated 1:200 dilution) secondary antibodies for 1 h at room temperature. Coverslips were then washed three times with DPBS and 0.02% Tween-20 and mounted onto glass slides using mounting medium (Vectashield H-1000) and sealed with nail polish. The cells were imaged using a Yokogawa CSU-W1 spinning-disc confocal system on a Nikon Ti motorized microscope equipped with a Nikon Plan Apo 100×/1.40 NA objective lens, and a Hamamatsu ORCA-Fusion BT CMOS camera. For the analysis, equal gamma, brightness and contrast were applied for each image using FiJi software. For quantification, at least three separate images were quantified for the number of mNEON puncta and nuclei. For live cells, mCherry–LC3B was integrated into HEK293 cells containing an endogenous mNEON tag on YIPF4. Cells were selected with puromycin to obtain a pure population. After selection, cells were plated onto glass-bottom dishes the day before imaging. A 2 h before imaging, DMEM was removed, and cells were resuspended in EBSS to initiate autophagy. The cells were imaged using a Yokogawa CSU-W1 spinning-disc confocal system on a Nikon Ti motorized microscope equipped with a Nikon Plan Apo 100×/1.40 NA objective lens, and a Hamamatsu ORCA-Fusion BT CMOS camera, and a live-cell chamber with temperature and carbon dioxide control. For analysis, equal gamma, brightness and contrast were applied for each image using FiJi software. Quantification of the number of ATG9 puncta (objects per cell) was carried out on four or more biological replicates using Cell Profiler. Pixel size 2–15 was used to identify ATG9 vesicles, followed by normalization to cell number. Plots were created and statistical analyses were carried out using Graphpad Prism.

## Proteomics workflow

Protocols for proteomics as used here are available at https://doi.org/10.17504/protocols.io.yxmvm32nbl3p/v1 and https://doi.org/10.17504/protocols.io.dm6gp3jb1vzp/v1.

**Total proteome sample preparation for TMT.** Cells were cultured to 70% confluency and washed with PBS three times. Cells were lysed in urea denaturing buffer (8 M urea, 150 mM NaCl, 50 mM EPPS pH 8.0, containing mammalian protease inhibitor cocktail (Sigma) and

PhosSTOP) Cell lysates were collected by cell scrapers and sonicated on ice for 10 s at level 5, and the resultant extracts were clarified by centrifugation for 10 min at 15,000g at 4 °C. Lysates were quantified by the BCA assay and about 50 μg of protein was reduced with TCEP (10 mM final concentration for 30 min) and alkylated with chloroacetamide (20 mM final concentration) for 30 min. Proteins were chloroform–methanol precipitated using the SL-TMT protocol[34], reconstituted in 200 mM EPPS (pH 8.5), digested by LysC for 2 h at 37 °C (1:200 wt/wt LysC/protein) and then treated with trypsin overnight at 37 °C (1:100 wt/wt trypsin/protein). About 25 μg of protein was labelled with 62.5 μg of TMT or TMTpro for 120 min at room temperature. After a labelling efficiency check, samples were quenched with hydroxylamine solution at about 0.3% final (wt in water), pooled and desalted by C18 solid-phase extraction (Sep-Pak, Waters). Pooled samples were offline fractionated with basic reverse-phase liquid chromatography (LC) into a 96-well plate and combined for a total of 24 fractions[35] before desalting using a C18 StageTip (packed with Empore C18; 3M Corporation), and subsequent LC–MS/MS analysis.

**Total proteome sample preparation for data-independent acquisition.** HEK293 cells (with or without amino acid withdrawal treatment) were cultured to about 70% confluency, washed twice with chilled PBS, and collected by cell scraping in PBS. Following centrifugation at 4 °C, cell pellets were lysed in a denaturation buffer (8 M urea, 150 mM NaCl, 50 mM EPPS pH 8.0, containing mammalian protease inhibitor cocktail (Sigma), and PhosSTOP) by sonication (three times at level 5 for 5 s, with a 30 s rest on ice). Cell extracts were clarified by centrifugation for 10 min at 15,000g at 4 °C. Lysates were quantified by BCA and protein was reduced with TCEP (5 mM final concentration for 30 min), alkylated with IAA (10 mM final concentration) in the dark for 30 min, and quenched with DTT (5 mM final concentration) for 30 min. A 100 μg quantity of protein was methanol–chloroform precipitated using the SL-TMT protocol[36], reconstituted in 100 mM EPPS (pH 8.5 at 1 mg ml⁻¹), digested by LysC for 2 h at 37 °C (1:100 wt/wt LysC/protein) and then by trypsin overnight at 37 °C (1:100 wt/wt trypsin/protein). A 30 μg quantity of protein digests was acidified with formic acid to pH ≈ 3–3.5, desalted using a C18 StageTip (packed 200-μl pipette tip with Empore C18; 3M Corporation), and subjected to data-independent acquisition (DIA) LC–MS/MS analysis.

**Sample preparation for MS–APEX2 proteomics.** For APEX2 proteomics, cells expressing various APEX2–Flag fusions were processed as described previously[20]. To induce proximity labelling in live cells, cells were incubated with 500 μM biotin phenol (LS-3500.0250, Iris Biotech) for 1 h and treated with 1 mM H₂O₂ for 1 min, and the reaction was quenched with three washes of 1× PBS supplemented with 5 mM Trolox, 10 mM sodium ascorbate and 10 mM sodium azide. Cells were then collected and lysed in radioimmunoprecipitation assay (RIPA) buffer. To enrich biotinylated proteins, about 2 mg of cleared lysates was subjected to affinity purification by incubation with streptavidin-coated agarose beads (catalogue no. 88817, Pierce) for 1.5 h at room temperature. Beads were subsequently washed twice with RIPA buffer, once with 1 M KCl, once with 0.1 M NaCO₃, once with PBS and once with water. For proteomics, biotinylated protein bound to the beads was reduced using TCEP (10 mM final concentration) in EPPS buffer at room temperature for 30 min. After reduction, samples were alkylated with the addition of chloroacetamide (20 mM final concentration) for 20 min. Beads were washed three times with water. Proteins bound to beads were then digested with LysC (0.5 μl) in 100 ml of 0.1 M EPPS (pH 8.5) for 2 h at 37 °C, followed by trypsin overnight at 37 °C (1 μl). To quantify the relative abundance of individual protein across different samples, each digest was labelled with 62.5 mg TMT11 or TMT16pro reagents for 2 h at room temperature (Thermo Fisher), mixed, and desalted with a C18 StageTip (packed with Empore C18; 3M Corporation) before SPS-MS³ analysis on an Orbitrap Fusion Lumos Tribrid Mass Spectrometer (Thermo

Fisher) coupled to a Proxeon EASY-nLC 1200 LC pump (Thermo Fisher). Peptides were separated on a 100-μm-inner-diameter microcapillary column packed with about 35 cm of Accucore150 resin (2.6 μm, 150 Å, Thermo Fisher) with a gradient consisting of 5%–21% (ACN, 0.1% FA) over a total 150-min run at about 500 nl min⁻¹ (ref. 37). The instrument parameters for each experiment are provided below.

**TMT data acquisition.** Samples were analysed on an Orbitrap Fusion Lumos Tribrid Mass Spectrometer coupled to a Proxeon EASY-nLC 1200 pump (Thermo Fisher). Peptides were separated on a 35-cm column packed using a 95- to 110-min gradient. MS1 data were collected using the Orbitrap (120,000 resolution). MS2 scans were carried out in the ion trap with CID fragmentation (isolation window 0.7 Da; rapid scan; NCE 35%). Each analysis used the Multi-Notch MS³-based TMT method[38], to reduce ion interference compared to MS2 quantification, combined in some instances with newly implemented Real Time Search analysis[39,40], and with the FAIMS Pro Interface (using previously optimized 3 CV parameters (−40, −60, −80) for TMT multiplexed samples[41]). MS³ scans were collected in the Orbitrap using a resolution of 50,000, and NCE of 65 (TMT) or 45 (TMTpro). The closeout was set at two peptides per protein per fraction, so that MS³ scans were no longer collected for proteins having two peptide–spectrum matches that passed quality filters.

**DIA.** Samples were analysed on an Orbitrap Exploris 480 Mass Spectrometer coupled to a Proxeon EASY-nLC pump 1000 (Thermo Fisher). Peptides were separated on a 15-cm column packed with Accucore150 resin (150 Å, 2.6-mm C18 beads Thermo Fisher) using an 80-min acetonitrile gradient. MS1 data were collected using the Orbitrap (60,000 resolution, 350–1,050 m/z, 100% normalized AGC, maxIT set to auto). DIA MS2 scans in the Orbitrap were carried out with overlapping 24-m/z windows for the first duty cycle (390–1,014 m/z) and for the second duty cycle (402–1,026 m/z) with 28% NCE, 30,000 resolution, for fixed 145–1,450 m/z range, 1,000% normalized AGC, and a 54-ms maxIT MS1 survey scan was carried out following each DIA MS/MS duty cycle.

**TMT data analysis.** Mass spectra were converted to mzXML and monoisotopic peaks were reassigned with Monocole[42] and then database searched using a Comet-based method[43,44] or Sequest-HT using Proteome Discoverer (v2.3.0.420 – Thermo Fisher). Database searching included all canonical entries from the Human reference proteome database (UniProt Swiss-Prot – 2019-01; https://ftp.uniprot.org/pub/databases/uniprot/previous_major_releases/release-2019_01/) and sequences of common contaminant proteins. Searches were carried out using a 20-ppm precursor ion tolerance, and a 0.6 Da product ion tolerance for ion trap MS/MS was used. TMT tags on lysine residues and peptide N termini (+229.163 Da for Amino-TMT or +304.207 Da for TMTpro) and carbamidomethylation of cysteine residues (+57.021 Da) were set as static modifications, and oxidation of methionine residues (+15.995 Da) was set as a variable modification. Peptide–spectrum matches were filtered to a 2% false discovery rate (FDR) using linear discriminant analysis as described previously[43] using the Picked FDR method[45], and proteins were filtered to the target 2% FDR level. For reporter ion quantification, a 0.003-Da window around the theoretical m/z of each reporter ion was scanned, and the most intense m/z was used. Peptides were filtered to include only those peptides with >200 summed signal-to-noise ratio across all TMT channels. An isolation purity of at least 0.5 (50%) in the MS1 isolation window was used for samples analysed without online real-time searching. For each protein, the filtered peptide–spectrum match TMT or TMTpro raw intensities were summed and log₂ normalized to create protein quantification values (weighted average). For protein TMT quantifications, TMT channels were normalized to the summed (protein abundance experiments)[46] or median (proximity labelling experiments)[47] TMT intensities for each TMT channel.

**DIA data analysis.** Mass spectra were converted to mzML using msconvert[48] with demultiplexing (overlap only at 10-ppm mass error). mzML files were processed with DIA-NN[49] using UniProt entries (UP000005640 [9606]). For DIA-NN, the following parameters were used: trypsin specificity ([RK]/P), N-term methionine excision enabled, fixed modification of carbamidomethylation on cysteines, in library-free mode, deep learning-based spectra and RTs enabled, MBR enabled, precursor FDR 1% filter, and quantification with Robust LC (high precision). Using the report.pg_matrix.tsv output from DIA-NN, we calculated the mean intensity across replicates for untreated and amino acid withdrawal treatment conditions ($n$ = 4 each) based on replicate intensities (observed in at least two biological replicates), which were used to estimate a protein copy number per cell using the proteome ruler method[15].

## Statistical analysis

Normalized $\log_2$ protein reporter ion intensities were compared using a Student's $t$-test and resultant $P$ values were corrected using the Benjamini–Hochberg adjustment. Volcano plots and other data visualizations were generated in R using resulting $q$ values and mean fold changes. Annotations for subcellular lists were derived from ref. 14 and designations were derived from ref. 32. Additional cytosol protein and Golgi transmembrane number annotations were derived from Uniprot. Gene Ontology annotations from Uniprot were appended to MS data to carry out Fisher's exact tests to identify Gene Ontology enrichment terms (corrected by Benjamini–Hochberg adjustment). Proteome ruler values were estimated using previously described methods[15,50]. The proportional contribution of the untreated WT TMT channels to the MS1 precursor area ($\text{TMT}^{\text{WT/UT}}/\text{TMT}^{\text{All}} \times \text{MS1}^{\text{Area}}$) was summed to the protein level for its constituent peptides. Resultant protein values were then used to calculate a TMT-based proteome ruler protein absolute abundance estimate. For imaging quantification, a Mann–Whitney $P$ value was calculated using GraphPad Prism9. $P$ values <0.05 were considered significant unless otherwise noted. Compartment protein copy number rank tests were carried out using a Wilcoxon test to calculate $P$ values. All data figures were generated in Adobe Illustrator, using R (4.1.3), Rstudio IDE(2021.09.3 Build 396, Posit) and GraphPad Prism9.

**RMSE calculation.** To generate our CAP list, we used known autophagy fluxers in autophagy-proficient (WT) or autophagy-deficient ($ATG7^{-/-}$ or $FIP200^{-/-}$) cells. For each known autophagy fluxer, the condition median $z$ score was used. From these protein condition medians, we took the median value across the known subset of proteins to estimate a condition median to build a consensus profile, which is analogous to 'protein correlation profiling'[51]. Using the consensus profile median values for known autophagy proteins as predicted, we then calculated the RMSE for each protein in the datasets.

$$\text{RMSE} = \sqrt{\sum_{i=1}^{\text{TMT}^n} \frac{(\text{Predicted}_i - \text{observed}_i)^2}{n^{\text{TMT channels}}}}$$

By calculating the RMSE for every quantified protein, we generated a group of CAPs in two distinct starvation conditions based on the top 10% of proteins with the lowest RMSE across the datasets respectively. The 10% cutoff aligns well with the rightmost tail of the density plot for the known autophagy fluxers and the top 30 autophagy factors from Fig. 2. Although the resulting 'autophagy' candidate list provides a defined collection of autophagy substrates, the RMSE calculation averages the error across a protein's abundance profile, potentially enabling some proteins that vary from the consensus profile in a single condition to make the candidate list. Also, some autophagy substrates with high replicate variance in abundance may not make the cutoff required despite largely following the known autophagy fluxer consensus profile.

**Prioritization of 'autophagy' cargo.** To prioritize the top candidate autophagy cargo, we ranked proteins on the basis of their starvation and autophagy turnover (Fig. 1) and proximity to ATG8 machinery (Fig. 2). To calculate a rank for starvation- and autophagy-dependent turnover, we determined the priority value on the basis of the lesser of either the absolute value of the WT $\log_2$ fold change in protein abundance from EBSS/untreated for $\log_2[\text{FC(EBSS/UT)}] \leq 0$ or the $ATG7^{-/-}$ $\log_2[\text{FC(EBSS/UT)}] - \text{WT} \log_2[\text{FC(EBSS/UT)}]$ for changes ≥0 (when both criteria are met). Proteins that did not meet both criteria were assigned a 0 priority. The priority values were then arranged in descending order and proteins were scaled ranked (protein rank/number of total proteins in the experiment). Scaled ranks were calculated for HeLa and HEK293 data separately and the minimum scaled rank found in at least one of the datasets was used. Proteins were reordered on the basis of priority and scaled ranked combining the two datasets to summarize the findings of Fig. 1. For ATG8 proximity ranks, we determined a priority value on the basis of the lesser of either the $\log_2$ fold change in protein abundance from WT EBSS + BafA1/Untreated for $\log_2[\text{FC(EBSS + BafA1/UT)}] \geq 0$ or the absolute value of the ATG8 LDS mutant $\log_2[\text{FC(EBSS + BafA1/UT)}] - \text{WT} \log_2\text{FC}[(\text{EBSS + BafA1/UT})]$ for changes ≤0 (only when both criteria are met). As above, proteins that did not meet both criteria were assigned a 0 priority. Using the priority values, scaled ranks were calculated for the APEX2–GABRAPL2 and APEX2–MAP1LC3B experiments separately, for which the minimum scaled rank found in at least one of the experiments was used. Proteins were reordered on the basis of priority and scaled ranked combining the two datasets to summarize the findings of Fig. 2. To prioritize candidates that exhibited both an autophagy- and starvation-dependent turnover and increased association with ATG8 during starvation, we summed the scaled ranks of Fig. 1 and Fig. 2 to generate a summed rank value that we sorted by ascending order to generate our final ranked list of candidates. To be a candidate in the final ranked list, the protein must have been identified in at least one experiment from the Fig. 1 experiments (HeLa or HEK293) and one experiment from the Fig. 2 experiments (APEX2–GABARAPL2 and APEX2–MAP1LC3B). LIR motifs were matched from the iLIR Autophagy Database (http://repeat.biol.ucy.ac.cy/iLIR/)[21]. Known autophagy proteins were derived from ref. 10. Although the RMSE approach may not capture every autophagy substrate, the prioritized collection of CAPs nevertheless allowed us to define the selectivity of macroautophagy during nutrient stress.

## Reporting summary

Further information on research design is available in the Nature Portfolio Reporting Summary linked to this article.

## Data availability

All MS data for HeLa and HEK293 cells (155 files) have been deposited to the ProteomeXchange Consortium through the PRIDE repository (http://www.proteomexchange.org/; project accession: PXD038358). Proteomic data for ES cells and iNeurons (15 files) are available with project accession PXD043923. All analysed proteomic data are available in Supplementary Tables 1, 2 and 4–9. Uncropped blots are provided in Supplementary Fig. 1. We used canonical protein entries from the Human reference proteome database in our study (UniProt Swiss-Prot – 2019-01; https://ftp.uniprot.org/pub/databases/uniprot/previous_major_releases/release-2019_01/). LIR motifs were based on the iLIR Autophagy Database (https://ilir.warwick.ac.uk/). Source data are provided with this paper.

## Code availability

Code and data analysis to generate paper figures is available at https://zenodo.org/record/8380684 and https://github.com/harperlaboratory/Golgiphagy.git. All data and data figures can be explored using

CARGO. CARGO is a ShinyApp interface generated in R and RStudio that can be accessed at https://harperlab.connect.hms.harvard.edu/CARGO_Cellular_Autophagy_Regulation_GOlgiphagy/ (RRID:SCR_024474).

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

**Acknowledgements** We thank members of the laboratory of J.W.H. for feedback. This work was financially supported by the Aligning Science Across Parkinson's (ASAP) initiative (J.W.H.), NIH RO1 NS110395 (J.W.H.), NIH RO1 AG011085 (J.W.H.), NIH RO1 GM132129 (J.A.P.) and a Merck-Helen Hay Whitney Foundation Fellowship (K.L.H.). The Michael J. Fox Foundation administers the grant ASAP-000282 on behalf of ASAP and itself. For the purpose of open access, the author has applied for a CC-BY public copyright licence to the Author's Accepted Manuscript version arising from this submission. We acknowledge the Nikon Imaging Center (Harvard Medical School) for imaging assistance, and R. Youle for providing engineered HeLa cells.

**Author contributions** This study was conceived by S.S., K.L.H. and J.W.H. The experiments and analysis were carried out by K.L.H., S.S., I.R.S., J.C.P., E.M.W. and J.A.P. Gene editing and cell line generation were carried out by S.S., K.L.H. and J.C.P. Proteomics was carried out by S.S., K.L.H., J.A.P., J.C.P. and I.R.S. Analysis of proteomic data was carried out by I.R.S. and K.L.H. I.R.S. created the CARGO website. Microscopy was carried out by K.L.H. and E.M.W. DNA constructs were prepared by K.L.H. and S.S. Cell biology, biochemistry and flow cytometry were carried out by K.L.H., E.M.W., J.C.P. and S.S. K.L.H. and J.W.H. wrote the manuscript with contributions from all authors.

**Competing interests** J.W.H. is a founder and consultant for Caraway Therapeutics and a co-founding board member of Interline Therapeutics. All other authors declare no competing interests.

**Additional information**
**Correspondence and requests for materials** should be addressed to J. Wade Harper.

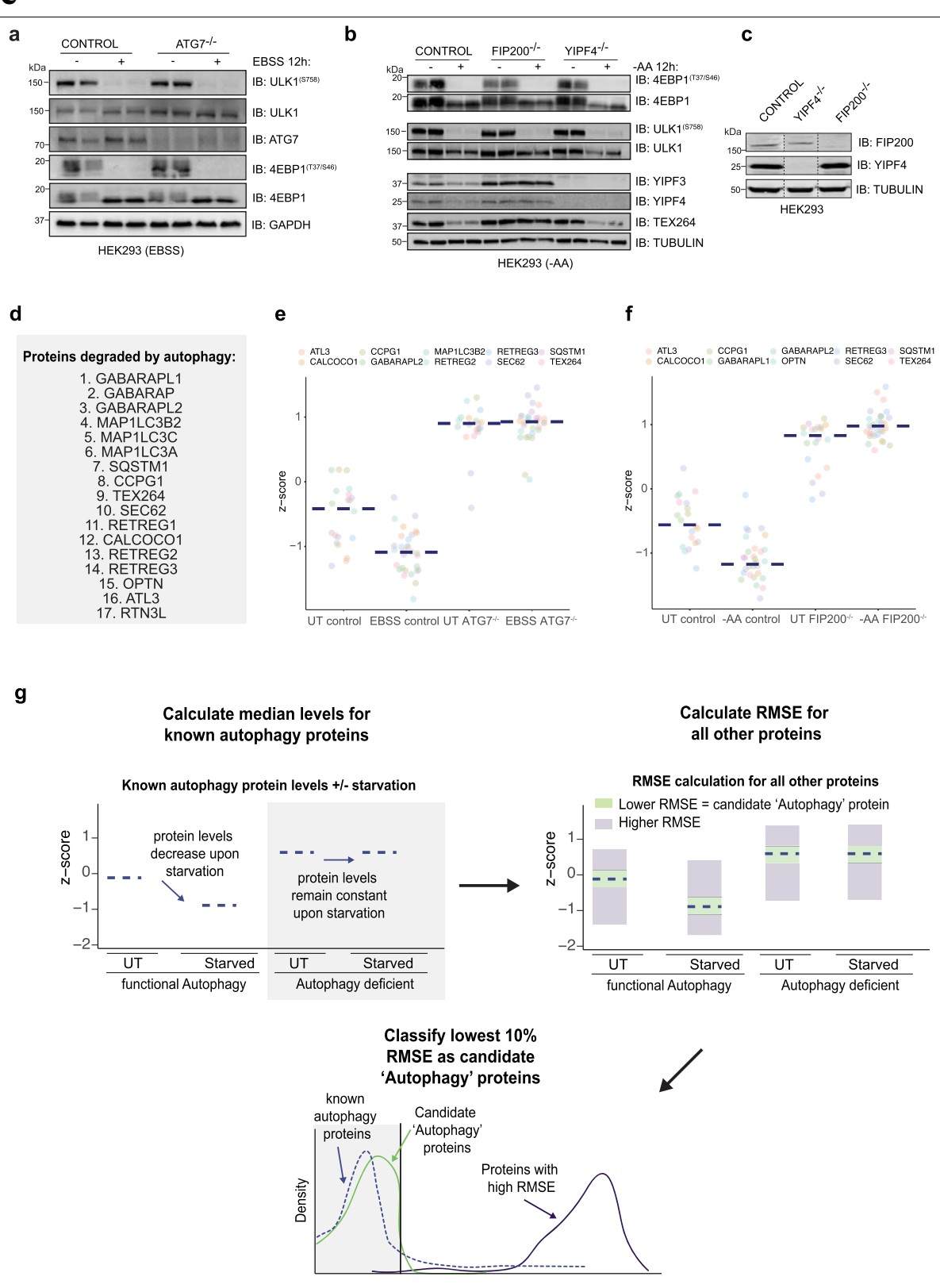

**Extended Data Fig. 1** | See next page for caption.

**Extended Data Fig. 1 | Experimental approach for identification of candidate autophagy proteins via quantitative proteomics. a**, Immunoblot for HEK293 control and ATG7$^{-/-}$ cells with or without EBSS starvation for 12h in duplicate with the indicated antibodies. Independently cultured replicate samples were loaded in adjacent lanes. **b**, Immunoblot for HEK293 control, FIP200$^{-/-}$ and YIPF4$^{-/-}$ cells with or without AA withdrawal for 12h in duplicate with the indicated antibodies. Independently cultured replicate samples were loaded in adjacent lanes. **c**, Immunoblot for HEK293 control, FIP200$^{-/-}$ and YIPF4$^{-/-}$ cells probed with the indicated antibodies. Dotted lines indicate separate lanes on samples analyzed on the same gel. This experiment was performed in biological triplicate with similar results. **d**, List of proteins that demonstrate autophagy dependent degradation during nutrient starvation. **e**, Dot plot of proteins from panel **d** in WT or ATG7$^{-/-}$ HEK293 cells treated with EBSS for 12h. Navy dashed line represents median protein abundance. **f**, Dot plot of proteins from panel **d** in WT or ATG7$^{-/-}$ HEK293 cells treated with AA withdrawal for 12h. Navy dashed line represents median protein abundance. **g**, Workflow for calculating RMSE of all proteins in HEK293 Control or ATG7$^{-/-}$ cells treated with EBSS for 12h, and HEK293 control, FIP200$^{-/-}$, and YIPF4$^{-/-}$ cells treated with AA withdrawal for 12h.

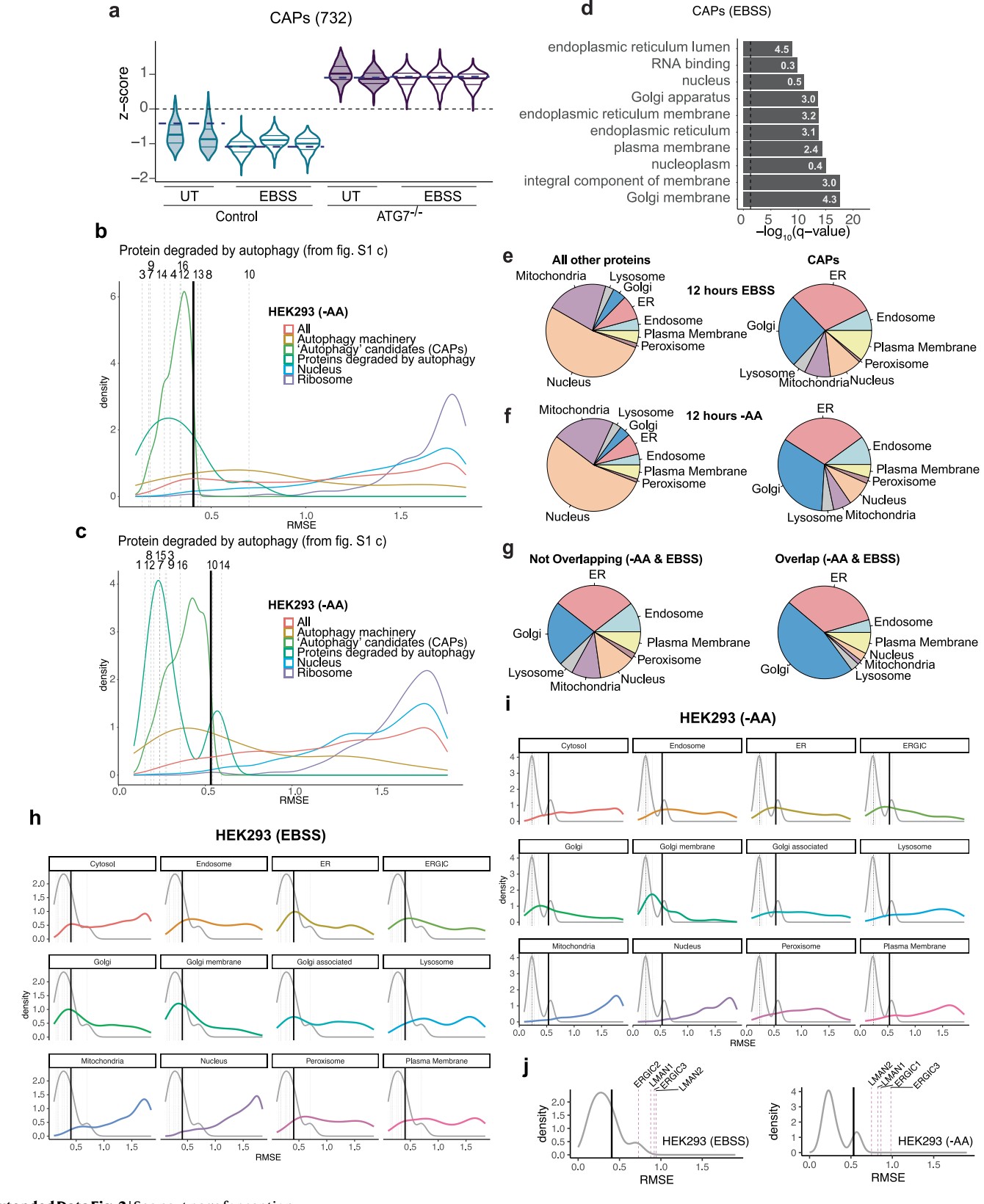

**Extended Data Fig. 2** | See next page for caption.

**Extended Data Fig. 2 | Golgi and ER are enriched in candidate autophagy proteins. a**, Violin plots for proteins identified as candidate 'autophagy' proteins in WT and ATG7$^{-/-}$ HEK293 cells with or without EBSS (12h). Navy dashed lines: median value for known autophagy proteins. **b**, RMSE plot for HEK293 cells treated with 12h of EBSS. CAPs are shown along with all autophagy machinery, nuclear, and ribosomal proteins. **c**, RMSE plot for HEK293 cells treated with 12h of AA Withdrawal. CAPs are shown along with all autophagy machinery, nuclear, and ribosomal proteins. **d**, Top 10 Gene Ontology terms identified for candidate 'autophagy' proteins from cells subjected to EBSS. **e**, Frequency of proteins with the indicated sub-cellular localizations for the candidate 'autophagy' proteins or all other proteins for cells subjected to EBSS treatment. **f**, Frequency of proteins with the indicated sub-cellular localizations for the candidate 'autophagy' proteins or all other proteins for cells subjected to AA withdrawal. **g**, Frequency of proteins with the indicated sub-cellular localizations for either overlapping or non-overlapping proteins. **h**, RMSE for each compartment shown from HEK293 EBSS experiment. **i**, RMSE for each compartment shown from HEK293 AA withdrawal experiment. **j**, RMSE plots for EBSS and −AA with known ERGIC proteins indicated in magenta.

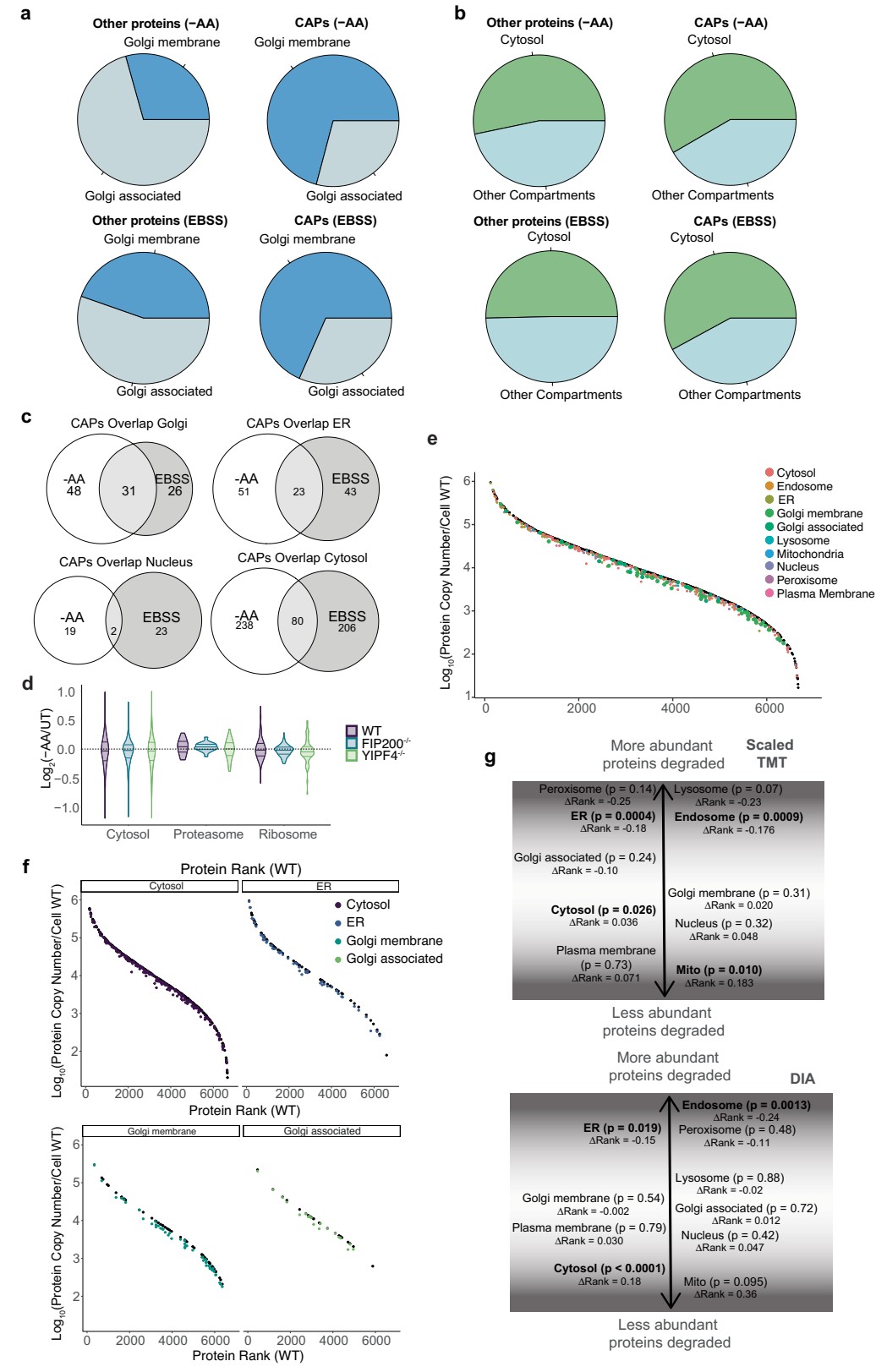

**Extended Data Fig. 3** | See next page for caption.

**Extended Data Fig. 3 | Subcellular localization analysis of candidate autophagy proteins. a**, Enrichment of Golgi-membrane and Golgi-associated proteins in the candidate 'autophagy' list and all other proteins for AA withdrawal and EBSS treatment. **b**, Enrichment for the cytosolic proteins in the candidate 'autophagy' list and all other proteins for AA withdrawal and EBSS treatment. **c**, Venn diagrams indicating the overlap of proteins identified in common within candidate 'autophagy' lists for AA withdrawal and EBSS treatment. Numbers within the diagram indicate the number of proteins present. **d**, Violin plots for Log$_2$FC (−AA/UT) for control, FIP200$^{-/-}$, or YIPF4$^{-/-}$ HeLa cells displayed for 38 proteasome and 84 ribosomal proteins as well as proteins annotated as cytosolic. Median values are indicated by solid bold line. **e**, TMT-scaled MS1 ranked plots. Protein copy number estimates for CAPs in HEK293 cells (black) in rank order. Among CAPs, the number of protein copies after loss by autophagy during amino acid starvation for each compartment as determined using protein abundance fold changes (AA withdrawal – untreated). **f**, Rank plot for cytoplasmic, ER and Golgi localized proteins. **g**, Model for possible selectivity of macroautophagy at the organelle level. Abundance rank change (ΔRank) between proteins in the 'autophagy' candidate list – all other proteins for each organelle, scaled to number of total proteins in both scaled TMT and DIA experiments. For each compartment, p-values are listed and organelles with significant differences are in bold. Interestingly, cytosolic CAPs display a bias toward less abundant proteins, while CAPs annotated as ER or endosomal are biased for more abundant proteins. In contrast, CAPs annotated as Golgi proteins do not present a significant bias toward more or less abundant proteins. p-values from two-sided Wilcoxon Rank Sum test.

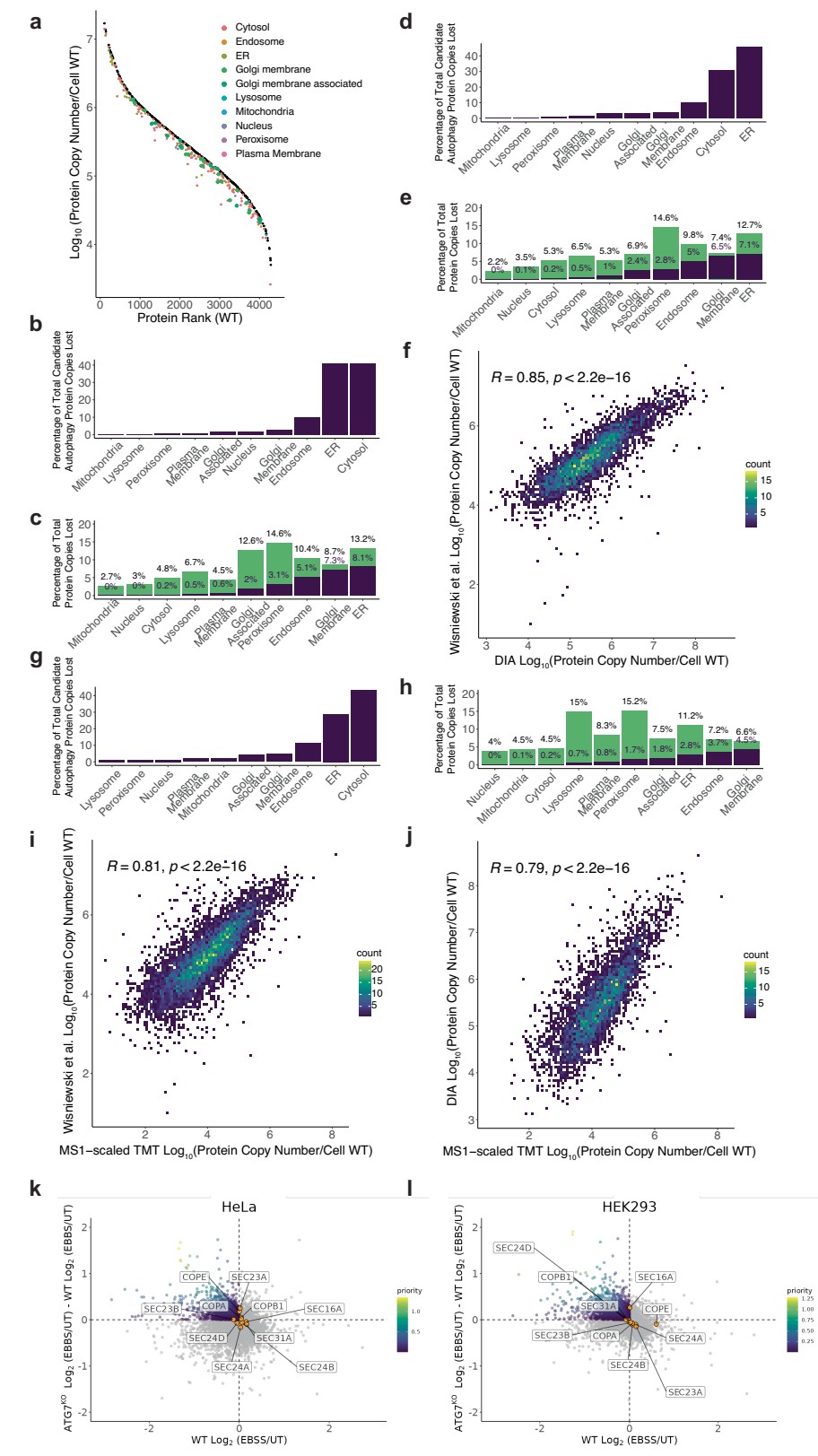

**Extended Data Fig. 4** | See next page for caption.

**Extended Data Fig. 4 | Proteome census analysis during nutrient stress.**
**a**, DIA ranked plots. Protein copy number in the untreated condition for candidate 'autophagy' proteins in HEK293 cells (black) in rank order. The number of protein copies after loss by autophagy during amino acid starvation for each compartment as determined using protein abundance fold changes (AA withdrawal – untreated) by DIA. **b**, Among the candidate autophagy proteins, percentage of total protein copy numbers lost via amino acid withdrawal ($3.0161 \times 10^7$ total). **c**, Percentage of all protein copies lost from 'autophagy' candidate list (purple) or other mechanisms (green) by amino acid withdrawal for subcellular compartments based on DIA values with histone-based proteome ruler values. **d,e**, Same as panels **b** and **c**, respectively, but based on DIA FC values mapped onto proteome ruler values from Wisniewski et al.[15] ($9.77 \times 10^6$ total). **f**, Correlation with DIA protein copy number estimates against Wisniewski et al.[15] protein copy numbers. $R = 0.85$, $p < 2.2 \times 10^{-16}$. Statistics are Pearson correlation. **g,h**, Same as panels **b** and **c**, respectively, based on TMT-scaled FC values mapped onto proteome ruler values from Wisniewski et al.[15] ($7.1573 \times 10^6$ total). **i**, Correlation plots for TMT-scaled MS1 protein signals against Wisniewski et al.[15] copy number. $R = 0.81$, $p < 2.2 \times 10^{-16}$. Statistics are Pearson correlation. **j**, Correlation plots for TMT-scaled MS1 protein copy numbers and DIA protein copy numbers. $R = 0.79$, $p < 2.2 \times 10^{-16}$. Statistics are Pearson correlation. **k**, Distribution of proteins with roles in coatamer (COPI/II) function in response to EBSS treatment in WT and ATG7$^{-/-}$ HeLa cells. These proteins do not display properties of CAPs. **l**, Distribution of proteins with roles in coatamer (COPI/II) function in response to EBSS treatment in WT and ATG7$^{-/-}$ HEK293 cells. These proteins do not display properties of CAPs.

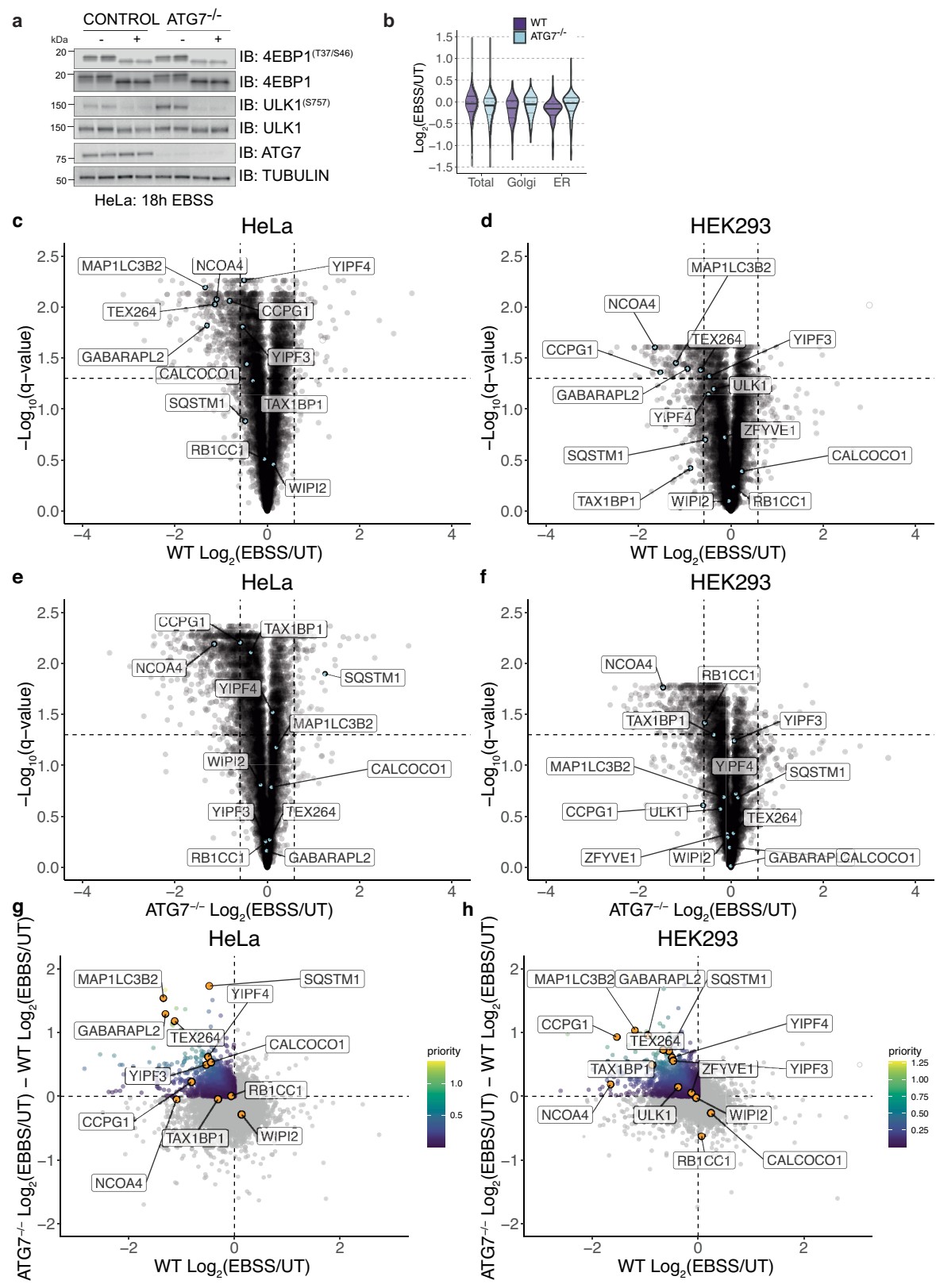

**Extended Data Fig. 5 | Identification of candidate autophagy proteins using total proteome analysis in ATG7$^{-/-}$ HEK293 or HeLa cells. a**, Western blot showing markers of starvation (ULK1, 4EBP dephosphorylation) and ATG7 in WT and ATG7$^{-/-}$ HEK293 cells grown in EBSS for 12h. Independently cultured replicate samples were loaded in adjacent lanes. **b**, Violin plots of relative total (8258), Golgi (160), or ER (344) protein abundance in response to EBSS treatment (12h) in WT and ATG7$^{-/-}$ HeLa cells. **c-f**, Volcano plots [WT Log$_2$

(12h EBSS/UT) versus -Log$_{10}$(q-value)] for HeLa (panel **c**) or HEK293 (panel **d**) or analogous plots for ATG7$^{-/-}$ HeLa (panel **e**) or HEK293 (panel **f**) cells. **g, h**, Plots of ATG7$^{-/-}$ Log$_2$(EBSS/UT) - WT Log$_2$(EBSS/UT) versus WT Log$_2$(EBSS/UT) for HeLa cells (panel **g**) and ATG7$^{-/-}$ Log$_2$(EBSS/UT) – WT Log$_2$(EBSS/UT) versus WT Log$_2$(EBSS/UT) for HEK293 cells (panel **h**) where priority for individual proteins is scaled based on the color code inset.

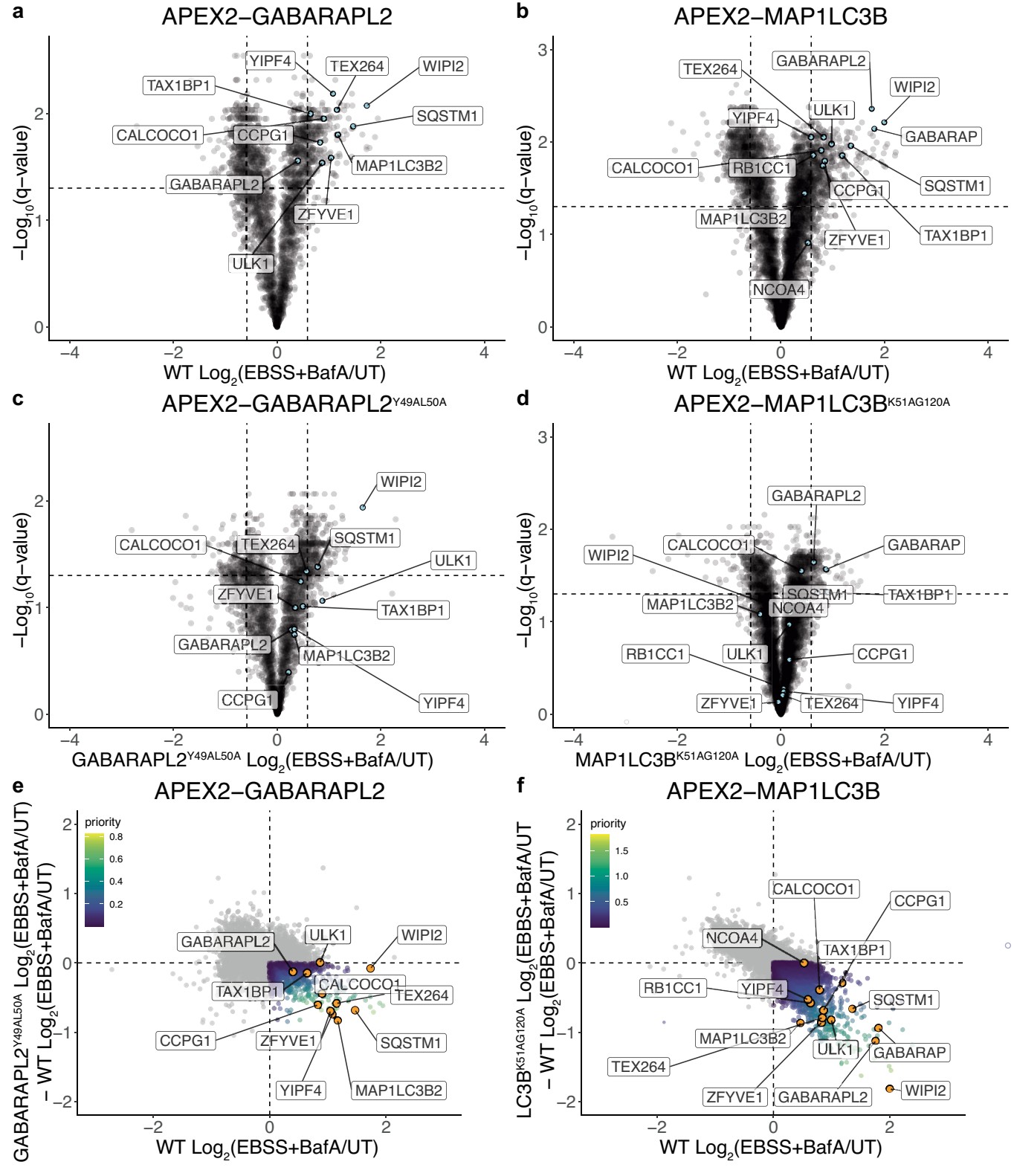

**Extended Data Fig. 6 | Proximity biotinylation of ATG8 proteins MAP1LC3B or GABARAPL2 in response to nutrient stress. a-d,** Volcano plots [WT Log$_2$(4h EBSS+BafA1/UT) versus −Log$_{10}$ (q-value)] for APEX2-GABARAPL2 (panel **a**) or APEX2-MAP1LC3B (panel **b**) or analogous plots for APEX2-GABARAPL2$^{Y49A/L50A}$ (panel **c**) or APEX2-MAP1LC3B$^{K51A/G120A}$ (panel **d**) in HeLa cells. **e,f,** Plots of GABARAPL2$^{Y49A/L50A}$ Log$_2$(EBSS+BafA1/UT) − WT Log$_2$(EBSS+BafA1/UT) versus WT Log$_2$(EBSS+BafA1/UT) (panel **e**) and MAP1LC3B$^{K51A/G120A}$ Log$_2$(EBSS+BafA1/UT) − WT Log$_2$(EBSS+BafA1/UT) versus WT Log$_2$(EBSS+BafA1/UT) (panel **f**) where priority for individual proteins is scaled based on the color code inset.

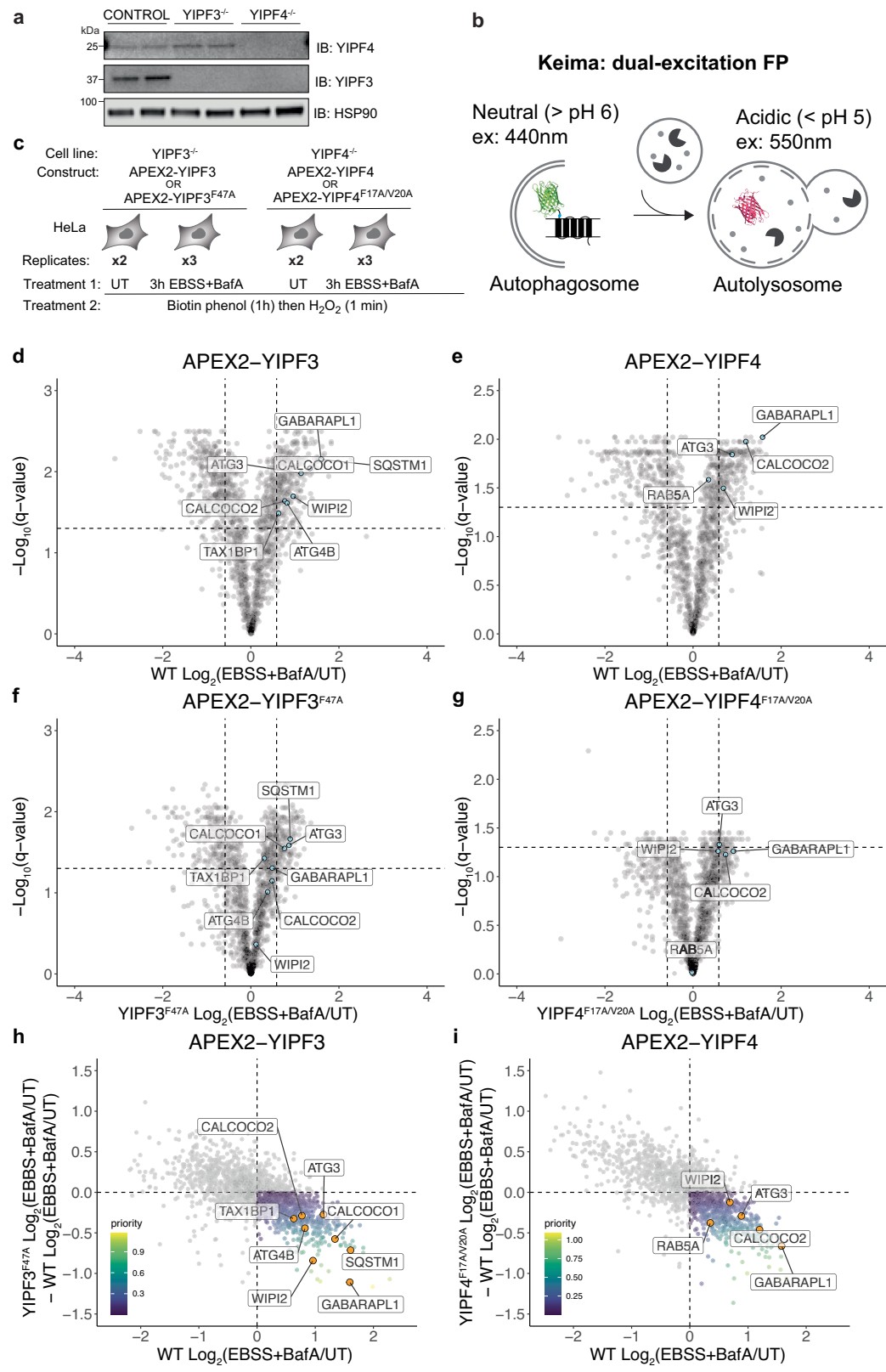

**Extended Data Fig. 7 | Proximity biotinylation of YIPF3 and YIPF4.**
**a**, Immunoblotting of WT, YIPF3$^{-/-}$, and YIPF4$^{-/-}$ HeLa cells probed in duplicate with the indicated antibodies. Anti-HSP90 was used as a loading control. Independently cultured replicate samples were loaded in adjacent lanes. **b**, Scheme outlining Keima-YIPF3/4 as reporters for Golgiphagic flux. **c**, Experimental scheme for proximity biotinylation using APEX2-YIPF3/YIPF4 (or LIR mutants) in response to nutrient stress (EBSS, 4h). **d-i**, Volcano plots

[WT Log$_2$(4h EBSS+BafA1/UT) versus −Log$_{10}$ (q-value)] for APEX2-YIPF3 (panel **d**) or APEX2-YIPF4 (panel **e**) or analogous plots for APEX2-YIPF3$^{F47A}$ (panel **f**) or APEX2-YIPF4$^{F17A/V20A}$ (panel **g**) in HeLa cells. **h,i**, Plots of APEX2-YIPF3$^{F47A}$ Log$_2$(EBSS+BafA1/UT) WT Log$_2$(EBSS+BafA1/UT) versus WT Log$_2$(EBSS+BafA1/UT) (panel **h**) and APEX2-YIPF4$^{F17A/V20A}$ Log$_2$(EBSS+BafA1/UT) WT Log$_2$(EBSS+BafA1/UT) versus WT Log$_2$(EBSS+BafA1/UT) (panel **i**) where priority for individual proteins is scaled based on the color code inset.

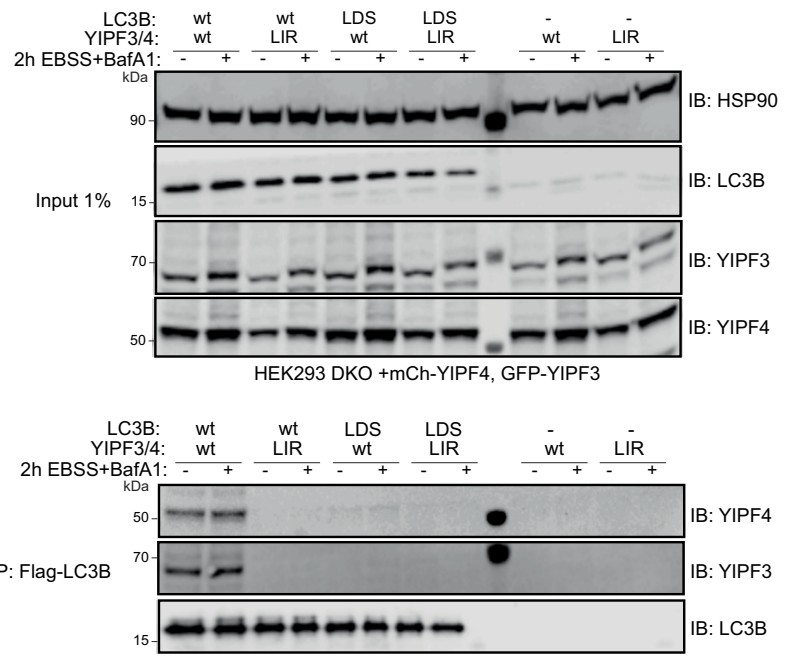

**a**

**HEK293 DKO (YIPF3⁻/⁻/YIPF4⁻/⁻)**

WT YIPF3/YIPF4 fusion proteins

GFP | YIPF3
mCherry | YIPF4

GOLGA2
YIPF3
YIPF4
Hoechst33342

Golgi (GOLGA2)

YIPF4

YIPF3

LIR_mut YIPF3/YIPF4 fusion proteins

F47A
GFP | YIPF3
F17A/V20A
mCherry | YIPF4

GOLGA2
YIPF3
YIPF4
Hoechst33342

Golgi (GOLGA2)

YIPF4

YIPF3

**c**

|  | Control | HKO | ΔLC3 | ΔRAP | Cell line |
|---|---|---|---|---|---|
|  | - + | - + | - + | - + | 12h EBSS |

kDa
38 — IB: YIPF3
25 — IB: YIPF4
38 — IB: TEX264
90 — IB: HSP90

HeLa

**b**

LC3B: wt / wt / LDS / LDS / - / -
YIPF3/4: wt / LIR / wt / LIR / wt / LIR
2h EBSS+BafA1: - + / - + / - + / - + / - + / - +

kDa
90 — IB: HSP90
Input 1%   15 — IB: LC3B
70 — IB: YIPF3
50 — IB: YIPF4

HEK293 DKO +mCh-YIPF4, GFP-YIPF3

LC3B: wt / wt / LDS / LDS / - / -
YIPF3/4: wt / LIR / wt / LIR / wt / LIR
2h EBSS+BafA1: - + / - + / - + / - + / - + / - +

kDa
50 — IB: YIPF4
IP: Flag-LC3B   70 — IB: YIPF3
15 — IB: LC3B

HEK293 DKO +mCh-YIPF4, GFP-YIPF3

**Extended Data Fig. 8 | Association of YIPF3/4 with ATG8 proteins.**
**a**, Immunofluorescence of WT and LIR mutant YIPF3 and YIPF4 exogenous expression constructs in DKO (YIPF3/4) HEK293 cells. Cells were imaged using confocal microscopy and co-stained with the Golgi marker GOLGA2 (yellow). Scale bars is 10 microns. **b**, FLAG-LC3B (WT or LDS mutant) was stably expressed in HEK293 cells expressing mCherry-YIPF3 and GFP-YIPF4 (WT or LIR motif mutants) via lentivirus and anti-FLAG immune complexes isolated with or

without a 2h EBSS+BafA1 treatment. Immune complexes were subjected to immunoblotting and probed with the indicated antibodies. This experiment was performed once. **c**, Immunoblotting and probing of indicated antibodies in ATG8 KO cell lines subjected to nutrient stress with EBSS. HKO (deletion of LC3A, B, C, GABARAP, L1, L2), ΔLC3 (deletion of LC3A, B, C), ΔRAP (deletion of GABARAP, L1, L2). Results are representative of experiments performed twice.

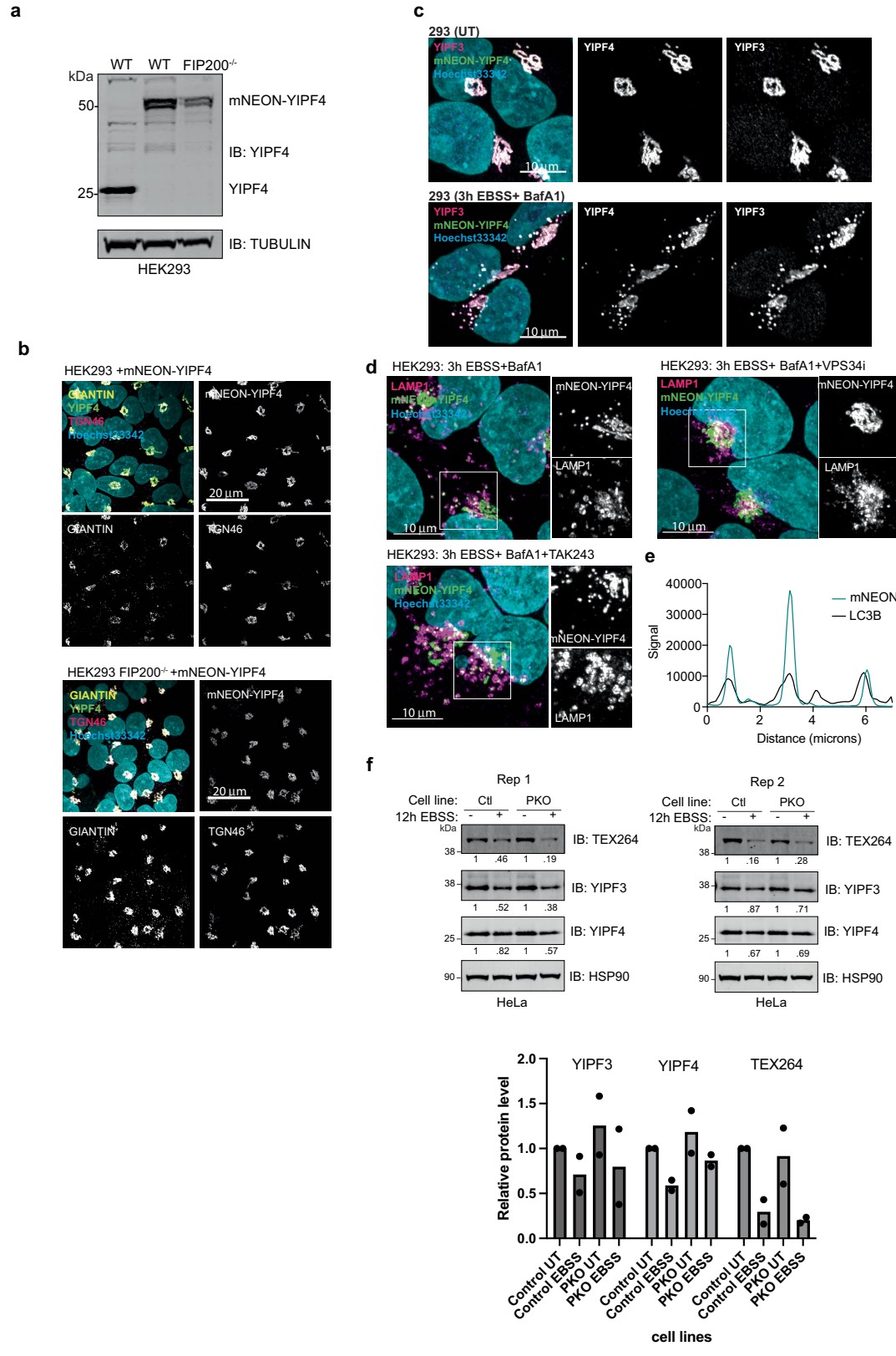

**Extended Data Fig. 9** | See next page for caption.

**Extended Data Fig. 9 | Analysis of YIPF4 localization in response to nutrient stress. a**, Immunoblot showing mNEON-YIPF4 endogenous tagging results at a higher molecular weight indicative of the total fusion protein length (~50 kDa). This experiment was performed in biological triplicate with similar results. **b**, HEK293 cells and HEK293 FIP200$^{-/-}$ cells expressing endogenous YIPF4 tagged on its N-terminus with mNEON (green) imaged using confocal microscopy and co-stained with cis (GOLGA2) or trans (TGN46) Golgi markers. Scale bars 20 microns. This experiment was performed in biological duplicate with similar results. **c**, HEK293 cells untreated or starved for 3h with EBSS in the presence of BafA1 (100 nM) expressing endogenous YIPF4 tagged on its N-terminus with mNEON (green) imaged using confocal microscopy and co-stained with an antibody against YIPF3. Scale bars 10 microns. This experiment was performed in biological triplicate with similar results. **d**, HEK293 cells expressing endogenous YIPF4 tagged on its N-terminus with mNEON (green) imaged using confocal microscopy and co-stained with LAMP1 (magenta). Cells were either left untreated (top left) or subjected to nutrient stress +BafA1 and VPS34i (3h) (top right) or subjected to nutrient stress +BafA1 and an E1 inhibitor (TAK243) (3h) (bottom) prior to imaging. Nuclei were labeled with Hoechst33342 dye (cyan). Scale bars 10 microns. This experiment was performed in biological triplicate with similar results. **e**, Line scan of HEK293 cells expressing endogenous YIPF4 tagged on its N-terminus with mNEON and MAP1LC3B show colocalization upon EBSS+BafA1 treatment for 3h. **f**, HeLa cells lacking OPTN, NDP52, SQSTM1, NBR1, and TAX1BP1 (Penta KO) were subjected to the indicated treatment for 12h and extracts probed with the indicated antibodies. Blots from two independent experiments were quantified based on LiCor intensities, then normalized to the UT sample for each genotype. The lower panel shows the individual values for each replicate, and error bars are S.D.

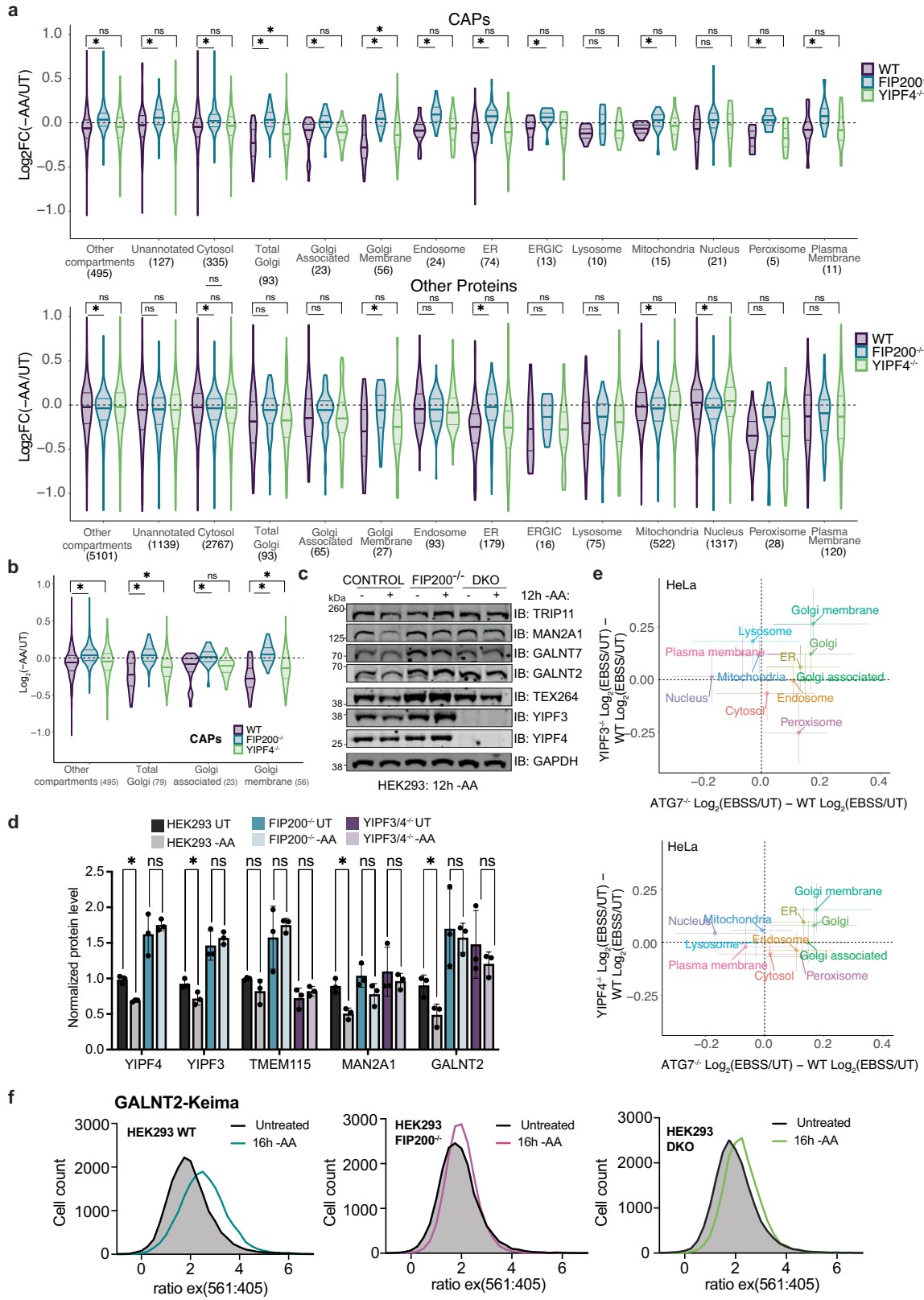

**Extended Data Fig. 10** | See next page for caption.

**Extended Data Fig. 10 | Contribution of YIPF3/4 to Golgi turnover by autophagy during nutrient stress. a**, Violin plots for $Log_2$(-AA/UT) for control, FIP200$^{-/-}$, or YIPF4$^{-/-}$ HeLa cells displayed for various classes of proteins with the indicated sub-cellular localizations for either the 'autophagy' candidates or all other proteins. Median values are indicated by solid bold line. N = 3 in biological replicates. Unpaired two-sided t-test; *, p < 0.05. ns, not significant. P-values for FIP200$^{-/-}$ vs WT CAPs from left to right: p = $2.14 \times 10^{-25}$, p = 0.0005, p = $4.12 \times 10^{-11}$, p = $5.98 \times 10^{-18}$, p = 0.004, p = $3.48 \times 10^{-17}$, p = $8.97 \times 10^{-6}$, p = $3.32 \times 10^{-11}$, p = 0.012, p = 0.065, p = 0.0446, p = 0.053, p = 0.0392, p = 0.043. p.values for YIPF4$^{-/-}$ vs WT CAPs from left to right: p = 0.494, p = 0.303, p = 0.765, p = 0.002, p = 0.712, p = 0.0013, p = 0.765, p = 0.911, p = 0.967, p = 0.504, p = 0.486, p = 0.349, p = 0.910, p = 0.863. p-values for FIP200$^{-/-}$ vs WT other proteins from left to right: p = 0.000015, p = 0313, p = 0.0331, p = 0.0733, p = 0.784, p = 0.005, p = 0.328, p = 0.0003, p = 0.148, p = 0.602, p = 0.011, p = $9.06 \times 10^{-8}$, p = 0.241, p = 0.415. p-values for YIPF4$^{-/-}$ vs WT other proteins from left to right: p = 0.792, p = 0448, p = 0.557, p = 0.488, p = 0.680, p = 0.519, p = 0.503, p = 0.841, p = 0.919, p = 0.658, p = 0.723, p = 0.227, p = 0.916, p = 0.944. **b**, Violin plots for $Log_2$(−AA/UT) for control, FIP200$^{-/-}$, or YIPF4$^{-/-}$ HeLa cells displayed for various classes of proteins with the indicated sub-cellular localizations for 'autophagy' candidates. Median values are indicated by solid bold line. n = 3 in biological replicates. Unpaired two-sided t-test;

*, p < 0.05. ns, not significant. Data is extracted from panel **a**. p-values for FIP200$^{-/-}$ vs WT from left to right: p = $2.14 \times 10^{-25}$, p = $5.98 \times 10^{-18}$, p = 0.004, p = $3.48 \times 10^{-17}$. p-values for YIPF4$^{-/-}$ vs WT from left to right: p = 0.494, p = 0.002, p = 0.712, p = 0.0013. **c**, Western blot showing Golgi protein levels in WT, FIP200$^{-/-}$, or DKO (YIPF3$^{-/-}$/YIPF4$^{-/-}$) HEK293 cells in response to AA withdrawal (12h). **d**, Quantification of western blots for the indicated Golgi proteins in HEK293 control, FIP200$^{-/-}$, and DKO (YIPF3/YIPF4) either untreated or starved for amino acids for 12h (as in panel **c**) performed in biological triplicate. Unpaired two-sided t-test; *, p < 0.05. ns, not significant. Bars are mean values and error bars represent S.D. p-values for control UT vs -AA from left to right: p = 0.000739, p = 0.03188, p = 0.1489, p = 0.006017, p = 0.027868. p-values for FIP200$^{-/-}$ UT vs -AA from left to right: p = 0.503841, p=0.456941, p = 0.540851, p = 0.11076, p = 0.733579. p.values for DKO (YIPF3/YIPF4$^{-/-}$) UT vs -AA from left to right: p = 0.344926, p = 0.555756, p = 0.398393. **e**, Correlation plot for alterations in protein abundance for proteins in the indicated sub-cellular compartments in HeLa cells after 18h of EBSS for YIPF3$^{-/-}$/WT or YIPF4$^{-/-}$/WT cells (y-axis) versus FIP200$^{-/-}$/WT cells (x-axis). **f**, GALNT2-Keima expressing HEK293 cells (WT, FIP200$^{-/-}$, DKO) were left untreated or subjected to nutrient stress for 16h and then analyzed by flow cytometry. Frequency distributions of 561/405 nm ex. ratios are shown ($n$ = 10,000 cells per condition).

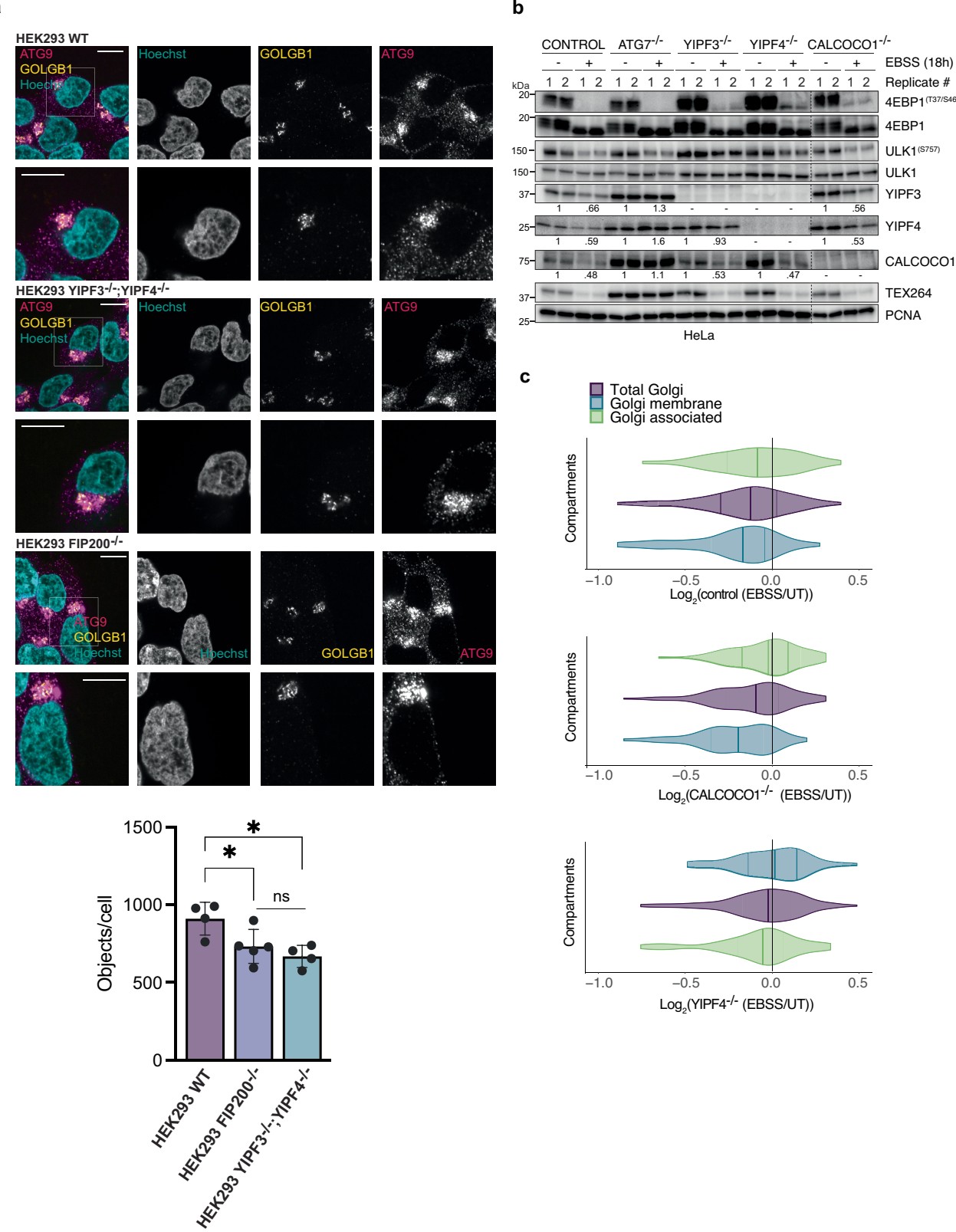

**Extended Data Fig. 11** | See next page for caption.

**Extended Data Fig. 11 | Analysis of YIPF3/4$^{-/-}$ cells for ATG9 vesicles and proteomic analysis reveals no obvious role for CALCOCO1 in Golgi turnover in HeLa cells. a**, HEK293 cells of the indicated genotypes were subjected to immunofluorescence with α-ATG9 (magenta) and α-GOLGB1 (yellow) antibodies. Nuclei were stained with Hoechst (cyan) Scale bars are 10 microns. ATG9 puncta (objects/cell) were quantified using Cell Profiler (lower panel), n = 4 or 5 as indicated by dots. Two-Tailed Mann-Whitney test, *, p-value < 0.05. ns, not significant. Number of cells analyzed for WT, YIPF3/4$^{-/-}$, and FIP200$^{-/-}$ genotypes were 80, 108, and 100, respectively. Bars are mean values and error bars represent S.D. p-values from left to right: p = 0.0317, p = 0.0286, p = 0.5556. **b**, Immunoblots of whole cell extracts from the indicated HeLa control and mutant cells in duplicate either left untreated or subjected to EBSS for 18h using the indicated antibodies. Independently cultured replicate samples were loaded in adjacent lanes and indicated by "Replicate #". α-PCNA was used as a loading control. These blots were visualized with chemiluminescence. We used densitometry of blots of different exposures to estimate signal intensities for YIPF3, YIPF4, CALCOCO1, and TEX264. Signal intensities were averaged across replicates and normalized to untreated cells of the same genotype, and relative values are provided under the corresponding samples. **c**, Violin plot for Golgi-membrane protein Log$_2$FC with or without 18h of EBSS in control, YIPF4$^{-/-}$ or CALCOCO1$^{-/-}$ HeLa cells. Mean abundance is indicated by bold line.

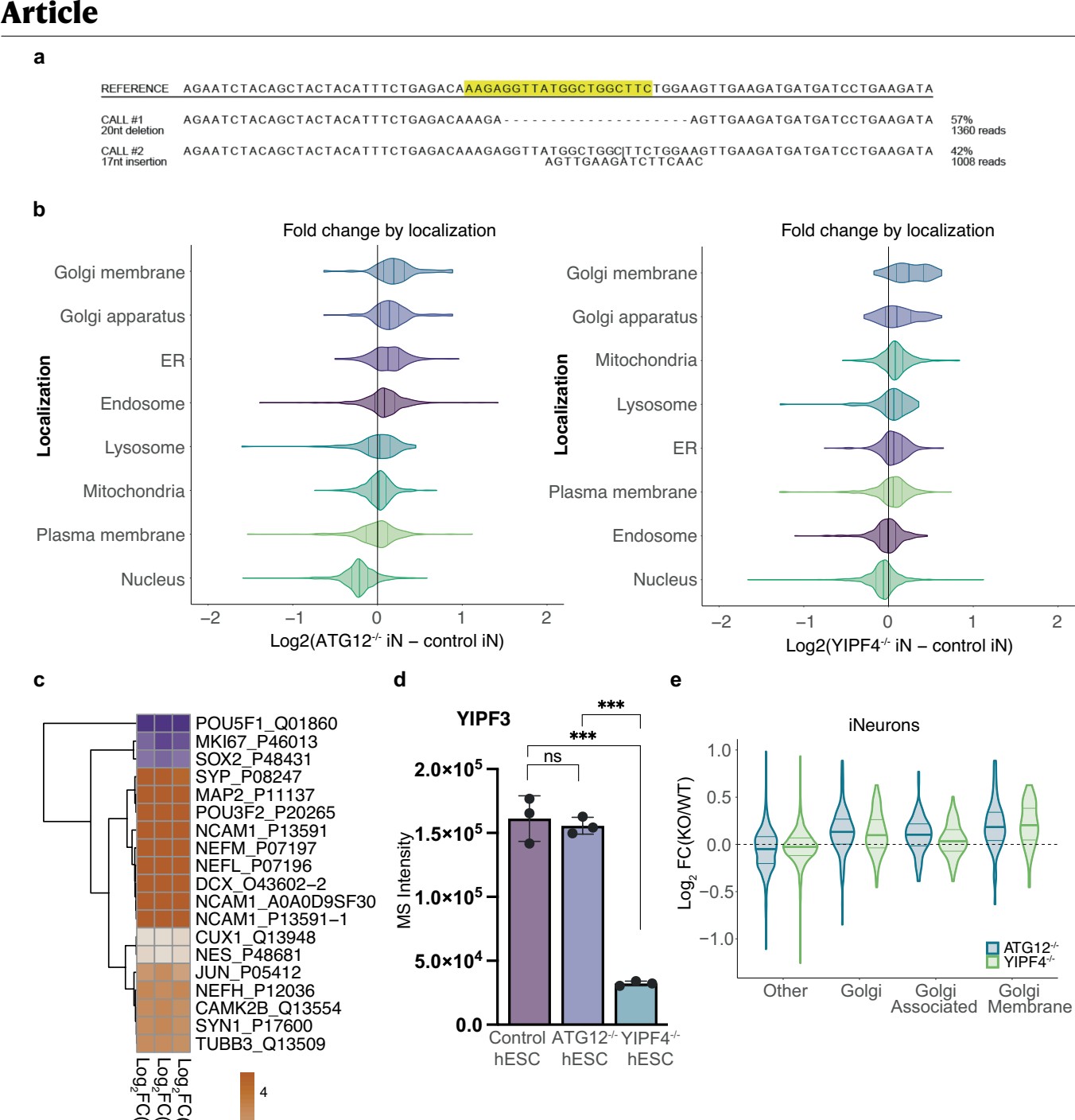

**Extended Data Fig. 12 | Role of YIPF4 in Golgi remodeling during differentiation of human ES cells to iNeurons. a**, Validation of CRISPR/Cas9 mediated deletion of YIPF4 in ES cells. **b**, Log$_2$FC (ATG12$^{-/-}$ iN – control iN) and Log$_2$FC (YIPF4$^{-/-}$ iN – control iN) values for proteins localized in individual subcellular compartments (12 day differentiation). Each sample/condition represents triplicate independent cultures. **c**, Heatmap of relative increase or decrease in the abundance of stem cell or iNeuron marker, comparing iNeurons versus ES cells. Each sample/condition represents triplicate independent

cultures. **d**, Quantification of the abundance of YIPF3 in ATG12$^{-/-}$ or YIPF4$^{-/-}$ ES cells. TMT intensities for triplicate analyses are shown. Unpaired two-sided t-test, p < 0.05, ***. Error bars represent S.D. n.s., not significant. Bars are mean values and error bars represent S.D. p-values from left to right: p = 0.631, p = 0.0002, p < 0.0001. **e**, Violin plots for Log$_2$FC (YIPF4$^{-/-}$/WT) iNeurons (day 12) for the indicated sets of proteins. The extent of Golgi-membrane protein stabilization in YIPF4$^{-/-}$ iNeurons is similar to that seen in ATG12$^{-/-}$ iNeurons.

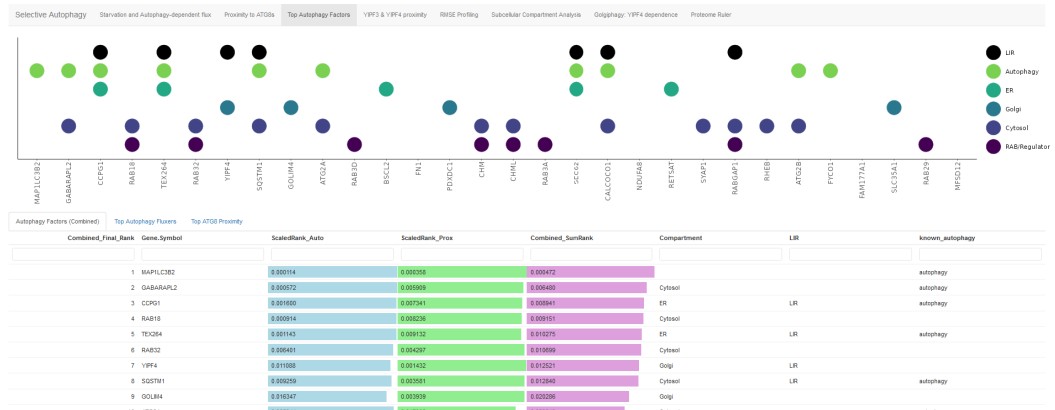

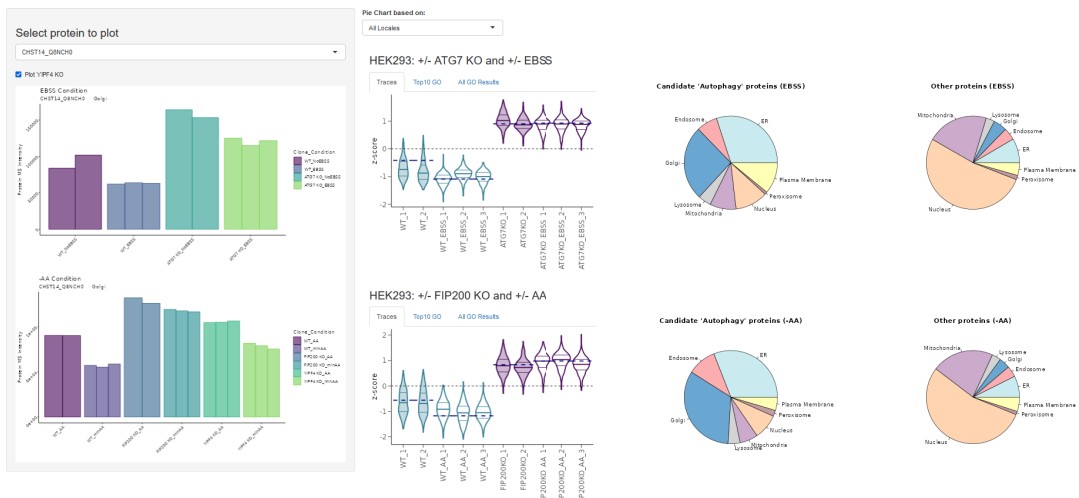

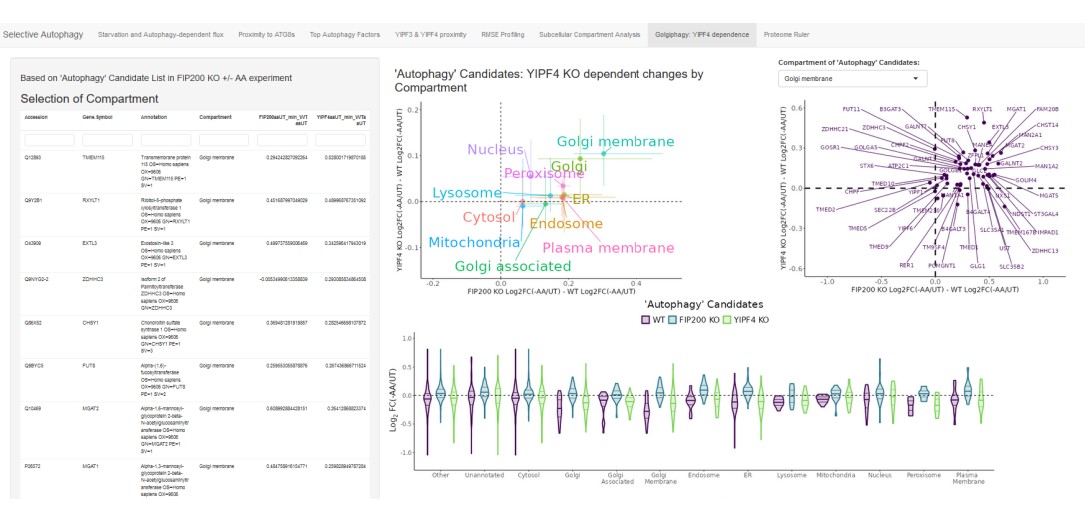

**Extended Data Fig. 13 | CARGO: an interactive website to interrogate Cellular Autophagy Regulation and Golgiphagy data from this work.** The website can be found at: https://harperlab.connect.hms.harvard.edu/ CARGO_Cellular_Autophagy_Regulation_GOlgiphagy/. **a**, Example of visualization data combining orthogonal proteomics methods to create a priority list of putative autophagy factors. **b**, Example of visualization data for CAPs and subcellular compartment analysis. **c**, Example of visualization tools for mapping Golgiphagy and CAPs.

# natureresearch

# Reporting Summary

Nature Research wishes to improve the reproducibility of the work that we publish. This form provides structure for consistency and transparency in reporting. For further information on Nature Research policies, see Authors & Referees and the Editorial Policy Checklist.

## Statistics

For all statistical analyses, confirm that the following items are present in the figure legend, table legend, main text, or Methods section.

| n/a | Confirmed | |
|---|---|---|
| ☐ | ☒ | The exact sample size (*n*) for each experimental group/condition, given as a discrete number and unit of measurement |
| ☐ | ☒ | A statement on whether measurements were taken from distinct samples or whether the same sample was measured repeatedly |
| ☐ | ☒ | The statistical test(s) used AND whether they are one- or two-sided<br>*Only common tests should be described solely by name; describe more complex techniques in the Methods section.* |
| ☒ | ☐ | A description of all covariates tested |
| ☐ | ☒ | A description of any assumptions or corrections, such as tests of normality and adjustment for multiple comparisons |
| ☐ | ☒ | A full description of the statistical parameters including central tendency (e.g. means) or other basic estimates (e.g. regression coefficient) AND variation (e.g. standard deviation) or associated estimates of uncertainty (e.g. confidence intervals) |
| ☐ | ☒ | For null hypothesis testing, the test statistic (e.g. *F*, *t*, *r*) with confidence intervals, effect sizes, degrees of freedom and *P* value noted<br>*Give P values as exact values whenever suitable.* |
| ☒ | ☐ | For Bayesian analysis, information on the choice of priors and Markov chain Monte Carlo settings |
| ☒ | ☐ | For hierarchical and complex designs, identification of the appropriate level for tests and full reporting of outcomes |
| ☒ | ☐ | Estimates of effect sizes (e.g. Cohen's *d*, Pearson's *r*), indicating how they were calculated |

*Our web collection on statistics for biologists contains articles on many of the points above.*

## Software and code

Policy information about availability of computer code

| Data collection | Orbitrap Eclipse Tribrid Mass Spectrometer (Cat#FSN04-10000) with FAIMS Pro Interface (#FMS02-10001) - Thermo Fisher Scientific<br>Orbitrap Fusion Lumos Tribrid MS (Cat#IQLAAEGAAPFADBMBHQ) with or without FAIMS Pro Interface (#FMS02-10001) - Thermo Fisher Scientific<br>Odyssey CLx Imager LI-COR bioscience<br>Nikon Ti motorized microscope equipped with a Nikon Plan Apo 100x/1.40 N.A objective lens, and Hamamatsu ORCA-Fusion BT CMOS camera- Nikon<br>Attune NxT Flow Cytometer (Cat#A28993)- Thermo Fisher Scientific |
|---|---|

| Data analysis | 1. Prism; GraphPad, v9 https://www.graphpad.com/scientific-software/prism/<br>2. SEQUEST-HT ; Eng et al., (1994) J Am Soc Mass Spectrom. 5 (11): 976-989. Implementation in Proteome Discoverer (v2.3.0.420 – Thermo Fisher Scientific)<br>3. Comet (v2018.01 rev. 2); Eng, J.K. et al. (2013), Proteomics 13, 22-24.<br>4. FlowJoTM; Vl0.5.2 https://www.flowjo.com<br>5. lmageStudiolite V 5.2.5 https://www.licor.com/bio/products/software/image_studio_lite<br>6. FiJi ImageJ V.2.0.0 https://imagej.net/Fiji<br>7. Rstudio (1.2.1335) + R(v_4.1.3)<br>8. Adobe Illustrator(CS5(15.0.0))<br>9. Monocole, Rad et al., J. Proteome Res. 20, 591-598 (2021)<br>10. Code and data analysis to generate paper figures can be found on GitHub at https://github.com/harperlaboratory/Golgiphagy.git. All data and data figures can be explored using CARGO (Cellular Autophagy Regulation and GOlgiphagy). CARGO is a ShinyApp interface generated in R and RStudio that can be accessed at https://harperlab.connect.hms.harvard.edu/CARGO_Cellular_Autophagy_Regulation_GOlgiphagy/. |
|---|---|

For manuscripts utilizing custom algorithms or software that are central to the research but not yet described in published literature, software must be made available to editors/reviewers. We strongly encourage code deposition in a community repository (e.g. GitHub). See the Nature Research guidelines for submitting code & software for further information.

# Data

Policy information about availability of data

All manuscripts must include a data availability statement. This statement should provide the following information, where applicable:
- Accession codes, unique identifiers, or web links for publicly available datasets
- A list of figures that have associated raw data
- A description of any restrictions on data availability

Data Availability
All mass spectrometry data for HeLa and HEK293 cells (155 files) have been deposited to the ProteomeXchange Consortium via the PRIDE repository (http://www.proteomexchange.org/): (Project Accession: PXD038358). Proteomic data for ES cells and iNeurons (15 files) is available on Project Accession: PXD043923. All analyzed proteomic data are in Supplementary Tables 1, 2, 4, 5, 6, 7, 8, and 9. Source codes for figures are provided in Source data Table 1 and all uncropped blots are provided in Supplementary Figure 1. We employed canonical protein entries from the Human reference proteome database in our study (UniProt Swiss-Prot – 2019-01; https://ftp.uniprot.org/pub/databases/uniprot/previous_major_releases/release-2019_01/). LIR motifs were based on the iLIR Autophagy Database (http://repeat.biol.ucy.ac.cy/iLIR/).

# Field-specific reporting

Please select the one below that is the best fit for your research. If you are not sure, read the appropriate sections before making your selection.

☒ Life sciences  ☐ Behavioural & social sciences  ☐ Ecological, evolutionary & environmental sciences

For a reference copy of the document with all sections, see nature.com/documents/nr-reporting-summary-flat.pdf

# Life sciences study design

All studies must disclose on these points even when the disclosure is negative.

| Sample size | No sample size calculation was done. For proteomics, we chose n=2, 3 or 4 biological replicates given the limitation of the available TMT channels and extensive work in the field has shown that this approach provides the necessary statistical significance. The number of replicates for all TMT experiments is shown in the schematic in the relevant figure. For flow cytometry, we analyzed >10,000 cells with biological triplicate experiments, which showed consistent results throughout the replication. The number of replicates for immunoblotting experiments is provided in the figure legends and is performed in triplicate unless otherwise noted. Confocal imaging experiments were performed in biological triplicate at a minimum (a subset involved five or 6 replicates). The number of data points in each plot represents the number of replicates used. Sample size was determined based on similar studies in this field. e.g. An et al Systematic quantitative analysis of ribosome inventory during nutrient stress. Nature. 2020 Jul;583(7815):303-309. doi: 10.1038/s41586-020-2446-y |
|---|---|
| Data exclusions | No data were excluded from the analyses. |
| Replication | We confirm that all attempts at replication were successful. The number of biological replicates is provided for each experiment in the figure legend. |
| Randomization | No randomization was necessary. Mass spectrometry and biochemistry samples were measured sequentially. Images were automatically acquired for the data analysis by high throughput imaging based methods. |
| Blinding | No blinding was applied in this study. Blinding was not possible as all samples were analyzed pairwise or multiple samples compared. In all assays in this study the treatment (or different conditions tested) cannot be disguised from the scientist. |

# Reporting for specific materials, systems and methods

We require information from authors about some types of materials, experimental systems and methods used in many studies. Here, indicate whether each material, system or method listed is relevant to your study. If you are not sure if a list item applies to your research, read the appropriate section before selecting a response.

## Materials & experimental systems

| n/a | Involved in the study |
|-----|-----------------------|
| ☐ | ☒ Antibodies |
| ☐ | ☒ Eukaryotic cell lines |
| ☒ | ☐ Palaeontology |
| ☒ | ☐ Animals and other organisms |
| ☒ | ☐ Human research participants |
| ☒ | ☐ Clinical data |

## Methods

| n/a | Involved in the study |
|-----|-----------------------|
| ☒ | ☐ ChIP-seq |
| ☐ | ☒ Flow cytometry |
| ☒ | ☐ MRI-based neuroimaging |

## Antibodies

**Antibodies used**

ATG7 (Cell Signaling Technology, 8558S; RRID:AB_10831194; dilution 1:1000). Lot: 4
FIP200 (Proteintech, 17250-1-AP; RRID: AB_10666428; dilution 1:1000). Lot: 00048639
LC3B (D11) XP(R) (Cell Signaling Technology, 3868; AB_2137707;  dilution 1:1000). Lot: 6
ULK1 (Cell Signaling Technology 8054; RRID:AB_11178668; dilution 1:1000). Lot:6
Phospho-ULK1 (ser757) (Cell Signaling Technology 14202; RRID:AB_2665508; dilution 1:1000). Lot: 5
4E-BP1 (Cell Signaling Technology 9644; RRID:AB_2097841; dilution 1:1000). Lot: 12
Phospho-4E-BP1 (Thr37/46) (Cell Signaling Technology 2855; RRID:AB_560835; dilution 1:1000). Lot: 26
TEX264 (Sigma, HPA017739; RRID:AB_1857910; dilution 1:1000). Lot: 000012723
Tubulin (Abcam, ab131205; RRID: AB_11156121; dilution 1:1000). Lot: GR3251127-3
YIPF3 (Invitrogen PA566621; RRID:AB_2664704; dilution 1:1000). Lot: YF3956672B
YIPF4 (Sino Biological 202844-T46; dilution 1:1000) Lot: HD12JL0934
HSP90 (Santa Cruz Biotechnology sc-69703; AB_2121191; dilution 1:1000). Lot: J2721
CALCOCO1 (Abclonal A7987; RRID:AB_2768684; dilution 1:1000). Lot: 0036240101
LAMP1 (Cell Signaling Technology 9091; RRID:AB_2687579; dilution 1:1000). Lot: 5
GOLGB1/Giantin (abcam ab37266; RRID:AB_880195; dilution 1:1000) Lot: GR3452700-3
GOLGA2 (Proteintech 11308; RRID:AB_2919024; dilution 1:1000). Lot: 00039607
PCNA (Santa Cruz PC10; sc-56 RRID:AB_628110; dilution 1:1000). Lot: L3015
IRDye 800CW Goat anti-Rabbit IgG H+L (LI-COR, 926-32213; AB_621848; dilution 1:10000). Lot: D21104-25
IRDye 680 RD Goat anti-Mouse IgG H+L (LI-COR, 926-680; RRID:AB_10956588; dilution 1:10000). Lot: D00825-11
Goat anti-Rabbit IgG, HRP-linked IgG (Cell Signaling Technology 7074P2, RRID: AB_2099233 dilution 1:10000). Lot: 28
Goat anti-Rabbit IgG HRP conjugate (Bio-Rad 1706515; RRID:AB_11125142; dilution 1:10000). Lot: 64559210
Goat anti-Mouse IgG HRP conjugate Bio-Rad 1706516; RRID:AB_11125547; dilution 1:10000). Lot: 64526160

**Validation**

1. FIP200, YIPF4, YIPF3, CALCOCOl, ATG7 antibody specificity determined by CRISPR deletion or tagging of endogenous gene (see figures Extended Data Fig. 1a-c, 7a).
2. Specificity of Tex264 was determined previously using TEX264-/- cells (Mol Cell, 74, 891 (2019)).
3. PCNA was validated using knockout cells (Biotechniques. 2017;62:80-82).
4. LAMP1 (D2D11) XP® Rabbit mAb recognizes endogenous levels of total LAMP1 protein, as reported by the vendor (https://www.cellsignal.com/products/primary-antibodies/lamp1-d2d11-xp-rabbit-mab/9091).
5. HSP90 antibody has is applicable for WB, RIP, IP, IHC, IF, FC, CoIP, ELISA and shows reactivity with human samples.  (https://www.ptglab.com/products/HSP90-Antibody-60318-1-Ig.htm).
6. Tubulin antibody has been shown to be excellent as a loading control antibody and reacts with human tubulin (https://www.abcam.com/products/primary-antibodies/alpha-tubulin-antibody-dm1a-loading-control-ab7291.html).
7. Giantin antibody is optimized for immunofluorescence and reacts with human giantin (https://www.abcam.com/products/primary-antibodies/giantin-antibody-9b6-golgi-marker-ab37266.html).
8. GOLGB1 Positive WB detected in HeLa and HEK293 cells (https://www.ptglab.com/products/GOLGA2,GM130-Antibody-11308-1-AP.htm).
9. 4E-BP1 (53H11) Rabbit mAb has been characterized by the vendor and detects endogenous levels of total human 4E-BP1 protein (https://www.cellsignal.com/products/primary-antibodies/4e-bp1-53h11-rabbit-mab/9644).
10. Phospho-4E-BP1 (Thr37/46) (236B4) Rabbit mAb has been characterized by the vendor and detects endogenous levels of 4E-BP1 only when phosphorylated at Thr37 and/or Thr46.(https://www.cellsignal.com/products/primary-antibodies/phospho-4e-bp1-thr37-46-236b4-rabbit-mab/2855).
11. GOLGA2 has been shown by the vendor to work for WB and IF applications and shows reactivity with human samples (https://www.ptglab.com/products/GOLGA2,GM130-Antibody-11308-1-AP.htm).
12. ULK1 (D8H5) Rabbit mAb recognizes endogenous levels of total human ULK1 protein, as validated by the vendor. (https://www.cellsignal.com/products/primary-antibodies/ulk1-d8h5-rabbit-mab/8054).
13. Phospho-ULK1 (D7O6U) Rabbit mAb recognizes endogenous levels of ULK1 protein only when phosphorylated at Ser758 of human ULK1 and ser757 in mouse ULK1 (https://www.cellsignal.com/products/primary-antibodies/phospho-ulk1-ser757-d7o6u-

# Eukaryotic cell lines

Policy information about cell lines

| | |
|---|---|
| Cell line source(s) | Human: HEK293 ATCC CRL-1573; RRID:CVCL_0045<br>Human: HeLa ATCC CCL-2; RRID: CVCL_0030<br>Human ES cells (clone H9), WiCell |
| Authentication | ATCC and WiCell preforms quality testing to ensure authentication of cell lines using Short Tandem Repeat (STR) analysis. Additionally, WiCell performs karyotyping on ES cells. No additional authentications were preformed. karyotyping (GTG-banded karyotype) of HEK293 and HeLa cells (from ATCC) was also performed by the cytogenomics core laboratory at Brigham and Women's Hospital. |
| Mycoplasma contamination | All cell lines were found to be free of mycoplasma using Mycoplasma Plus PCR assay kit (Agilent). |
| Commonly misidentified lines<br>(See ICLAC register) | none |

# Flow Cytometry

## Plots

Confirm that:

☒ The axis labels state the marker and fluorochrome used (e.g. CD4-FITC).

☒ The axis scales are clearly visible. Include numbers along axes only for bottom left plot of group (a 'group' is an analysis of identical markers).

☒ All plots are contour plots with outliers or pseudocolor plots.

☒ A numerical value for number of cells or percentage (with statistics) is provided.

## Methodology

| | |
|---|---|
| Sample preparation | No tissue processing were used. |
| Instrument | Attune NxT Flow Cytometer- Thermo Fisher Scientific |
| Software | FlowJoTM; V10.5.2 https://www.flowjo.com |
| Cell population abundance | 10,000 cells were recorded per replicate |
| Gating strategy | 1. live cells were gated by SSC1 hight/FSC1 hight (G1) followed by live cells by SSC1 hight/SSC1-width (G2). 2. Keima signal was measured by 405ex/620(20)em and 561ex/620(20)em and data exported to prism for ratio-metric calculation. |

☒ Tick this box to confirm that a figure exemplifying the gating strategy is provided in the Supplementary Information.

