## [Peer Review File · Nature]

Manuscript Title: Proteome census upon nutrient stress reveals Golgiphagy membrane receptors

Reviewer Comments & Author Rebuttals

Reviewer Reports on the Initial Version:

Referees' comments:

Referee #1 (Remarks to the Author):

In this manuscript, Hickey et al. set out to map the cargo spectrum of bulk autophagy during nutrient stress and define the selectivity of this process. Using TMT proteomics to measure protein abundance changes in parental and autophagy defective cells grown in fed and starved conditions the authors identified ~700 candidate autophagy substrates among which ER and Golgi proteins were strongly enriched. Unexpected according to popular belief neither abundant cellular protein complexes nor cytosolic proteins featured prominently among these candidates. Focusing on understudied golgiphagy the authors employed proximity proteomics and abundance profiling to identify the dimerizing Golgi proteins YIPF4 and YIPF3 as two potential receptors for this pathway. Consistent with their proposed role, a number of known autophagy proteins (including ATG8 family members) were found in the neighborhood of YIPF4 and YIPF3. Binding assays confirmed that both proteins interact with ATG8 proteins in a LIR dependent manner. Importantly, the authors went on to show that endogenously tagged YIPF4 is found in smaller vesicular structures following starvation and that these puncta formed in a BafA1 and VPS34 dependent manner and colocalize with LAMP1 and LC3B. Lastly, the authors again performed protein abundance profiling in fed and starved YIPF3 and YIPF4 double KO cells and unveiled a considerable number of Golgi proteins whose autophagic turnover was blocked to a similar extent that in FIP200 KO cells. The authors confirmed some of these YIPF3/YIPF4 targets by WB and impressively even by Keima assays which the authors benchmarked with YIPF3 and YIPF4. Overall, the work of Hickey is a superb study which elegantly does the split between an important resource to the proteostasis and autophagy fields and a mechanistic study of a poorly studied selective autophagy pathway. Importantly, the authors also clear up with the concept that starvation-induced autophagy is a bulk and unspecific process. Since this is already a complete study, I do only have a minor comment which the authors may want to address.

Given that GABARAPL1 was detected as the most prominent proximity partner, it would be more coherent to blot for GABARAPL1 (instead of LC3B) in the IP shown in Figure 4h.

Referee #2 (Remarks to the Author):

Hickey and colleagues here describe the identification of a pathway called "Golgiphagy". In a broad view they provide extensive, beautifully done experiments which suggest that Golgi membrane proteins (most enriched are single pass membrane proteins) are more likely to be targets of autophagy than typical cytosolic proteins which have been seen as bulk cargo. This data is well described and compelling. They use several elegant approaches and appropriate controls (ATG7^{-/-}, FIP200^{-/-}, EBSS, APEX proximity, ATG8 protein mutants and rescues, LIR and LDS rescues etc.) Using these systems, and tools they identify two 5 TM spanning Golgi proteins YIPF3 and YIPF4 which they propose are receptors which bind to ATG8s and mediate Golgiphagy. They obtain proteomic data catalogued in Supplementary tables which list the putative Golgi membrane proteins which would be cargo. However, in Figure 6 they only validate one GALNT2, which is one of several glycosyltransferases found in their putative candidate list.

This validate is nice but very standard.

The issue is not what is in the manuscript but what is not. Given the APEX data from YIPF3 and YIPF4, and the experience of the Harper lab in organelle-IP this manuscript lacks the identification of the vesicles that YIPF3 and 4 are in or helping to form (see model in Figure 6) and the impact of manipulating these vesicular carriers on some aspect of cell function, (eg secretion, mobility, endocytosis, signalling). Other aspects to be addressed - are the vesicles coated?, how do they pinch off? do they bind motors? And some hint about the bigger question, why would the cell want to eat its Golgi during autophagy?

Referee #3 (Remarks to the Author):

In their study, Hickey et al. employ quantitative proteomic approaches and functional assays to investigate the selectivity of macroautophagy during nutrient stress.

By decoupling autophagic turnover from other processes that affect protein abundance, such as translational suppression and proteasomal degradation, the authors provide a global assessment of macroautophagy selectivity during nutrient stress. This approach yields both known and new autophagy cargo-associated proteins (CAPs).

Among the newly identified CAPs are YIPF3 and YIPF4, Golgi resident proteins that are shown to interact with ATG8 proteins in a starvation-dependent and LIR-dependent manner, as evidenced by proximity biotinylation proteomics and direct immunoprecipitation experiments. YIPF3 and YIPF4 mobilize into vesicles in an autophagy-dependent process, leading to their degradation and coordinating the autophagic turnover of primarily Golgi-membrane proteins.

This is a novel and well executed study, which reveals for the first time the ability of autophagy to target the Golgi apparatus.

Despite the novelty and strengths of the manuscript, the mechanisms underlying the coupling of YIPF3 and YIPF4 with the autophagy machinery require further elucidation.

Moreover, while the study suggests the possibility of Golgi-phagy being involved in Golgi quality control processes and signals beyond nutrient stress, its physiological roles are not elucidated.

Specific suggestions for improvement are listed below.

Comments

1. The authors detect YPF3 and YPF4 in APEX experiments with both GABARAPL2 and LC3B. Also, YPF3/4 bind to LC3B in a LIR-dependent manner (Fig. 4h). Is there a preference of YPF3/4 for LC3B vs GABARAPL2, or other ATG8 proteins?

2. Is YPF3-YPF4 dimerization required for this interaction? The authors could express recombinantly the individual N-terminal regions of YPF3 and YPF4 and test their sufficiency for binding to various ATG8 family proteins.

3. Although the capture of YPF4-positive vesicles into autophagosomes/lysosomes is shown convincingly (Fig. 5), the mechanism of capture remains unclear. The authors rule out ubiquitin-binding adaptors based on the lack of effect of the E1 inhibitor TAK243, but such treatment could have pleiotropic effects. This reviewer's suggestion is to systematically identify (via direct immunostaining and/or immunoblotting) the autophagic adaptors recruited to the starvation-induced YPF4 vesicles, then knock them down individually or in subsets to rule in/rule out their involvement.

4. Connected to the previous point, some mechanistic insight for the budding/excision of the YPF4-positive vesicles from the Golgi would strengthen the manuscript. For example, are the canonical

coats involved in budding from the Golgi (Rab, Arf GTPases) required for Golgi-phagy?

5. To get more insights into the clients of YPF3/4, as well as possibly on the mechanism of budding, the authors could carry out pull down and proteomic profiling of these vesicles using suitable affinity tags.

6. The physiological relevance of YPF3/4-dependent Golgi-phagy remains undefined. While the study provides valuable insights into the selectivity of macroautophagy and the role of YIPF3 and YIPF4 in Golgiphagy, it is essential to discuss the implications of these findings in the context of cellular homeostasis, adaptation to nutrient stress, and specific biological processes.

- Do other stressors beside starvation induce Golgi-phagy, such as defective glycosylation of cargo proteins?
- Is the autophaged Golgi apparatus damaged/non-functional (as is the case in mitophagy)?

7. While the study primarily focuses on the identification and characterization of YIPF3 and YIPF4 as receptors for Golgiphagy, it would be necessary to elucidate the functional implications of Golgiphagy in cellular processes. How are core Golgi functions such as protein trafficking, glycosylation or secretion affected by YPF3/4 loss?

8. Are there any diseases associated with YIPF3/4 gene mutations or SNPs? What about other identified CAPs?

Minor points

9. In figure 5, it would be beneficial to see the FIP200 $-/-$ cells (these are included in the supplemental, but would be beneficial to see side-by-side)

Author Rebuttals to Initial Comments:

RESPONSE TO REVIEWER COMMENTS – Nature manuscript 2023-03-05432

Introductory comments: We appreciate the opportunity to resubmit our manuscript. In response to the reviewer comments, we communicated with the Editor concerning possible additions to the paper that would address the reviewer's main comments concerning physiological pathways where Golgiphagy may be operational. Although we had originally intended to develop these findings further with this ultimately being a separate study delving into the potential role of Golgiphagy in neuronal proteome remodeling, in consultation with the editor, it was decided that this data could be added in its current form to address the reviewer comments concerning physiological pathways where Golgiphagy is operative.

Specifically, we have added experiments examining the role of YIPF4 in turnover of Golgi during conversion of ES cells to induced neurons (iNeurons) in vitro. We had previously observed defects in Golgi and ER remodeling during conversion of ES cells to induced neurons in cells lacking autophagy (ATG12^{-/-}) (Ordureau et al., 2021). This is a well-validated system for studying basic properties of neurogenesis, as proteome remodeling and the formation of axons and dendrites are highly stereotyped and efficient. Differentiation is initiated by a regulatable transcription factor (NGN2). Due to the existence of autophagy dependent Golgi remodeling during differentiation, we hypothesized that YIPF3/4 may play a role as receptors in this context. Indeed, we found that the pattern of Golgi protein accumulation during differentiation observed in cells lacking ATG12 is paralleled in cells lacking YIPF4, marking YIPF4 as a major Golgiphagy receptor in this system. Interestingly, when we look at the behavior of candidate autophagy proteins (CAPs) from our nutrient stress data, we see strikingly similar behavior in the iNeurons lacking YIPF4 or ATG12, suggesting an overlapping cohort of Golgi proteins are being selected for autophagy in two very different experimental systems.

Together, these new data support the conclusion that YIPF4 and its interacting partner YIPF3 function broadly as receptors for programmed Golgiphagy, and we believe this addition, and data to address other reviewer comments strengthens our manuscript.

Reviewer comments are in blue font and our responses in black font.

Referee #1 (Remarks to the Author):

In this manuscript, Hickey et al. set out to map the cargo spectrum of bulk autophagy during nutrient stress and define the selectivity of this process. Using TMT proteomics to measure protein abundance changes in parental and autophagy defective cells grown in fed and starved conditions the authors identified ~700 candidate autophagy substrates among which ER and Golgi proteins were strongly enriched. Unexpected according to popular belief neither abundant cellular protein complexes nor cytosolic proteins featured prominently among these candidates. Focusing on understudied golgiphagy the authors employed proximity proteomics and abundance profiling to identify the dimerizing Golgi proteins YIPF4 and YIPF3 as two potential receptors for this pathway. Consistent with their proposed role, a number of known autophagy proteins (including ATG8 family members) were found in the neighborhood of YIPF4 and YIPF3. Binding assays confirmed that both proteins interact with ATG8 proteins in a LIR dependent manner. Importantly, the authors went on to show that endogenously tagged YIPF4 is found in smaller vesicular

structures following starvation and that these puncta formed in a BafA1 and VPS34 dependent manner and colocalize with LAMP1 and LC3B. Lastly, the authors again performed protein abundance profiling in fed and starved YIPF3 and YIPF4 double KO cells and unveiled a considerable number of Golgi proteins whose autophagic turnover was blocked to a similar extent that in FIP200 KO cells. The authors confirmed some of these YIPF3/YIPF4 targets by WB and impressively even by Keima assays which the authors benchmarked with YIPF3 and YIPF4. Overall, the work of Hickey is a superb study which elegantly does the split between an important resource to the proteostasis and autophagy fields and a mechanistic study of a poorly studied selective autophagy pathway. Importantly, the authors also clear up with the concept that starvation-induced autophagy is a bulk and unspecific process. Since this is already a complete study, I do only have a minor comment which the authors may want to address.

Given that GABARAPL1 was detected as the most prominent proximity partner, it would be more coherent to blot for GABARAPL1 (instead of LC3B) in the IP shown in Fig 4h.

We thank the reviewer for their positive comments and excitement for our manuscript. In the original version, we demonstrated association of LC3B with GFP-YIPF3 in co-IP western experiments. The reviewer asked whether YIPF3/4 can also associate with GABARAPs, which we identified in the APEX experiments. To address the comment above, we have repeated our immunoprecipitation experiment as in Fig 4h, but using mCherry-YIPF4 as a handle, and probing for GABARAP proteins (antibody against GABARAP, L1 and L2; i.e. pan-GABARAP) and LC3B. We are able to readily co-immunoprecipitate YIPF3, and LC3B and GABARAP proteins, in line with our APEX datasets, and there was again an increase in association of the lipidated form of GABARAP in response to starvation. We have replaced the existing panel with this data (Fig 3f), which includes additional untagged controls. Please see additional related comments for reviewer 3 that addresses the requirements for LC3 and GABARAP proteins in Golgiphagy.

Referee #2 (Remarks to the Author):

Hickey and colleagues here describe the identification of a pathway called "Golgiphagy". In a broad view they provide extensive, beautifully done experiments which suggest that Golgi membrane proteins (most enriched are single pass membrane proteins) are more likely to be targets of autophagy than typical cytosolic proteins which have been seen as bulk cargo. This data is well described and compelling. They use several elegant approaches and appropriate controls (ATG7^{-/-}, FIP200^{-/-}, EBSS, APEX proximity, ATG8 protein mutants and rescues, LIR and LDS rescues etc.) Using these systems, and tools they identify two 5 TM spanning Golgi proteins YIPF3 and YIPF4 which they propose are receptors which bind to ATG8s and mediate Golgiphagy.

Point 1. They obtain proteomic data catalogued in Supplementary tables which list the putative Golgi membrane proteins which would be cargo. However, in Figure 6 they only

validate one GALNT2, which is one of several glycosyltransferases found in their putative candidate list.

We thank the reviewer for their evaluation of our manuscript, and the comment that our work is well described and compelling. In Fig 6d (now extended Fig 10c), we used western blotting of several Golgi proteins that were identified as CAPs to examine their turnover in response to nutrient stress. These data broadly validated our proteomics, and the blots from multiple experiments were quantified in Extended Data Fig 9b (now extended Fig 10d). Based on this experiment, we tried to make Keima fusions with several candidates, but so far only GALNT2 was properly localized and stably expressed. Because many Golgi membrane proteins have complex biosynthetic pathways, it may not be straightforward to generate such reporters broadly, as tagging either the N or C-terminus can be detrimental to their trafficking. We hope that the orthogonal immunoblotting experiments are sufficient to address this point.

This validate is nice but very standard.

Point 2. The issue is not what is in the manuscript but what is not. Given the APEX data from YIPF3 and YIPF4, and the experience of the Harper lab in organelle-IP this manuscript lacks the identification of the vesicles that YIPF3 and 4 are in or helping to form (see model in Figure 6)

We appreciate that the reviewer feels that there are no issues with the data that is in our manuscript. The reviewer asks if we can perform organelle-IPs on the YIPF3/4 vesicles. We reviewed how this component of the paper was written, and we realized that we had not effectively communicated the current data on YIPF3/4-containing vesicles. The vesicles that our model proposes are ones in which YIPF3/4 are within autophagosomes that have engulfed Golgi membranes and subsequently fused with lysosomes (in this case they become autolysosomes because we add BafA to prevent degradation). We do not think that YIPF3/4 promote the formation of vesicles that are released from Golgi and then subsequently captured by autophagosomes, but rather that they direct the autophagy machinery to specific regions of the Golgi and allow templating of the phagophore on the Golgi membrane via interactions with ATG8 [akin to current models for ER-phagy [PMID: 37225994; PMID: 37225996]. This idea is supported by the finding that the YIPF4-containing vesicles are not released from Golgi in response to starvation if either VPS34 is inhibited or FIP200 is deleted. These two proteins play key essential steps in initiation of phagophore assembly on membrane cargo, as described in detail previously for both mitophagy and ER-phagy. This idea is further supported by the necessity of BafA to observe the accumulation of mNEON-YIPF4 puncta in response to starvation (manuscript Fig 4), which would not be necessary if they were another type of secretory/endosome/atg9 vesicles.

To address the reviewer's comment and further validate this model, we sought to directly observe mNEON-YIPF4 inside LC3-positive autophagosomes/autolysosomes in live cells. We integrated a mCherry-LC3B fusion protein expression cassette into our endogenously

tagged mNEON-YIPF4 cell line. We then starved cells to initiate autophagy and monitored colocalization of LC3B-positive autophagosomes/autolysosomes (We have to perform these experiments in the presence of BafA to prevent mNEON-YIPF4 quenching as it is trafficked to the lysosome, so we cannot differentiate autophagosomes from autolysosomes). We find that mNEON-YIPF4 signal is encased within LC3B-positive vesicles within single confocal images sliced through the cell. In addition, we find that the mNEON signal clearly moves with the LC3-positive vesicle over several frames of the live-cell imaging session, confirming colocalization. This behavior is analogous to that seen with the extensively validated ER-Phagy receptor TEX264, as described previously (An et al 2019). We believe that this result strongly strengthens an important conclusion of the paper.

The reviewer suggested organelle IPs to examine the YIPF3/4-positive vesicles. Our data demonstrate that YIPF3/4 positive structures are encased in either an autophagosome or are within lysosomes, once the autophagosome is fused with a lysosome. As such, the only way the YIPF3/4 proteins would be accessible to antibodies for vesicle-IP is if the autophagic or lysosomal membrane was disrupted with detergent, which would also break any alternative Golgi-derived vesicles containing YIPF3/4. In addition, in the context of attempts to use YIPF3/4 as an organelle IP handle, we would no doubt have to purify any vesicles away from the much more abundant Golgi itself in order to specifically profile the vesicles. This would require development of a new method which we feel would be very challenging, and beyond the scope of the paper.

Taken together, we believe that the new imaging data provided with our APEX2 proximity data that enables the identification of interacting proteins within such autophagosome/autolysosomes, makes a strong case for autophagic capture of YIPF3/4 containing Golgi membranes upon nutrient stress through an autophagy-dependent process akin to that of ER-phagy. We also note that our previous study (PMID: 24695223) using proteomics of purified autophagosomes from MCF10A cells treated with chloroquine detects YIPF3 as being enriched, providing additional evidence that it can be captured within autophagosomes. We have updated the text to clarify our model that YIPF3/4 positive puncta represent capture within autophagosomes/autolysosomes.

Point 3...and the impact of manipulating these vesicular carriers on some aspect of cell function, (eg secretion, mobility, endocytosis, signalling).

We very much appreciate the question. We estimate that only ~5-8% of the Golgi is degraded by autophagy over a 12 hour period of nutrient stress. Therefore, ostensibly one would not expect an absence of Golgiphagy to dramatically affect the overall performance of the Golgi in terms of protein modification/secretion. However, assessing this is difficult since starvation will cause a dramatic shutdown in translation, which will greatly affect flux of secreted and membrane proteins through the Golgi complex and make challenging any attempt to measure/interpret an effect of YIPF3/4 deletion.

We did however, try to assay whether loss of YIPF3 and YIPF4 would affect Golgi morphology. Using immunofluorescence in the YIPF3/4 double KO cells, we do not observe any defects at the level of confocal microscopy (i.e. no evidence of fragmentation, for example). We have added these imaging panels into Extended data Fig 11a. In addition, we examined whether YIPF3/4 affect the formation of phagosome precursor membranes, ATG9 vesicles, that are derived from the Golgi. Because other forms of autophagy (e.g. ER-phagy) still occur in YIPF3/4 mutants (as noted in the paper), we expect that ATG9 vesicle release from the Golgi is not affected. However, we have directly examined this point using ATG9 immunofluorescence and have added the data to extended data Fig 11a. We still observed abundant ATG9-positive structures in the cytosol, comparable with WT cells, so there does not seem to be an effect on release of ATG9 vesicles from the Golgi upon FIP200 or YIPF3/4 deletion.

Point 4...Other aspects to be addressed - are the vesicles coated?,

As described above, we believe that the visualized puncta of mNEON-YIPF4 represent Golgi membranes within autophagosomes/ autolysosomes, rather than being independent vesicles. Because of this, we do not expect the structures to be coated vesicles. Consistent with this, COPI coat proteins do not feature prominently in the Golgi-associated proteins that are degraded by autophagy. We have plotted this data and added it to Extended data Fig 4k-l, and added additional clarification regarding this in the text. The absence of turnover of coat proteins by autophagy makes it unlikely that such complexes together with YIPF3/4 containing vesicles are the target for autophagy.

Point 5...how do they pinch off?

A major question in the field of selective organelle-phagy concerns how autophagosomes pinch off target organelle membranes. Unfortunately, at this point, even for the best understood process of this kind - ERphagy - there is no established mechanism for membranes being severed to allow full capture within an autophagosome. Moreover, there are no established methods to measure this biophysical process that we are aware of. Even in the very well-developed Nature papers on ER-phagy that recently appeared (PMID: 37225994, 37225996), there is no indication of how the pinching mechanism occurs. We believe that there are likely membrane fusion proteins that facilitate these processes, but that there are likely broader mechanisms of general membrane dynamics. We have added this as a future direction and have included question marks in the summary/model figure.

Point 6...do they bind motors?

As described above, our live cell imaging experiment using mCherry-LC3B and mNEON-YIPF4 suggest that these vesicles are autophagosomes/autolysosomes, and there is evidence that such organelles can in fact traffic on microtubules toward a perinuclear region, sometime near the mitotic organizing center. Since the mitotic organizing center is

near the Golgi, it may be that trafficking is limited to a relatively local movement. In our live cell imaging time course, we do observe limited movement of some mNEON-YIPF4/mCh-LC3B positive puncta, which remain co-incident through the trafficking process (see Fig 4i,j and movie 1). However, although we think it's an interesting future direction, we feel that defining relevant motors and adaptors is likely required to address this point and we would submit is beyond the scope of this study.

Point 7...And some hint about the bigger question, why would the cell want to eat its Golgi during autophagy?

This is an intriguing evolutionary question. We speculate that ER and Golgi membrane bound organelles have evolved specific receptors to mediate selective turnover of organelles. Likely this is a mechanism where organelles experiencing a myriad of stressors can not only selectively recycle membrane proteins, but also lipids that are contained in either the ER or the Golgi. Since both of these classes of macromolecules are beneficial for surviving nutrient stress, cells may have optimized the process for recycling of both types of molecules through a common mechanism. To address this important point, we have further expanded this topic, which was already touched upon, in the discussion.

Referee #3 (Remarks to the Author):

In their study, Hickey et al. employ quantitative proteomic approaches and functional assays to investigate the selectivity of macroautophagy during nutrient stress.

By decoupling autophagic turnover from other processes that affect protein abundance, such as translational suppression and proteasomal degradation, the authors provide a global assessment of macroautophagy selectivity during nutrient stress. This approach yields both known and new autophagy cargo-associated proteins (CAPs).

Among the newly identified CAPs are YIPF3 and YIPF4, Golgi resident proteins that are shown to interact with ATG8 proteins in a starvation-dependent and LIR-dependent manner, as evidenced by proximity biotinylation proteomics and direct immunoprecipitation experiments. YIPF3 and YIPF4 mobilize into vesicles in an autophagy-dependent process, leading to their degradation and coordinating the autophagic turnover of primarily Golgi-membrane proteins.

This is a novel and well executed study, which reveals for the first time the ability of autophagy to target the Golgi apparatus.

Despite the novelty and strengths of the manuscript, the mechanisms underlying the coupling of YIPF3 and YIPF4 with the autophagy machinery require further elucidation. Moreover, while the study suggests the possibility of Golgi-phagy being involved in Golgi quality control processes and signals beyond nutrient stress, its physiological roles are not elucidated.

Specific suggestions for improvement are listed below.

We appreciate the reviewer's positive comments, indicating that this is a "novel and well executed study"

Comments

1. The authors detect YPF3 and YPF4 in APEX experiments with both GABARAPL2 and LC3B. Also, YPF3/4 bind to LC3B in a LIR-dependent manner (Fig. 4h). Is there a preference of YPF3/4 for LC3B vs GABARAPL2, or other ATG8 proteins?

We appreciate the reviewer's comment. We and others have performed systematic genetic studies on the 6 ATG8 proteins (LC3A,B,C and GABARAP, GABARAPL1, and GABARAPL2) [PMID: 29038162, PMID: 27864321]. Cells lacking all 3 LC3 proteins display largely wild-type autophagic flux, while in contrast, cells lacking the 3 GABARAPs accumulate ubiquitin aggregates, autophagy receptors, and display defects in Parkin-dependent mitophagy, all of which can be rescued by introduction of any single GABARAP protein. Moreover, a single GABARAP can rescue a delta-ATG8 hexa-knockout. Taken together, these data suggest that GABARAPs are functionally necessary for these types of autophagic flux, but that the specific form isn't a dominant feature (i.e. functional redundancy). In addition, because these are small proteins, they are often difficult to detect in proteomic studies and the availability of antibodies that can distinguish, for example, individual GABARAPs is limited. As such, we feel that the precise identity of ATG8 detected may not be particularly relevant in terms of biological functions. Operationally, cells expressing members of both classes of ATG8 proteins will have both classes present in individual autophagosomes and therefore allow cargo to be detected by for example APEX fused with either class of ATG8 proteins, or both classes of ATG8 proteins detected when the target membrane (cargo) is APEX modified.

Nevertheless, in order to address this question further, we performed additional immunoblotting of YIPF3 immune complexes from cells engineered to ectopically express tagged YIPF3/YIPF4 with or without LIR motifs in YIPF3/4 DKO cells. We demonstrated association with GABARAPs, as well as LC3B, in a LIR dependent manner (see detailed description in the response to reviewer 1). Thus, YIPF3/4 can bind to both ATG8 subfamilies in cells in a LIR-dependent manner. We have added this data into Fig 3f.

2. Is YPF3-YPF4 dimerization required for this interaction? The authors could express recombinantly the individual N-terminal regions of YPF3 and YPF4 and test their sufficiency for binding to various ATG8 family proteins.

In terms of the dimerization question, our data from co-immunoprecipitation and proteomics show a LIR dependent interaction between YIPF3/4 and ATG8s. Therefore, we believe that this is the domain on YIPF3/4 that is necessary for ATG8 interaction. As for sufficiency, this is more difficult to directly examine. In particular, we found that in the

absence of YIPF4, YIPF3 is absent (presumably rapidly degraded via a quality control pathway). In this case, Golgiphagy is defective based on proteomics and flux measurements, therefore implying that this Golgiphagy pathway is non-functional. Given that YIPF3 abundance apparently depends on its interaction with YIPF4, we are unsure technically how we would control or interpret an experiment that employs mutants at the YIPF3-YIPF4 interface. It is not clear whether the proteins would be properly trafficked to the Golgi and assemble properly in the membrane, as indicated by the loss of YIPF3 in YIPF4^{-/-} cells.

In terms of recombinant N-terminal region binding, although this experiment could be done, our finding that both LC3B and GABARAPs co-precipitate with YIPF3/4 in cells and this requires both the LIR motif and is blocked by the LDS mutant in ATG8 proteins strongly suggests a functional interaction in cells. To further examine this point, we performed YIPF3/4 turnover experiments using a knockout cell line of all 6 ATG8s (HKO), a cell line lacking all three LC3 proteins (A,B,C – Δ LC3), and cells lacking all three GABARAPs (Δ RAP) to demonstrate a requirement for the relevant class of ATG8 proteins. Our data demonstrate: 1) Δ LC3 cells maintain the ability to degrade YIPF3/4 during nutrient stress similar to WT cells, 2) Δ RAP and HKO cells are defective in YIPF3/4 degradation during nutrient stress. This data demonstrates that ATG8s, and specifically the GABARAPs, are necessary for Golgiphagy, similar to other types of autophagy, as reported previously [PMID: 29038162, PMID: 27864321].

3. Although the capture of YPF4-positive vesicles into autophagosomes/lysosomes is shown convincingly (Fig. 5), the mechanism of capture remains unclear. The authors rule out ubiquitin-binding adaptors based on the lack of effect of the E1 inhibitor TAK243, but such treatment could have pleiotropic effects. This reviewer's suggestion is to systematically identify (via direct immunostaining and/or immunoblotting) the autophagic adaptors recruited to the starvation-induced YPF4 vesicles, then knock them down individually or in subsets to rule in/rule out their involvement.

To directly address the possibility that ubiquitin-binding autophagy receptors are required for Golgiphagy, we performed western blotting to measure YIPF3/4 turnover in response to amino acid withdrawal in HeLa cells lacking the 5 ubiquitin-binding adaptors (OPTN, TAX1BP1, CALCOCO2, p62/SQSTM1, and NBR1). We also used TEX264 as a positive control, as this ER-phagy receptor functions independently of these ubiquitin binding adaptors. Our data show that like TEX264, YIPF3 and YIPF4 do not require the ubiquitin binding autophagy receptors, consistent with our imaging experiments in the presence of an E1 inhibitor. We believe that this straightforward experiment directly addresses the functional requirement for these adaptors, and we have added the data into extended data Fig 9f.

4. Connected to the previous point, some mechanistic insight for the budding/excision of the YPF4-positive vesicles from the Golgi would strengthen the manuscript. For example,

are the canonical coats involved in budding from the Golgi (Rab, Arf GTPases) required for Golgi-phagy?

As described above in response to reviewer 2, we do not see coated vesicle proteins being degraded by autophagy in response to nutrient stress. Therefore, this data suggests that canonical coat proteins are not part of the YIPF3/4 autophagosome encapsulation mechanism.

5. To get more insights into the clients of YPF3/4, as well as possibly on the mechanism of budding, the authors could carry out pull down and proteomic profiling of these vesicles using suitable affinity tags.

We address this point in detail in the response to Point 2 from reviewer 2. We refer Reviewer 3 to the detailed comments above rather than repeating the comments here.

6. The physiological relevance of YPF3/4-dependent Golgi-phagy remains undefined. While the study provides valuable insights into the selectivity of macroautophagy and the role of YIPF3 and YIPF4 in Golgiphagy, it is essential to discuss the implications of these findings in the context of cellular homeostasis, adaptation to nutrient stress, and specific biological processes.

- Do other stressors beside starvation induce Golgi-phagy, such as defective glycosylation of cargo proteins?

To address the idea that Golgiphagy may influence cell physiology beyond nutrient stress, we have evaluated the role of Golgiphagy in broader cell physiology. In our previous study examining the involvement of autophagy in proteome remodeling during conversion of human embryonic stem cells (hESCs) cells to induced neurons (iNeurons), we identified Golgi as an organelle that is remodeled over a 12 day time course (PMID: 34699746). When we mined this data, we found that YIPF3/4 are dramatically accumulating in the autophagy mutant (ATG12^{-/-}), much like occurs with ER-phagy receptors. Therefore, to test the possible involvement of YIPF3/4, we genetically edited human embryonic stem cells (hESCs) to delete YIPF4 (which also depletes levels of YIPF3). Upon neuronal differentiation together with cells lacking ATG12 as a control, we employed quantitative proteomics to measure Golgi remodeling that is autophagy or YIPF3/4 dependent. In this context, we observed that Golgi has significant autophagy-based remodeling over 12 days of differentiation, as expected based on the prior study. Importantly YIPF4 knockout neurons display a defect in Golgi remodeling that is both similar to cells lacking ATG12 and has significant overlap in the Golgi proteins that are stabilized in the context of nutrient stress. We feel that this data strongly points to a role for YIPF3/4 in cellular physiology beyond nutrient stress-induced recycling. We have added this data into Fig 5 d-f, and Extended data Fig 12.

- Is the autophaged Golgi apparatus damaged/non-functional (as is the case in mitophagy)?

This Golgiphagy pathway is leveraged under multiple conditions (see above), making it possible that it would be used to clear damaged Golgi as well. In terms of nutrient stress, we believe this recycling is a mechanism to increase both amino acids and lipids for the changing needs of starved cells. One very interesting idea is that defects in the function of specific Golgi enzymes leads to activation of some type of quality control pathway that removes the damaged proteins by Golgiphagy, analogous to what is thought to occur with misfolded collagen being removed by ER-phagy. However, we would submit that elucidating such a new pathway is beyond the scope of the paper. We feel that our finding that YIPF3/4 based Golgi remodeling occurs through differentiation provides an initial glimpse of such a quality control mechanisms.

7. While the study primarily focuses on the identification and characterization of YIPF3 and YIPF4 as receptors for Golgiphagy, it would be necessary to elucidate the functional implications of Golgiphagy in cellular processes. How are core Golgi functions such as protein trafficking, glycosylation or secretion affected by YPF3/4 loss?

We estimate that only ~5-8% of the Golgi is degraded by autophagy over a 12 hour period of nutrient stress. Therefore, ostensibly one would not expect an absence of Golgiphagy to dramatically affect the overall performance of the Golgi in terms of protein modification/secretion. However, assessing this is difficult since starvation will cause a dramatic shutdown in translation, which will greatly affect flux of secreted and membrane proteins through the Golgi complex and make challenging any attempt to measure/interpret an effect of YIPF3/4 deletion.

We did however, assay whether loss of YIPF3 and YIPF4 would affect Golgi morphology. Using immunofluorescence in the YIPF3/4 double KO cells, we do not observe any defects at the level of confocal microscopy (i.e. no evidence of fragmentation, for example) extended data Fig 12a. In addition, we examined whether YIPF3/4 affect the formation of phagosome precursor membranes, ATG9 vesicles, that are derived from the Golgi. Because other forms of autophagy (e.g. ER-phagy) still occur in YIPF3/4 mutants (as noted in the paper), we expect that ATG9 vesicle release from the Golgi is not affected. However, we have directly examined this point using ATG9 immunofluorescence and have added the data to extended data Fig 11a. We still observed abundant ATG9-positive structures in the cytosol, comparable with WT cells, so there does not seem to be an effect on release of ATG9 vesicles from the Golgi upon YIPF3/4 deletion.

8. Are there any diseases associated with YIPF3/4 gene mutations or SNPs? What about other identified CAPs?

There are no known disease mutations within YIPF3/4 that we have been able to identify either in the literature or in disease databases (e.g. <https://www.disgenet.org/>). Within the identified CAPs, there are many Golgi proteins that are associated with diseases, particularly those of glycosylation. Many of these diseases lead to neurodegenerative

phenotypes and cancers. Examples of these proteins include: RXYTL1, EXTL3, CHSY1, FUT8, MGAT2, GALNT2, GALNT3, GOLGA5, GOLGB1, ST3GAL4, GORS1, ATP2C1, TRIP11, TRAPPC11. We would be happy to speculate on the possibility that regulation of one or more of these proteins by Golgiphagy is related to disease. However, with the addition of additional data and the strict word limits, we were not able to add speculative discussion points in the revised manuscript. Perhaps it will be better to address this interesting connection in a future publication that provides insight into disease connections.

Minor points

9. In figure 5, it would be beneficial to see the FIP200 $-/-$ cells (these are included in the supplemental, but would be beneficial to see side-by-side)

We have added this panel into the main Fig 4 to improve the ability for a side by side comparison.

Reviewer Reports on the First Revision:

Referees' comments:

Referee #1 (Remarks to the Author):

The authors adequately addressed my critical point. Therefore, I recommend to accept this manuscript for publication. Congratulation to this great tour de force study!

Referee #2 (Remarks to the Author):

The authors have nicely addressed my comments and I have no further points.

Referee #3 (Remarks to the Author):

The authors have satisfactorily addressed my previous concerns. I support acceptance and rapid publication of this novel and impactful story.

Referee #4 (Remarks to the Author):

A. Summary of the key results

The authors extensively used proteomics to identify the predominate targets of macroautophagy during nutrient stress in mammalian cell culture lines. Interestingly, the authors found that ER and Golgi GO terms were highly enriched in their proteomics datasets comparing control and autophagy knock-out cells in untreated and nutrient starvation. Next, the authors used proximity labeling to identify Golgiphagy receptors in the context of nutrient starvation. From their results, the authors focused on Golgi proteins YIPF3 and YIPF4 as candidate Golgiphagy receptors. Using additional proximity labeling experiments with YIPF3/4, the authors showed that YIPF3/4 proximity to known autophagy factors was dependent on the LIR motifs. In complementary cell biological experiments, the authors demonstrated that YIPF3/4 localized to the Golgi and colocalized with and were degraded by autophagic compartments. Using YIPF4 KO cells, the authors next showed that YIPF4 predominantly affected Golgi resident proteins, further supporting the role of YIPF3/4 as Golgiphagy receptors. Finally, the authors examined the role of YIPF3/4 in Golgi proteome remodeling outside of nutrient starvation, turning instead to in vitro neuronal differentiation. Using proteomics, the authors found that, congruent with their results in immortalized cell lines, Golgi membrane proteins accumulated in YIPF4 KO iNeurons and ATG12 KO iNeurons at similar levels.

B. Originality and significance

Identifying Golgiphagy receptors is novel and significant to the field. The results presented here are of immediate interest to the field. The significance and importance are further supported by the similar BioRxiv preprint.

C. Data & methodology

Data appears broadly of high quality and presented well. To ensure rigor and reproducibility, the authors need to clarify how many cells and biological replicates were used in Figure 4 and Extended Data Figure 11 (see Suggested Improvements for more details). Since cell biological techniques were generally used to confirm validity of the proteomics data, the cell biological experiments also need to be appropriately quantified and reproduced reliably to ensure rigor. The appropriate number of replicates may have been collected, but it was not always apparent in the current manuscript.

D. Appropriate use of statistics and treatment of uncertainties

Figure 4 and Extended Data Figure 12d are missing error bars. Extended Data Figures 9f; 10a, b, d; and 12d are missing statistical significance information. Otherwise, statistics appear to be used and presented appropriately.

E. Conclusions

In general, the cell biological data should be more robust and quantitated to ensure rigor and reproducibility of the findings. As the cell biological experiments are used to confirm the proteomics findings, it is important that they are also robust and rigorous.

F. Suggested improvements

In the last results section about neuronal differentiation, the authors generally appear to discuss more data than is presented in Figure 5 or Extended Data Figure 12. For example, on line 312, the authors report “we observed accumulation of ER and Golgi proteins in ATG12^{-/-} cells through differentiation (Extended Data Fig. 12b).” However, no data is presented comparing differentiated iNeurons to hESCs, although Figure 5d implies that data was acquired. As the data are currently presented, conclusions cannot be drawn about differentiation, since only Day 12 iNeuron data are presented. The authors could include comparisons between Day 12 iNeurons and hESCs for control, ATG12^{-/-} and YIPF4^{-/-}, or the conclusions should be adjusted. Including the comparisons to hESCs would maintain the significance of the authors’ findings.

An important experiment to support YIPF3/4 as Golgiphagy receptors would be to examine autophagosomes in control and YIP3/4 DKO cells by immunofluorescence. In control cells, autophagosomes should contain Golgi membrane proteins. In contrast, in DKO cells, fewer autophagosomes should contain Golgi membrane proteins or autophagosomes should contain lower amount of Golgi membrane proteins.

Figure 4. It is unclear if these experiments have biological replicates – it is not addressed in the text, figure legend, or methods. The experiments presented in Figure 4 need to have biological replicates (≥ 3) and also have additional micrographs quantified (a minimum of 5 per biological replicate) to ensure rigor and reproducibility of the authors’ conclusions. Graphs in e, f and j have a minimum of 3 micrographs quantified for several conditions. It is unclear how many cells were quantified. The relevant significance information is not presented in f; the last column and the middle column should be compared. The inset and line scan indicators appear to be inappropriately moved to the right in panel c.

Lines 242-243/ Extended Data Figure 9c. The authors conclusions are not supported by the data presented in 9c. To support the conclusions as presented (“...depended on the presence of BafA1 to block mNEON quenching...”), the authors would need to present data from 293 cells after 3h of EBSS treatment without BafA. However, as these are fixed cells, it is not clear that it is possible to detect a

change in mNeon quenching in the lysosome, since fixation and permeabilization would alter the acidic environment of the fixed lysosome. It is likely easier to change the conclusions/wording to accurately reflect the data presented.

Extended Data Figure 11 a-b: Do the experiments in 11a have biological replicates? It is not clear from the figure legend. The data for both panels should be quantified to be able to assess the authors' conclusions and ensure rigor and reproducibility of the findings.

G. References

This manuscript references previous literature appropriately.

H. Clarity and context

The abstract is clear, accessible, and appropriately put into the larger context of the field.

Author Rebuttals to First Revision:

INTRODUCTION

We very much appreciate the reviewer's comments on the paper, which have led to significant improvements in the findings and presentation. In this further revision, we have addressed in full the comments of reviewer 4, including placement of statistical information on the figures noted by the reviewer or by changes in the text, as detailed below.

Our responses are provided in blue font.

Referees' comments:

Referee #1 (Remarks to the Author):

The authors adequately addressed my critical point. Therefore, I recommend to accept this manuscript for publication. Congratulation to this great tour de force study!

We thank the reviewer for the positive comments.

Referee #2 (Remarks to the Author):

The authors have nicely addressed my comments and I have no further points.

We thank the reviewer for the positive comments.

Referee #3 (Remarks to the Author):

The authors have satisfactorily addressed my previous concerns. I support acceptance and rapid publication of this novel and impactful story.

We thank the reviewer for the positive comments.

Referee #4 (Remarks to the Author):

A. Summary of the key results

The authors extensively used proteomics to identify the predominate targets of macroautophagy during nutrient stress in mammalian cell culture lines. Interestingly, the authors found that ER and Golgi GO terms were highly enriched in their proteomics datasets comparing control and autophagy knock-out cells in untreated and nutrient starvation. Next, the authors used proximity labeling to identify Golgiphagy receptors in the context of nutrient starvation. From their results, the authors focused on Golgi proteins YIPF3 and YIPF4 as candidate Golgiphagy receptors. Using additional proximity labeling experiments with YIPF3/4, the authors showed that YIPF3/4 proximity to known autophagy factors was dependent on the LIR motifs. In complementary cell biological experiments, the authors demonstrated that YIPF3/4 localized to the Golgi and colocalized with and were degraded by autophagic compartments. Using YIPF4 KO cells, the authors next showed that YIPF4 predominantly affected Golgi resident proteins, further supporting the role of YIPF3/4 as Golgiphagy receptors. Finally, the authors examined the role of YIPF3/4 in Golgi proteome remodeling outside of nutrient starvation, turning instead to in vitro neuronal differentiation. Using proteomics, the authors found that, congruent with their results in immortalized cell lines, Golgi membrane proteins accumulated in YIPF4 KO iNeurons and ATG12 KO iNeurons at similar levels.

B. Originality and significance

Identifying Golgiphagy receptors is novel and significant to the field. The results presented here are of immediate interest to the field. The significance and importance are further supported by the similar BioRxiv preprint.

We appreciate the reviewer noting that the results presented here are “novel and significant” and “of immediate interest” to the field.

C. Data & methodology

Data appears broadly of high quality and presented well. To ensure rigor and reproducibility, the authors need to clarify how many cells and biological replicates were used in Figure 4 and Extended Data Figure 11 (see Suggested Improvements for more details). Since cell biological techniques were generally used to confirm validity of the proteomics data, the cell biological experiments also need to be appropriately quantified and reproduced reliably to ensure rigor. The appropriate number of replicates may have been collected, but it was not always apparent in the current manuscript.

We appreciate the reviewer’s comments and apologize for not making these aspects as clear as we might have. We have now addressed all of the figures and analyses pointed out by the reviewer, as described in detail below.

D. Appropriate use of statistics and treatment of uncertainties

Figure 4 and Extended Data Figure 12d are missing error bars.

We have now added all of the required statistical analysis information in the figure and legend for Figure 4. This includes the number of replicates, the number of cells analyzed, and statistical test information. This is also accompanied by extensive information in the source data table for construction of the image analysis figures themselves. For Fig 12d, we have now added error bars and the statistical test in the figure and legend.

Extended Data Figures 9f; 10a, b, d; and 12d are missing statistical significance information. Otherwise, statistics appear to be used and presented appropriately.

We have added statistical information to the corresponding panels and legends, with additional detail of each statistical test in the source data. See below, where we have addressed these specific comments in the more detailed part F (Suggested improvements) section from the reviewer.

E. Conclusions

In general, the cell biological data should be more robust and quantitated to ensure rigor and reproducibility of the findings. As the cell biological experiments are used to confirm the proteomics findings, it is important that they are also robust and rigorous.

We thank the reviewer for this comment. We have addressed the specific points concerning robustness of the data above and below.

F. Suggested improvements

In the last results section about neuronal differentiation, the authors generally appear to discuss more data than is presented in Figure 5 or Extended Data Figure 12. For example, on line 312, the authors report “we observed accumulation of ER and Golgi proteins in ATG12^{-/-} cells through differentiation (Extended Data Fig. 12b).” However, no data is presented comparing differentiated iNeurons to hESCs, although Figure 5d implies that data was acquired. As the

data are currently presented, conclusions cannot be drawn about differentiation, since only Day 12 iNeuron data are presented. The authors could include comparisons between Day 12 iNeurons and hESCs for control, ATG12^{-/-} and YIPF4^{-/-}, or the conclusions should be adjusted. Including the comparisons to hESCs would maintain the significance of the authors' findings.

We are sorry for the confusion here. The comparison of ES cells with differentiated cells (iNeurons) is actually in the original **Extended Data Fig 12c**. We now realize, however, that the way the section was written might have been confusing and the figure citation might not have been clear. In order to address the concern, we tried to rewrite the section, and have made a direct statement concerning the analysis of pluripotency and differentiation markers.

We now state: "Therefore, to examine the potential involvement of YIPF3/4 in Golgi remodeling beyond nutrient stress, we created YIPF4^{-/-} hESCs, differentiated control, ATG12^{-/-}, and YIPF4^{-/-} hESCs into iNeurons, and quantified proteomes at day 0 and 12 (**Fig. 5d, Extended Data Fig. 12a-c and Supplementary Table 9**). The expected alterations in the abundance pluripotency and neurogenesis factors comparing hESCs with iNeurons were observed in all genotypes, indicating that ATG12 or YIPF4 deletion did not alter differentiation (**Extended Data Fig. 12c and Supplementary Table 9, see METHODS**). Consistent with HeLa cells, YIPF3 levels were reduced in ES cells lacking YIPF4 (**Extended Data Fig. 12d**). As expected³², we observed accumulation of ER and Golgi proteins in ATG12^{-/-} cells through differentiation (**Extended Data Fig. 12b**). Strikingly, YIPF4^{-/-} iNeurons displayed selective accumulation of Golgi membrane proteins to an extent approaching that of ATG12^{-/-} iNeurons (**Fig. 5e, Extended Data Fig. 12b,e**), and with a pattern of accumulation similar to nutrient stress-derived CAPs (**Fig. 5f**). These results highlight broader functions of YIPF3/4 as autophagy-based Golgi remodelers in response to both nutrient stress and cell state changes.

An important experiment to support YIPF3/4 as Golgiphagy receptors would be to examine autophagosomes in control and YIP3/4 DKO cells by immunofluorescence. In control cells, autophagosomes should contain Golgi membrane proteins. In contrast, in DKO cells, fewer autophagosomes should contain Golgi membrane proteins or autophagosomes should contain lower amount of Golgi membrane proteins."

We thank the reviewer for the suggested experiment. Overall, we feel that the wealth of proteomic data address this point in a highly quantitative manner. Because the evidence suggests that there may well be additional pathways beyond YIPF3/4 for delivery of Golgi to autophagosomes (as is the case for ER where multiple receptors have been identified), we feel that imaging-based approaches may not be sufficiently quantitative to fully demonstrate a lower amount of Golgi membrane proteins within autophagosomes. In addition, this is – in essence – what the Keima experiments demonstrate, where we have used GALNT2-Keima to demonstrate a defect in Golgiphagy in cells lacking YIPF3/4.

In addition, we note that a recent BioRxiv paper (<https://doi.org/10.1101/2023.08.09.552599>) (Kitta et al) reported similar findings based on EBSS treatment for starvation and used both proteomics and autophagic flux measurements to characterize YIPF3/4's involvement in Golgiphagy. We now mention in the discussion both this preprint and our earlier 2022 preprint reporting YIPF3/4 as Golgiphagy receptors.

Figure 4. It is unclear if these experiments have biological replicates – it is not addressed in the text, figure legend, or methods. The experiments presented in Figure 4 need to have biological replicates (≥ 3) and also have additional micrographs quantified (a minimum of 5 per biological replicate) to ensure rigor and reproducibility of the authors' conclusions. Graphs in e, f and j

have a minimum of 3 micrographs quantified for several conditions. It is unclear how many cells were quantified.

We thank the reviewer for pointing out that we neglected to explicitly state the replicate, and cell numbers for our imaging data. All micrographs were collected and quantified ($n \geq 3$ biological replicates) and each imaging field was taken with a wide field camera allowing for an average of 20 cells/imaging field. Using triplicate fields for each replicate, we quantified >40 cells per condition. We now included the total number of cells in each condition in the figure and indicate in the legend that the total number of cells is based on three or more biological replicates. We have also included all of this information and the number of cells analyzed in each case in the source data table. In addition we have added error bars to each condition, and additional detail about statistical information within the source data.

The relevant significance information is not presented in f; the last column and the middle column should be compared. The inset and line scan indicators appear to be inappropriately moved to the right in panel c.

We have now added the comparison requested on panel f, which does represent a significant difference. We also very much thank the reviewer for noticing that the position of the linescan had inadvertently moved during rearrangement of the imaging figure (panel c). We have now placed the linescan in the proper location.

Lines 242-243/ Extended Data Figure 9c. The authors conclusions are not supported by the data presented in 9c. To support the conclusions as presented (“...depended on the presence of BafA1 to block mNEON quenching...”), the authors would need to present data from 293 cells after 3h of EBSS treatment without BafA. However, as these are fixed cells, it is not clear that it is possible to detect a change in mNeon quenching in the lysosome, since fixation and permeabilization would alter the acidic environment of the fixed lysosome. It is likely easier to change the conclusions/wording to accurately reflect the data presented.

To address this concern, we have changed the language describing this data. We now state: “Strikingly, within 3h of starvation (EBSS+BafA1), numerous mNEON-YIPF4-positive and YIPF3-positive puncta were observed (**Fig. 4b, Extended Data Fig. 9c**). Importantly, a subset of mNEON-YIPF4 puncta were found to co-localize with LAMP1, indicating trafficking to the lysosome (**Fig. 4c**). Moreover, the appearance of mNEON-YIPF4 puncta required FIP200 and VPS34 (**Fig. 4d-f, Extended Data Fig. 9d**), suggesting an essential role for autophagy in YIPF3/4 capture from Golgi during nutrient stress, as is also seen with ER-phagy receptors²⁶⁻²⁸.”

Extended Data Figure 11 a-b: Do the experiments in 11a have biological replicates? It is not clear from the figure legend. The data for both panels should be quantified to be able to assess the authors’ conclusions and ensure rigor and reproducibility of the findings.

We have added information about the biological replicates into the figure legend ($n \geq 4$). For Extended Data Fig. 11a, we have quantified ATG9 intensity for WT, FIP200^{-/-}, and YIPF3/4^{-/-} genotypes and added this quantification with proper error bars and statistical information into an additional panel together with the panel for Extended Data Fig. 11a.

For Extended Data Fig. 11b, two biological replicates were grown and treated separately but analyzed at the same time on by SDS-PAGE. We have addressed this by adding “Replicate #1 and #2” on the labels for the gel and we now also state in the legend that the experiment was performed in duplicate. To further address this point, we performed a quantification of the blots.

For these particular immunoblotting experiments (as opposed to all others in the paper), chemiluminescence was used to detect the proteins via immunoblotting. As this technique is known to have limitations in the linearity of the signal, quantification of such blots is typically limited to semi-quantitative results. Nevertheless, we performed this analysis as now described in the legend to the figure and include the relative abundance for TEX264, YIPF3, YIPF4, and CALCOCO1. We also note in the legend that the values are based on densitometry analysis of chemiluminescence blots to make it clear that these are only semi-quantitative. However, the results of this analysis fit completely with what is obvious by visual analysis of the blots – no defect in turnover of YIPF3, YIPF4, or TEX264 in the CALCOCO1^{-/-} cells.

G. References

This manuscript references previous literature appropriately.

We appreciate the reviewer's positive comment.

H. Clarity and context

The abstract is clear, accessible, and appropriately put into the larger context of the field.

We appreciate the reviewer's positive comment.

Reviewer Reports on the Second Revision:

Referees' comments:

Referee #4 (Remarks to the Author):

The authors addressed my comments sufficiently. Importantly, errors bars, biological replicates, and statistical tests are now appropriately included in all panels and figures.

Author Rebuttals to Second Revision:

Referee #4 (Remarks to the Author):

The authors addressed my comments sufficiently. Importantly, errors bars, biological replicates, and statistical tests are now appropriately included in all panels and figures.

We thank the reviewer for the positive comments.